# A Primal-Dual-Assisted Penalty Approach to Bilevel Optimization with Coupled Constraints

**Liuyuan Jiang**[†]**, Quan Xiao**[†]**, Victor M. Tenorio**[⋆]**, Fernando Real-Rojas**[⋆]
**Antonio G. Marques**[⋆]**, Tianyi Chen**[†]

[†]Rensselaer Polytechnic Institute, Troy, NY, United States
[⋆]King Juan Carlos University, Madrid, Spain
{jiangl7, xiaoq5, chent18}@rpi.edu
{victor.tenorio, antonio.garcia.marques}@urjc.es; f.real.2018@alumnos.urjc.es

## Abstract

Interest in bilevel optimization has grown in recent years, partially due to its applications to tackle challenging machine-learning problems. Several exciting recent works have been centered around developing efficient gradient-based algorithms that can solve bilevel optimization problems with provable guarantees. However, the existing literature mainly focuses on bilevel problems either without constraints, or featuring only simple constraints that do not couple variables across the upper and lower-levels, excluding a range of complex applications. Our paper studies this challenging but less explored scenario and develops a (fully) first-order algorithm, which we term **BLOCC**, to tackle **Bi**Level **O**ptimization problems with **C**oupled **C**onstraints. We establish rigorous convergence theory for the proposed algorithm and demonstrate its effectiveness on two well-known real-world applications - hyperparameter selection in support vector machine (SVM) and infrastructure planning in transportation networks using the real data from the city of Seville.

## 1 Introduction

Bilevel optimization (BLO) approaches are pertinent in various machine learning problems, including hyperparameter optimization [49, 24], meta-learning [22], and reinforcement learning [67, 63]. Moreover, the ability to handle BLO with constraints is particularly important, as these constraints appear in applications such as pricing [15], transportation [50, 1], and kernelized SVM [30]. Although there is extensive research on BLO problems without constraints or with uncoupled constraints [31, 12, 39, 42, 62], solutions for BLO problems with coupled constraints (CCs) remain limited; see details in Table 1. However, it is of particular interest to investigate BLO with lower-level CCs. Taking infrastructure planning in a transportation network as an example, the lower-level seeks to optimize a utility constrained by the upper-level parameter, network configuration.

Motivated by this, we consider the coupled-constrained BLO problem in the following form

$$\min_{x \in \mathcal{X}} f(x, y_g^*(x)) \tag{1a}$$

$$\text{s.t.} \quad y_g^*(x) := \arg\min_{y \in \mathcal{Y}(x)} g(x, y) \quad \text{with} \quad \mathcal{Y}(x) := \{y \in \mathcal{Y} : g^c(x, y) \le 0\} \tag{1b}$$

where $f : \mathbb{R}^{d_x} \times \mathbb{R}^{d_y} \to \mathbb{R}$ is the upper-level objective, $g : \mathbb{R}^{d_x} \times \mathbb{R}^{d_y} \to \mathbb{R}$ is the lower-level

---

The work was supported by NSF project 2412486, NSF SCALE-MoDL project 2401297, NSF CAREER project 2047177, Cisco Research Award, Amazon Research Award, the IBM-Rensselaer Future of Computing Research Collaboration, and the Comunidad de Madrid (via URJC grants F840 and F978).

38th Conference on Neural Information Processing Systems (NeurIPS 2024).

objective which is strongly convex, $g^c(x,y) : \mathbb{R}^{d_x} \times \mathbb{R}^{d_y} \to \mathbb{R}^{d_c}$ defines the lower-level CCs, and $\mathcal{X} \subseteq \mathbb{R}^{d_x}, \mathcal{Y} \subseteq \mathbb{R}^{d_y}$ are the domain of $x$ and $y$ that are easy to project, such as the Euclidean ball.

The challenge in solving (1) arises from the coupling of the upper and lower-level problems. Prior work addressed this by starting with the unconstrained BLO problem, using implicit gradient descent (IGD) methods [27, 31, 34, 12, 13, 40, 61, 43, 66] and penalty-based methods [45, 44, 61, 41, 42]. To solve BLO with CCs, AiPOD [71] and GAM [72] investigated the IGD method under different constraint settings. However, AiPOD only considered equality constraints, and GAM lacked finite-time convergence guarantees. Leveraging a penalty reformulation, [75] developed a Hessian-free method with finite-time convergence. However, a key algorithm step in [75] is a joint projection of the current iterate $(x, y)$ onto the coupled constraint set. This projection, required at each iteration, becomes particularly challenging when $g^c(x, y)$ is not jointly convex and can be computationally expensive for large-scale problems with a high number of variables $(d_x, d_y)$ or constraints $(d_c)$.

To this end, this paper aims to address the following question

*Can we develop an efficient algorithm that bypasses joint projections on $g^c(x, y)$ and quickly solves the BLO problem with coupled inequality constraints in (1)?*

We address this question affirmatively, focusing on the setting where the lower-level objective, $g(x, y)$, is strongly convex in $y$, and the constraints $g^c(x, y)$ are convex in $y$. To avoid implementing a joint projection, we put forth a novel single-level primal-dual-assisted penalty reformulation that decouples $x$ and $y$. Specifically, with $\mu \in \mathbb{R}^{d_c}$ denoting the Lagrange multiplier of (1b), we propose solving

$$\min_{x \in \mathcal{X}} \quad F_\gamma(x) := \max_{\mu \in \mathbb{R}_+^{d_x}} \min_{y \in \mathcal{Y}} f(x, y) + \underbrace{\gamma(g(x, y) - v(x))}_{\text{penalty term}} + \underbrace{\langle \mu, g^c(x, y) \rangle}_{\text{Lagrangian term}} \tag{2a}$$

$$\text{where} \quad v(x) := \min_{y \in \mathcal{Y}(x)} g(x, y) \tag{2b}$$

where the penalty constant $\gamma$ controls the distance between $y$ and $y_g^*(x)$ by penalizing $g(x, y)$ to its value function $v(x)$, and the Lagrangian term penalizes the constraint violation of $g^c(x, y)$.

However, recognizing the max-min subproblems involved in (2a), it becomes computationally costly to evaluate the penalty function $F_\gamma(x)$ and its gradient. To this end, we pose the following question

*Can we develop efficient algorithms to solve the max-min subproblem and evaluate $\nabla F_\gamma(x)$?*

We answer this question by proving that this reformulation exhibits several favorable properties, including smoothness. These properties are critical for designing gradient-based algorithms and characterizing their performance. However, the presence of the CCs renders the calculation of the gradient $\nabla F_\gamma(x)$ more challenging than for its unconstrained counterpart [62]. Building upon this, we design a primal-dual gradient method with rigorous convergence guarantees for the BLO with general inequality CCs, and provide an improved result for the case of $g^c$ being affine in $y$.

## 1.1 Main contributions

In a nutshell, our main contributions are outlined below.

**C1)** In Section 2, leveraging the Lagrangian duality theorem, we introduce the function $F_\gamma(x)$ in (2a) as a penalty-based reformulation of (1), establish the continuity and smoothness of $F_\gamma(x)$, and develop a novel way to compute its gradient.

**C2)** In Section 3, we develop BLOCC, a fully first-order algorithm to tackle BLO problems with CCs. With $\epsilon$ being the target error for the generalized gradient norm square of $F_\gamma(x)$, we establish that the iteration complexity under the generic constraint $g^c(x, y)$ in (1b) is $\tilde{\mathcal{O}}(\epsilon^{-2.5})$. We establish, for the first time, the linear convergence of a strongly convex-concave max-min problem with linear interaction and a constrained maximization parameter, reducing BLOCC's complexity to $\tilde{\mathcal{O}}(\epsilon^{-1.5})$ when the constraint $g^c(x, y)$ is affine in $y$.

**C3)** In Section 4, we apply our BLOCC algorithm to two real-world applications: SVM model training and transportation network planning. By comparison with LV-HBA [75] and GAM [72], we demonstrate the algorithm's effectiveness and its robustness to large-scale problems.

| | LL constraint | First Order | Complexity |
|---|---|---|---|
| **BLOCC** | $g^c(x, y) \leq 0$ convex in $y$ and LICQ holds; Special case: $g^c(x, y)$ affine in $y$ | ✓ | $\tilde{\mathcal{O}}(\epsilon^{-2.5})$; $\tilde{\mathcal{O}}(\epsilon^{-1.5})$ |
| LV-HBA | $g^c(x, y) \leq 0$ convex in $x \times y$ | ✓ | $\mathcal{O}(\epsilon^{-3})$ |
| GAM | $g^c(x, y) \leq 0$ convex in $x \times y$ and LICQ holds | ✗ | ✗ |
| BVFSM | $g^c(x, y) \leq 0$ satisfying other requirements | ✓ | ✗ |
| AiPOD | $g^c(x, y) = Ay - b(x) = 0$ | ✗ | $\mathcal{O}(\epsilon^{-1.5})$ |

Table 1: Comparison of our work with LV-HBA [75], BVFSM [46], AiPOD [71], and Gradient Approximation method (GAM) [72]. LL convergence is on metric the squared distance of $y_t$ to its optimal solution, and UL convergence is on squared (generalized) gradient norm.

## 1.2 Related works

BLO has a long history, dating back to the seminal work of [8]. It has inspired a rich body of literature, e.g., [76, 68, 14, 65]. The recent focus on BLO is centered on developing efficient gradient-based approaches with provable finite-time guarantees.

**Methods for BLO without constraints.** A cluster of BLO gradient-based approaches gravitates around the implicit gradient descent (IGD) method [55], where the key idea is to approximate the hypergradient by the implicit function theorem. The finite-time convergence of IGD was first established in [27] for unconstrained strongly-convex lower-level problems. Subsequent works improved the convergence rates and/or relaxed the assumptions under various settings; see [31, 34, 12, 13, 40, 61, 43, 66, 73, 35, 70]. Another cluster of works is based on iterative differentiation (ITD) [49, 23, 53, 60], which estimates the hypergradient by differentiating the entire iterative algorithm used to solve the lower-level problem with respect to the upper-level variables. The finite-time guarantee was first established in [28, 47, 33, 6]. Viewing the lower-level problem as a constraint such as in [64], penalty-based methods have also emerged as a promising approach for BLO. Dated back to [77], this line of works [45, 51, 44, 41, 62, 25, 48] reformulated the original BLO as the single-level problems with various penalty terms and leveraged first-order methods to solve them.

**BLO with constraints.** While substantial progress has been made for unconstrained BLO, the analysis for constrained BLO is more limited. Upper-level constraints of the form $x \in \mathcal{X}$ were considered in [31, 12]. For the lower-level uncoupled constraint, SIGD [39] considered the uncoupled constraint $Ay \leq b$ and achieved asymptotic convergence, [42, 62] employed penalty reformulation and considered both upper and lower uncoupled constraints. However, the literature on **BLO with CCs** is scarce. BVFSM [46] conducted a penalty-based method to avoid the calculation of the Hessian, as IGD methods do. However, only asymptotic convergence was achieved. GAM [72] investigated the IGD method under inequality constraints while failing to provide finite-time convergence results as well. AiPOD [71] also applied IGD and successfully achieved finite-time convergence, but it only considered equality constraints. LV-HBA [75] considered inequality constraints and constructed a penalty-based reformulation. However, it employed a joint projection of $(x, y)$ onto $\{\mathcal{X} \times \mathcal{Y} : g^c(x, y) \leq 0\}$ which is computationally inefficient when there are many constraints or when $g^c(x, y)$ is not jointly convex. After our initial submission, we found a concurrent work [74] posted on ArXiv, which used Lagrange duality theory differently from ours, applying it to construct a new smoothed penalty term. However, it does not quantify the relationship between the relaxation of this penalty and the relaxation of lower-level optimality, and it does not guarantee lower-level feasibility. We summarized prior works on BLO with lower-level CCs in Table 1.

## 2 Primal-dual Penalty-based Reformulation

In this section, our goal is to construct a primal-dual-assisted penalty reformulation for our BLOCC problem. The technical challenge comes from finding a suitable penalty function for BLO with CCs.

### 2.1 The challenges in BLO with coupled constraints

Here, we will elaborate on the two technical challenges of BLO with CCs.

The *first challenge* associated with the presence of CCs is the difficulty to find the descent direction for $x$, which involves finding the closed-form expression of the gradient $\nabla v(x)$. The expression without CCs, $\nabla v(x) = \nabla_x g(x, y_g^*(x))$ provided in [62, 42, 41, 44], is not applicable.

For example, when $g(x, y) = (y - 2x)^2$ and $g^c(x, y) = 3x - y$, the optimal lower-level solution is $y_g^*(x) = 3x$ and thus, $v(x) = x^2$ with $\nabla v(x) = 2x$. However, $\nabla_x g(x, y_g^*(x)) = -4x \neq \nabla v(x)$. The closed form gradient for BLO with CCs should be (8), which considers a Lagrange term that will be illustrated later in this paper. In Figure 1, we present the gradient $\nabla v(x)$ without the Lagrange multiplier in [62, 42, 41, 44] and ours with the Lagrange term. In this example, the gradient without the Lagrange term leads to the opposite direction to the true gradient.

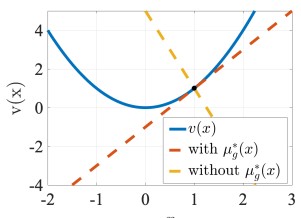

Figure 1: Calculation of $\nabla v(x)$. The blue line is $v(x)$, the the yellow dashed line is calculated by the formulation given in [62, 41], while red dashed line is derived by our BLOCC. It can be seen that $\nabla v(x)$ without the Lagrange multiplier is very biased from the true gradient.

The *second challenge* associated with the presence of CCs is the difficulty of performing the joint projection. If we directly extend the penalty reformulation in [62, 41], we could treat the coupling constraint set $\mathcal{Y}(x)$ as a joint constraint $\{(x, y) : g^c(x, y) \leq 0\}$, and employ a joint projection of $(x, y)$ to ensure the feasibility. However, this can be computationally inefficient when the problem is large-scale, i.e. the number of variables or constraints is large. The detailed analysis of computational cost can be seen in Appendix H. Moreover, when $g^c(x, y)$ is complex, the projection may not have a closed form and may not even be well-defined if $g^c(x, y)$ is not jointly convex.

## 2.2 The Lagrangian duality-based penalty reformulation

Before proceeding, we summarize the assumptions considered as follows.

**Assumption 1** (Lipschitz Continuity). *Assume that $f$, $\nabla f$, $\nabla g$, $g^c$ and $\nabla g^c$, and are respectively $l_{f,0}$, $l_{f,1}$, $l_{g,1}$, $l_{g^c,0}$, and $l_{g^c,1}$-Lipschitz jointly over $(x, y) \in \mathcal{X} \times \mathcal{Y}$, and $g$ is $l_{g_x,0}$-Lipschitz in $x \in \mathcal{X}$.*

**Assumption 2** (Convexity in $y$). *For any given $x \in \mathcal{X}$, $g(x, y)$ and $g^c(x, y)$ are $\alpha_g$-strongly convex and convex in $y \in \mathcal{Y}$, respectively.*

**Assumption 3** (Domain Feasibility). *Domain $\mathcal{X} \subseteq \mathbb{R}^{d_x}$ and $\mathcal{Y} \subseteq \mathbb{R}^{d_y}$ are non-empty, closed, and convex. For any given $x \in \mathcal{X}$, $\mathcal{Y}(x) := \{y \in \mathcal{Y} : g^c(x, y) \leq 0\}$ is non-empty.*

**Assumption 4** (Constraint Qualification). *For any given $x \in \mathcal{X}$, $g^c$ satisfies the Linear Independence Constraint Qualification (LICQ) condition for every $y\mathcal{Y}$ in a neighborhood of $y_g^*(x)$.*

The Lipschitz continuity in Assumption 1 and the strong convexity of $g$ over $y$ in Assumption 2 are conventional [27, 31, 41, 71, 35, 13]. Moreover, we only require $g^c(x, y)$ to be convex in $y$ rather than: i) jointly convex in $(x, y)$ as in [75], or ii) linear as in [39, 71]. Assumption 3 pertains to the convexity and closeness of the domain, which is also conventional, and Assumption 4 is a standard constraint qualification condition.

To build a penalty reformulation for BLO with lower-level CCs, the first challenge is to find a penalty that regulates $\|y - y_g^*(x)\|$. In the following lemma, we show that $g(x, y) - v(x)$ is a good choice.

**Lemma 1.** *Suppose that Assumptions 2-4 hold and $v(x)$ is defined as in* (2b)*. Then, it holds that*

*c1)* $g(x, y) - v(x) \geq \frac{\alpha_g'}{2}\|y - y_g^*(x)\|^2$*, for some $\alpha_g' > 0$, for all $y \in \mathcal{Y}(x)$; and*

*c2)* $g(x, y) = v(x)$ *if and only if $y = y_g^*(x)$, for all $y \in \mathcal{Y}(x)$.*

The technical challenge of proving this lemma lies in showing the quadratic growth property as in c1) of Lemma 1. For BLO without the CCs, this naturally holds as the lower-level objective $g(x, y)$ is strongly convex in $y$. For BLO with CCs, one needs to use the Lagrangian duality theorem. The full proof of Lemma 1 is given in Appendix B.1, where the key challenge is to show the invariance of the modulus of strong convexity in the Lagrangian reformulated lower-level objective.

With $\gamma$ denoting a penalty constant, we consider the following penalty reformulation:

$$\min_{(x,y)\in\{\mathcal{X}\times\mathcal{Y}:g^c(x,y)\leq 0\}} f(x, y) + \gamma(g(x, y) - v(x)) \tag{3}$$

where the penalty term confines the squared Euclidean distance from $y$ to $y_g^*(x)$. Furthermore, to avoid projecting $(x, y)$ onto the $\{\mathcal{X} \times \mathcal{Y} : g^c(x, y) \leq 0\}$, we propose the primal-dual-assisted penalty

reformulation, which was defined in (2). The approximate equivalence of the reformulation (2) to the original BLO problem (1) is established in the following theorem.

**Theorem 1** (Equivalence). *Suppose that $f$ is Lipschitz in $y$ and $\nabla f$ is $l_{f,1}$-Lipschitz in $(x,y)$ in Assumption 1 hold, and Assumptions 2-4 hold. Then, solving the $\epsilon$-approximation problem of* (1):

$$\min_{(x,y)\in\{\mathcal{X}\times\mathcal{Y}:g^c(x,y)\leq 0\}} f(x,y) \quad s.t. \quad \|y - y_g^*(x)\|^2 \leq \epsilon, \tag{4}$$

*is equivalent to solve the primal-dual penalty reformulation in* (2) *with $\gamma = \mathcal{O}(\epsilon^{-0.5})$ and $\gamma > \frac{l_{f,1}}{\alpha_g}$.*

The detailed proof is provided in Appendix B.2. By setting $\gamma = \mathcal{O}(\epsilon^{-0.5})$, we effectively state the equivalence between the penalty reformulation in (3) and the approximated original problem (4). We can then decouple the joint minimization on $(x,y)$ to a min-min problem on $x$ and $g^c$ constrained $y$. For the inner minimization problem in $y$, we choose $\gamma$ in a way that $\gamma\alpha_g - l_{f,1} > 0$ holds. This ensures the objective in (2a) being strongly convex in $y$, as $l_{f,1}$-smoothness ensures a lower bound for negative curvature of $f(x,y)$. Furthermore, the convexity of $g^c(x,y)$ in $y$ validates the strong duality theorem in [59, 32], thereby enabling the max-min primal-dual reformulation in (2).

## 2.3 Smoothness of the penalty reformulation

To evaluate $F_\gamma(x)$ defined in (2a), we can find the solution to the inner max-min problem:

$$(\mu_F^*(x), y_F^*(x)) := \arg\max_{\mu\in\mathbb{R}_+^{d_x}}\min_{y\in\mathcal{Y}} \underbrace{f(x,y) + \gamma(g(x,y) - v(x)) + \langle\mu, g^c(x,y)\rangle}_{=:L_F(\mu,y;x)}. \tag{5}$$

The uniqueness of $y_F^*(x)$ and $\mu_F^*(x)$ is guaranteed under Assumptions 2 and 4 (see Lemma 5 in Appendix A). Therefore, $F_\gamma(x)$ in (2a) can be evaluated using the unique optimal solutions by

$$F_\gamma(x) = L_F(\mu_F^*(x), y_F^*(x); x). \tag{6}$$

Similarly, we can evaluate $v(x) = L_g(\mu_g^*(x), y_g^*(x); x)$, where

$$(\mu_g^*(x), y_g^*(x)) := \arg\max_{\mu\in\mathbb{R}_+^{d_x}}\min_{y\in\mathcal{Y}} \underbrace{g(x,y) + \langle\mu, g^c(x,y)\rangle}_{=:L_g(\mu,y;x)}. \tag{7}$$

The penalty reformulation $F_\gamma(x)$ can hardly be convex, as $-v(x)$ may be concave, even when $g(x,y) = g(y)$ and $g^c(x,y) = A^\top y - x$; see Lemma 4.24 in [59]. Instead, in the subsequent lemma, we not only show that $v(x)$ is differentiable, but also provide a closed-form expression of $\nabla v(x)$.

**Lemma 2** (Danskin-like theorem for $v(x)$). *Suppose that Assumptions 1–4 hold, and let $B_g < \infty$ be a constant such that $\|\mu_g^*(x)\| < B_g$ for all $x \in \mathcal{X}$. Then, it holds that*

*1. $y_g^*(x)$ and $\mu_g^*(x)$ defined in (7) are $L_g$-Lipschitz for some finite constant $L_g \geq 0$.*

*2. $v(x)$ defined in (2b) is $l_{v,1}$-smooth where $l_{v,1} \leq (l_{g,1} + B_g l_{g^c,1})(1 + L_g) + l_{g^c,0}L_g$ and*

$$\nabla v(x) = \nabla_x g(x, y_g^*(x)) + \langle\mu_g^*(x), \nabla_x g^c(x, y_g^*(x))\rangle. \tag{8}$$

The assumption of the existence of the upper bound for the Lagrange multiplier is a consequence of the LICQ condition [69, Theorem 1]. This assumption is mild and traditional [75]. Finding $\nabla v(x)$ is crucial for the design of a gradient-based method to solve $\min_{x\in\mathcal{X}} F_\gamma(x)$. Leveraging Lemma 2, the gradient $\nabla F_\gamma(x)$ can be obtained by the next lemma.

**Lemma 3** (Danskin-like theorem for $F_\gamma(x)$). *Suppose that the conditions in Lemma 2 hold. Moreover, assume that $\gamma > \frac{l_{f,1}}{\alpha_g}$, and there exist $B_F < \infty$ such that $\|\mu_F^*(x)\| < B_F$, $\forall x \in \mathcal{X}$. Then, it holds that*

*1. $y_F^*(x)$ and $\mu_F^*(x)$ defined in (5) are $L_F$-Lipschitz for some constant $L_F \geq 0$.*

*2. $F_\gamma(x)$ is $l_{F,1}$-smooth with $l_{F,1} \leq (l_{f,1} + \gamma l_{g,1} + B_F l_{g^c,1})(1 + L_F) + \gamma l_{v,1} + l_{f^c,0}L_F$, and*

$$\nabla F_\gamma(x) = \nabla_x f(x, y_F^*(x)) + \gamma\left(\nabla_x g(x, y_F^*(x)) - \nabla v(x)\right) + \langle\mu_F^*(x), \nabla_x g^c(x, y_F^*(x))\rangle. \tag{9}$$

The proof of Lemmas 2 and 3 is provided in Appendix C. Similar to [62], the Danskin-like theorems in Lemmas 2 and 3 rely on the Lipschitzness of the solutions in (5) and (7). Different from BLO without CCs [62], the results here also hinge on the Lagrange multipliers $\mu_g^*(x)$ and $\mu_F^*(x)$.

# 3 Main Results

We will first introduce an algorithm tailored for **Bi**Level **O**ptimization problems with inequality **C**oupled **C**onstraints, and present its convergence analysis. In Section 3.2, we will propose a primal-dual solver for the inner max-min problems and characterize the overall convergence.

## 3.1 BLOCC algorithm

As $F_\gamma(x)$ features differentiability and smoothness, we can apply a projected gradient descent (PGD)-based method to solve $\min_{x \in \mathcal{X}} F_\gamma(x)$. At iteration $t$, update

$$x_{t+1} = \text{Proj}_{\mathcal{X}}(x_t - \eta g_{F,t}) \qquad (10)$$

with stepsize $\eta$ and $g_{F,t}$ as an estimate of $\nabla F_\gamma(x_t)$.

In the previous section, we have obtained the closed-form expressions of $\nabla v(x)$ in (8) and $\nabla F_\gamma(x)$ in (9). Evaluating the closed-form expression requires finding $(y_g^*(x_t), \mu_g^*(x_t))$ and $(y_F^*(x_t), \mu_F^*(x_t))$, the solutions to the max-min problem $L_g(\mu, y; x)$ in (7) and $L_F(\mu, y; x)$ in (5) respectively.

---

**Algorithm 1** Meta algorithm: BLOCC

1: **inputs:** initial points $x_0, y_{g,0}, \mu_{g,0}, y_{F,0}, \mu_{F,0}$; stepsize $\eta, \eta_g, \eta_F$; counters $T_g, T_F$.
2: **for** $t = 0, 1, \ldots, T$ **do**
3: $\quad (y_{g,t}^{T_g}, \mu_{g,t}^{T_g}) = \text{MaxMin}(T_g, T_y^g)$
4: $\quad (y_{F,t}^{T_F}, \mu_{F,t}^{T_F}) = \text{MaxMin}(T_F, T_y^F)$
5: $\quad$ update $x_{t+1}$ via (10)
$\qquad\qquad\qquad\qquad \triangleright$ with $g_{F,t}$ in (13)
6: **end for**
7: **outputs:** $(x_T, y_{F,T}^{T_F})$

---

Given $x_t \in \mathcal{X}$, we can use some minmax optimization solvers with $T_g$ iterations on (7) and $T_F$ iterations on (5) to find an $\epsilon_g$-solution $(y_{g,t}^{T_g}, \mu_{g,t}^{T_g})$ and an $\epsilon_F$-solution $(y_{F,t}^{T_F}, \mu_{F,t}^{T_F})$ satisfying

$$\|(y_{g,t}^{T_g}, \mu_{g,t}^{T_g}) - (y_g^*(x_t), \mu_g^*(x_t))\|^2 = \mathcal{O}(\epsilon_g); \quad \|(y_{F,t}^{T_F}, \mu_{F,t}^{T_F}) - (y_F^*(x_t), \mu_F^*(x_t))\|^2 = \mathcal{O}(\epsilon_F) \quad (11)$$

for target estimation accuracy $\epsilon_g, \epsilon_F > 0$. Such an effective minmax optimization solver will be introduced in the following Section 3.2. In this way, $\nabla v(x_t)$ in (9) can be estimated as

$$g_{v,t} = \nabla_x g(x_t, y_{g,t}^{T_g}) + \langle \mu_{g,t}^{T_g}, \nabla_x g^c(x_t, y_{g,t}^{T_g}) \rangle. \qquad (12)$$

Leveraging $g_{v,t}$, the gradient $\nabla F_\gamma(x)$ can be estimated via (8) as

$$g_{F,t} = \nabla_x f(x_t, y_{F,t}^{T_F}) + \gamma \left( \nabla_x g(x_t, y_{F,t}^{T_F}) - g_{v,t} \right) + \langle \mu_F^{T_F}, \nabla_x g^c(x, y_{F,t}^{T_F}) \rangle. \qquad (13)$$

We summarize the oracle for finding $\min F_\gamma(x)$ as in Algorithm 1, which we term BLOCC, an algorithm designed for BiLevel Optimization with Coupled Constraints. Notably, our BLOCC algorithm can be seamlessly integrated with any MaxMin Solver (or min-max solver) that converges to the optimal solutions of the max-min subproblems by achieving (11). In the following, we present the convergence result of it allowing estimation error $\epsilon_g, \epsilon_F > 0$ from the MaxMin Solver.

**Theorem 2.** *Suppose that the assumptions in Lemma 3 hold. Run Algorithm 1 with some effective inner MaxMin solver to find $(y_{g,t}^{T_g}, \mu_{g,t}^{T_g})$ and $(y_{F,t}^{T_F}, \mu_{F,t}^{T_F})$ respectively $\mathcal{O}(\epsilon_g)$ and $\mathcal{O}(\epsilon_F)$-optimal in squared distance as in* (11)*. Set $\eta \leq \frac{1}{l_{F,1}}$ with some $l_{F,1}$ defined in Lemma 3. It then holds that*

$$\frac{1}{T} \sum_{t=0}^{T-1} \|G_\eta(x_t)\|^2 := \frac{1}{T\eta^2} \sum_{t=0}^{T-1} \|(x_{t+1} - x_t)\|^2 = \mathcal{O}(\gamma T^{-1} + \gamma^2 \epsilon_F + \gamma^2 \epsilon_g). \qquad (14)$$

The proof is available in Appendix D.1. Using the projected gradient $G_\eta = \eta^{-1}(x_{t+1} - x_t)$ as the convergence metric for constrained optimization problems is standard [26]. The term $\mathcal{O}(\gamma^2 \epsilon_F + \gamma^2 \epsilon_g)$ arises from estimation errors using specific MaxMin Solver. Although the inner oracle can be any one that achieves (11), we value the computational effectiveness and therefore present a particular efficient solver Algorithm 2 in the following section.

## 3.2 MaxMin Solver for the BLO with inequality CCs

In this section, we specify the MaxMin solver in Algorithm 1. By viewing $L_g(\mu, y; x)$ in (7) and $L_F(\mu, y; x)$ in (5) as $L(\mu, y)$ for fixed given $x \in \mathcal{X}$, we consider the following max-min problem

$$\max_{\mu \in \mathbb{R}_+^{d_c}} \min_{y \in \mathcal{Y}} L(\mu, y) \qquad (15)$$

which is concave (linear) in $\mu$ and strongly convex in $y$. We can evaluate the dual function of $L(\mu, y)$ defined below by finding $y_\mu^*(\mu)$.

$$D(\mu) := \min_{y \in \mathcal{Y}} L(\mu, y) = L(\mu, y_\mu^*(\mu)) \quad (16)$$

where $\quad y_\mu^*(\mu) := \arg\min_{y \in \mathcal{Y}} L(\mu, y).$

According to Danskin's theorem, we have $\nabla D(\mu) = \nabla_\mu L(\mu, y_\mu^*(\mu))$. Taking either $L_g(\mu, y; x)$ or $L_F(\mu, y; x)$ as $L(\mu, y)$ for given $x \in \mathcal{X}$, $D(\mu)$ defined as (16) exhibits favorable properties namely smoothness and concavity, with details illustrated in Lemma 10 in Appendix D.2. We can, therefore, apply accelerated gradient methods designed for smooth and convex functions such as [52, 4].

---

**Algorithm 2** Subroutine on $\mathsf{MaxMin}(T, T_y)$

1: **inputs:** initial points $y_0$, $\mu_0$; stepsizes $\eta_1, \eta_2$; counters $T, T_y$. Blue part is the version with acceleration; and red part is without.
2: **for** $t = 0, \ldots, T-1$ **do**
3: update $\mu_{t+\frac{1}{2}}$ via (17) **or** $\mu_{t+\frac{1}{2}} = \mu_t$
4: **for** $t_y = 0, \ldots, T_y - 1$ **do**
5: update $y_{t,t_y+1}$ via (18) $\quad \triangleright$ set $y_{t,0} = y_t$
6: **end for**
7: update $\mu_{t+1}$ via (19) $\quad \triangleright$ set $y_{t+1} = y_{t,T_y}$
8: **end for**
9: **outputs:** $(y_T, \mu_T)$

---

At each iteration $t$, we first perform a momentum-based update step to update $\mu_{t+\frac{1}{2}}$ as

$$\mu_{t+\frac{1}{2}} = \mu_t + \frac{t-1}{t+2}(\mu_t - \mu_{t-1}), \quad \text{with } \mu_{-1} = \mu_0. \quad (17)$$

To evaluate $\nabla D(\mu_{t+\frac{1}{2}}) = \nabla_\mu L(\mu_{t+\frac{1}{2}}, y_\mu^*(\mu_{t+\frac{1}{2}}))$, with an arbitrary small target accuracy $\epsilon > 0$, we can run $T_y = \mathcal{O}(\ln(\epsilon^{-1}))$ PGD steps on $L(\mu_{t+\frac{1}{2}}, y)$ in $y$ via

$$y_{t,t_y+1} = \text{Proj}_{\mathcal{Y}}\left(y_{t,t_y} - \eta_1 \nabla_y L(\mu_{t+\frac{1}{2}}, y_{t,t_y})\right). \quad (18)$$

Defining the output after $T_y$ iterations as $y_{t+1}$, since strongly convexity of $L(\mu, \cdot)$ ensures that PGD converges linearly [9, Theorem 3.10], it implies that $\|y_{t+1} - y_\mu^*(\mu_{t+\frac{1}{2}})\| = \mathcal{O}(\epsilon)$. We can conduct

$$\mu_{t+1} = \text{Proj}_{\mathbb{R}_+^{d_c}}(\mu_{t+\frac{1}{2}} + \eta_2 \nabla_\mu L(\mu_{t+\frac{1}{2}}, y_{t+1})). \quad (19)$$

We summarize this oracle in Algorithm 2 based on an accelerated method [52]. When we skip the momentum update, i.e. setting $\mu_{t+\frac{1}{2}} = \mu_t$, it is a simple PGD method on $D(\mu)$.

However, for a convex function $-D(\mu)$, the standard results only provide convergence of the function value, i.e. $\max_{\mu \in \mathbb{R}_+^{d_c}} D(\mu) - D(\mu_t) \overset{t \to \infty}{\longrightarrow} 0$. To establish the convergence of $\|\mu_{g,t}^{T_g} - \mu_g^*(x_t)\|$ and $\|\mu_{F,t}^{T_F} - \mu_F^*(x_t)\|$, we define the dual functions associated with inner problems (7) and (5) as

$$D_g(\mu) = \min_{y \in \mathcal{Y}} L_g(\mu, y; x) \quad \text{and} \quad D_F(\mu) = \min_{y \in \mathcal{Y}} L_F(\mu, y; x) \quad (20)$$

and make the following curvature assumption near the optimum.

**Assumption 5.** *There exist $\delta_g, \delta_F > 0$ and $C_{\delta_g}, C_{\delta_F} > 0$ such that*

$$\langle -\nabla D_g(\mu) + \nabla D_g(\mu_g^*(x)), \mu - \mu_g^*(x)\rangle \geq C_{\delta_g}\|\mu - \mu_g^*(x)\|^2, \quad \forall \mu \in \mathcal{B}(\mu_g^*(x); \delta_g), \quad (21a)$$

$$\langle -\nabla D_F(\mu) + \nabla D_F(\mu_F^*(x)), \mu - \mu_F^*(x)\rangle \geq C_{\delta_F}\|\mu - \mu_F^*(x)\|^2, \quad \forall \mu \in \mathcal{B}(\mu_F^*(x); \delta_F). \quad (21b)$$

It is worth noting that $\langle -\nabla D_g(\mu) + \nabla D_g(\mu_g^*(x)), \mu - \mu_g^*(x)\rangle \geq 0$ holds for all $\mu$ due to concavity and in a neighborhood of the optimal, the equality only happens at the optimal due to the uniqueness of $\mu_g^*(x)$. The same argument applies to $D_F$ due to the concavity of the dual functions. Therefore, Assumption 5 essentially asserts a positive lower bound on the curvature of the left-hand side term, which is mild as it only applies to the neighborhood of the optima $\mu_g^*(x)$ and $\mu_F^*(x)$. It is also weaker than the local strong concavity or global restricted secant inequality (RSI) conditions.

By choosing Algorithm 2 with acceleration as the $\mathsf{MaxMin}$ solver, we provide the convergence analysis of Algorithm 1 next, the proof of which can be found in Appendix D.2.

**Theorem 3.** *Suppose that Assumptions 1–5 and the conditions in Theorem 2 hold. Let $\gamma > \frac{l_{f,1}}{\alpha_g}$, $\epsilon_g \leq \frac{C_{\delta_g}}{2}\delta_g$ and $\epsilon_F \leq \frac{C_{\delta_F}}{2}\delta_F$. If we choose Algorithm 2 with acceleration as the inner loop and input $T_g = \mathcal{O}(\epsilon_g^{-0.5}), T_F = \mathcal{O}(\epsilon_F^{-0.5})$, and $T_y^g = \mathcal{O}(\ln(\epsilon_g^{-1})), T_y^F = \mathcal{O}(\ln(\epsilon_F^{-1}))$ with proper constant stepsizes in Remark 3, then the iterates generated by the Algorithm 1 satisfy* (11):

$$\|(y_{g,t}^{T_g}, \mu_{g,t}^{T_g}) - (y_g^*(x_t), \mu_g^*(x_t))\|^2 = \mathcal{O}(\epsilon_g); \quad \|(y_{F,t}^{T_F}, \mu_{F,t}^{T_F}) - (y_F^*(x_t), \mu_F^*(x_t))\|^2 = \mathcal{O}(\epsilon_F).$$

Theorem 3 concludes the $\mathcal{O}(\epsilon_g^{-0.5})$ complexity for achieving $\epsilon_g$-optimal solutions for the constrained concave-strongly-convex problem $\max_{\mu \in \mathbb{R}_+^{d_x}} \min_{y \in \mathcal{Y}} L(\mu, y)$, and so as for $\epsilon_F$. For solving an $\epsilon$-approximation problem of BLO defined in (4), we solve $\min_{x \in \mathcal{X}} F_\gamma(x)$ with $\gamma = \mathcal{O}(\epsilon^{-0.5})$ according to Theorem 1. In this way, to achieve $\frac{1}{T} \sum_{t=0}^{T-1} \|G_\eta(x_t)\|^2 = \mathcal{O}(\gamma T^{-1} + \gamma^2 \epsilon_F + \gamma^2 \epsilon_g) \leq \epsilon$ by (14), we need $\epsilon_g, \epsilon_F = \mathcal{O}(\epsilon^2)$, i.e. complexity $\tilde{\mathcal{O}}(\epsilon^{-1})$ for the MaxMin solver in Algorithm 2, and $T = \mathcal{O}(\gamma \epsilon^{-1}) = \mathcal{O}(\epsilon^{-1.5})$ for the number of iteration in BLOCC (Algorithm 1). Therefore, the overall complexity is $\tilde{\mathcal{O}}(\epsilon^{-2.5})$, where $\tilde{\mathcal{O}}$ omits the $\ln$ terms.

## 3.3 Special case of the MaxMin Solver: $g^c(x, y)$ being affine in $y$ and $\mathcal{Y} = \mathbb{R}^{d_y}$

In this section, we investigate a special case of BLO with CCs where Assumption 5 automatically holds. Specifically, we focus on the case where $g^c$ is affine in $y$ and $\mathcal{Y} = \mathbb{R}^{d_y}$, i.e.

$$g^c(x, y) = g_1^c(x)^\top y - g_2^c(x). \tag{22}$$

In this case, fixing $x$, taking $L_g(\mu, y; x)$ in (7) and $L_F(\mu, y; x)$ as $L(\mu, y)$, (16) gives

$$D_g(\mu) = \min_{y \in \mathbb{R}^{d_y}} -\langle g_2^c(x), \mu \rangle + \langle y, g_1^c(x)\mu \rangle + g(x, y) \quad \text{and} \tag{23a}$$

$$D_F(\mu) = \min_{y \in \mathbb{R}^{d_y}} -\langle g_2^c(x), \mu \rangle + \langle y, g_1^c(x)\mu \rangle + f(x, y) + \gamma(g(x, y) - v(x)). \tag{23b}$$

When $g_1^c(x)$ is of full column rank, both $D_g(\mu)$ and $D_F(\mu)$ are strongly concave according to Lemma 13 in Appendix so that Assumption 5 holds globally. Moreover, when applying PGD on $-D_g(\mu)$ and $-D_F(\mu)$, strongly convexity guarantees linear convergence (Theorem 3.10 in [9]). Therefore, even without acceleration, Algorithm 2 with $T_y$ sufficiently large performs PGD on $-D(\mu)$ and it converges linearly up to inner loop accuracy. Moreover, PGD on $L(\mu, y)$ in $y$ also converges linearly. This motivates us to implement a single-loop version ($T_y = 1$) of Algorithm 2.

When both $y$ and $\mu$ are unconstrained, the analysis has been established in [19]. However, as $\mu$ is constrained to $\mathbb{R}_+^{d_c}$ in the Lagrangian formulation for inequality constraints, the convergence analysis in [19] is not applicable, and the extension is nontrivial due to the non-differentiability of the projection. We address this technical challenge by treating $\mathbb{R}_+^{d_c}$ as an inequality constraint and reformulating it as an unconstrained problem using Lagrange duality theory. This approach demonstrates that a single-loop $T_y = 1$ update in Algorithm 2, without acceleration, achieves linear convergence for the max-min problem (15). The detailed proof is provided in Appendix D.3.

Thus, by selecting the single-loop version ($T_y = 1$) of Algorithm 2 without acceleration as the MaxMin solver in Algorithm 1, we establish the following theorem with proof available in Appendix D.3.

**Theorem 4** (Inner linear convergence). *Consider BLO with $\mathcal{Y} = \mathbb{R}^{d_y}$ and $g^c(x, y)$ defined in (22). Suppose Assumptions 1–4 and the conditions in Theorem 2 hold. Suppose for any $x \in \mathcal{X}$, there exist constants $s_{\min}$ and $s_{\max}$ such that $0 < s_{\min} \leq \sigma_{\min}(g_1^c(x)) \leq \sigma_{\max}(g_1^c(x)) \leq s_{\max} < \infty$. Let $\gamma > \frac{l_{f,1}}{\alpha_g}$. If we choose Algorithm 2 without acceleration as the inner loop and input $T_g = \mathcal{O}(\ln(\epsilon_g^{-1})), T_F = \mathcal{O}(\ln(\epsilon_F^{-1}))$, and $T_y^g = T_y^F = 1$ with proper constant stepsizes in Remark 4, then the iterates generated by Algorithm 1 satisfy (11):*

$$\|(y_{g,t}^{T_g}, \mu_{g,t}^{T_g}) - (y_g^*(x_t), \mu_g^*(x_t))\|^2 = \mathcal{O}(\epsilon_g); \quad \|(y_{F,t}^{T_F}, \mu_{F,t}^{T_F}) - (y_F^*(x_t), \mu_F^*(x_t))\|^2 = \mathcal{O}(\epsilon_F).$$

Theorem 4 establishes, for the first time, the linear convergence of a strongly convex-concave max-min problem with a constrained maximization parameter. Similar to the previous analysis, we choose $\gamma = \mathcal{O}(\epsilon^{-0.5})$ to solve the equivalent (2a) to $\epsilon$-approximation problem of BLO (4). To achieve $\frac{1}{T} \sum_{t=0}^{T-1} \|G_\eta(x_t)\|^2 \leq \epsilon$, the number of iteration in BLOCC $T = \mathcal{O}(\gamma \epsilon^{-1}) = \mathcal{O}(\epsilon^{-1.5})$ by (14). As the inner MaxMin solver convergences linearly, the overall complexity is $\tilde{\mathcal{O}}(\epsilon^{-1.5})$ where $\tilde{\mathcal{O}}$ omits the $\ln$ terms. We summarized the overall iteration complexity of our BLOCC algorithm in different settings in Table 1.

## 4 Numerical Experiments

This section reports the results of numerical experiments for three different problems: a toy example used to validate our method, an SVM training application, and a network design problem in both

synthetic and real-world transportation scenarios. We provide sensitivity analysis and insights for hyper-parameter choices in Appendix G.1. In the two real-world experiments, we compare the proposed algorithm with two baselines, LV-HBA [75] and GAM [72]. The code is available at https://github.com/Liuyuan999/Penalty_Based_Lagrangian_Bilevel.

## 4.1 Toy example

Consider the BLO problem with an inequality coupled constraint $\mathcal{Y}(x) := \{y \in \mathcal{Y} : y - x \leq 0\}$, given by

$$\min_{x \in [0,3]} \ f(x, y_g^*(x)) = \frac{e^{-y_g^*(x)+2}}{2 + \cos(6x)} + \frac{1}{2}\ln\big((4x-2)^2 + 1\big)$$

$$\text{with} \ \ y_g^*(x) \in \arg\min_{y \in \mathcal{Y}(x)} \ g(x, y) = (y - 2x)^2. \tag{24}$$

Problem (24) satisfies all assumptions for Theorem 2 and Theorem 4. The lower-level problem is strongly convex and $y_g^*(x) = x$. Therefore, the BLO problem with inequality constraint in (24) reduces to $\min_{x \in [0,3]} f(x, y)|_{y=x}$. In Figure 2, we plot the dashed line as the intersected line of the surface $f(x, y)$ and the plane $f(x, y_g^*(x))$, and the red points as the converged points by running BLOCC with $\gamma = 5$ and with 200 different initialization values. It can be seen that BLOCC consistently finds the local minima, verifying the effectiveness.

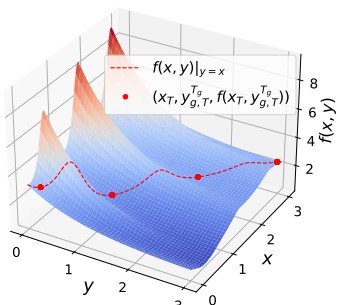

Figure 2: 3-D plot of the upper-level objective $f(x, y)$ of the toy example, with the line $f(x, y)|_{y=x}$ shown in dashed red and the convergence points marked as red dots.

## 4.2 Hyperparameter optimization for SVM

We test the performance of our algorithm BLOCC with $\gamma = 12$ when training a linear SVM model on the diabetes [20] and fourclass datasets [29]. The model is trained via the BLO formulation (1) with inequality CCs; see the details in Appendix E. We compared performance with two baselines, LV-HBA [75] and GAM [72], as they are the only existing algorithms addressing coupled constraints and experimentally evaluated on SVM problems in their respective papers.

| Methods | diabetes | fourclass |
|---|---|---|
| **BLOCC** **(Ours)** | $\mathbf{0.767 \pm 0.039}$ $(\mathbf{1.729 \pm 0.529})$ | $\mathbf{0.761 \pm 0.014}$ $(\mathbf{1.922 \pm 0.108})$ |
| LV-HBA | $0.765 \pm 0.039$ $(2.899 \pm 1.378)$ | $0.748 \pm 0.060$ $(2.404 \pm 0.795)$ |
| GAM | $0.721 \pm 0.047$ $(8.752 \pm 4.736)$ | $0.715 \pm 0.056$ $(13.481 \pm 0.970)$ |

Table 2: Numerical results on the training outcome of our BLOCC in comparison with LV-HBA [75] and GAM [72]. The first row represents accuracy mean $\pm$ standard deviation, and the second row between brackets represents the running time until the upper-level objective's update is smaller than $1e^{-5}$.

Table 2 shows that the model trained by our BLOCC algorithm outperforms that of GAM [72] significantly and is of a similar level as that of LV-HBA [75]. We present some of the performance plots for the diabetes dataset as in Figure 3. Looking into the test accuracy (left), our algorithm achieves more than 0.76 accuracy in the first 2 iterations, which is significantly better than the other ones. For the upper-level objective (middle), in the first few iterations, the loss decreases significantly under all algorithms, in which our BLOCC achieves the lowest results. Moreover, in Figure 3 (right), the lower-level optimum for LV-HBA was not attained until the very end. This indicates that the decrease of upper loss between 0-40 iterations in the middle figure may be due to the suboptimality of lower-level variables. For our BLOCC and GAM, the lower-level minimum is attained as lower-level objective $g \geq 0$ in this case.

## 4.3 Transportation network design problem

BLO is particularly relevant in transportation, where network planning must consider different time horizons and actors. Those problems are large-scale with a large number of upper- and lower-level variables and CCs, challenging the use of traditional BLO techniques, which involve expensive calculations of second order information. As a result, our final experiment considers a network design problem, where we act as an operator whose profit is modeled as the upper-level objective that is determined by the passengers' behavior, modeled in the lower-level. We considered three networks:

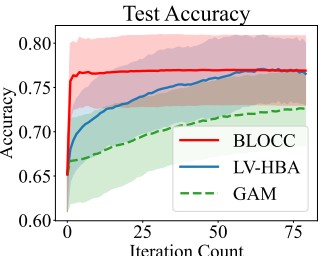 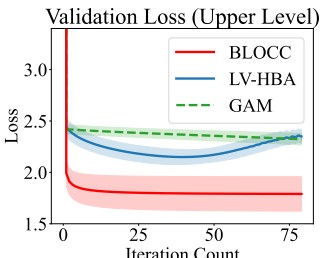 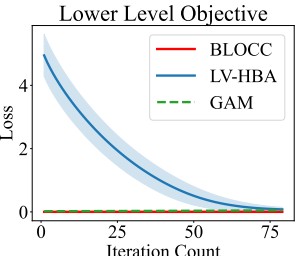

Figure 3: Test accuracy (left), upper loss $f(x, y)$ (middle), and lower loss $g(x, y)$ (right) for the SVM on the diabetes dataset. The experiments are executed for 50 different random train-validation-test splits, with the bold line representing the mean, and the shaded regions being the standard deviation.

| Methods | NN = 3, NV = 48 NC = 24, NZ = 126 Runtime (s) | UL utility | NN = 9, NV = 2,262 NC = 678, NZ = 6,222 Runtime (s) | UL utility | NN = 26, NV = 49,216 NC = 13,336, NZ = 129,256 Runtime (s) | UL utility |
|---|---|---|---|---|---|---|
| LV-HBA | $3.51e2$ | 1.53 | / | / | / | / |
| BLOCC (Ours)-$\gamma = 2$ | $2.02e1$ | 1.07 | $7.27e2$ | 8.09 | $6.82e4$ | 98.40 |
| BLOCC (Ours)-$\gamma = 3$ | $2.00e1$ | 1.69 | $8.50e2$ | 10.37 | $6.42e4$ | 111.39 |
| BLOCC (Ours)-$\gamma = 4$ | $2.01e1$ | 1.71 | $8.68e2$ | 11.04 | $6.70e4$ | 138.78 |

Table 3: Results of the transportation experiment, both in terms of running time (Runtime) and convergenced upper-level objective value (UL utility, larger there better), with stepsize $\eta = 1.6e-4$. We use "/" for algorithms that cannot converge within 24 hours of execution. NN (Number of Nodes) is the number of stations in the network. Analogously, NV (Number of Variables), NC (Number of Constraints), and NNZ (Number of non-zero elements) are the number of optimization variables, constraints, and non-zero elements of the constraints matrix, respectively.

two synthetic networks of 3 and 9 nodes, respectively, and a real-world network of 26 nodes in the city of Seville, Spain. Further details about the formulation and the experiment can be found in Appendix F.

In this experiment, we only compare our BLOCC with LV-HBA [75] as the sole baseline, which is the only existing algorithm that addresses both coupled inequality $g^c(x, y) \leq 0$ and domain constraints $\mathcal{Y}$. GAM [72] and BVFSM [46] cannot handle lower-level domain constraints as they rely on the hypergradient of lower-level and require LL stationarity in an unconstrained space. From Table 3, we can see that LV-HBA failed to work efficiently, especially for large networks. This is mainly because the increased constraints render the projection step impracticable. Our BLOCC, in contrast, is much faster, and it successfully converges even with large real-world networks. We provided computational complexity analysis in Appendix H and we can conclude that our BLOCC is robust to large-scale problems.

## 5   Conclusions and Future Work

This paper proposed a novel primal-dual-assisted penalty reformulation for BLO problems with coupled lower-level constraints, and developed a new first-order method BLOCC to solve the resultant problem. The non-asymptotic convergence rate of our algorithm is $\tilde{\mathcal{O}}(\epsilon^{-2.5})$, tightening to $\tilde{\mathcal{O}}(\epsilon^{-1.5})$ when the lower-level constraints are affine in $y$ without any other constraints. Our method achieves the best-known convergence rate and is projection-free, making it more favorable for large-scale, high-dimensional, constrained BLO problems. Experiments on SVM model training and transportation network planning showcased the effectiveness of our algorithm.

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

# Supplementary Material for "A Primal-Dual-Assisted Penalty Approach to Bilevel Optimization with Coupled Constraints"

## Table of Contents

## A  Preliminaries

This section will provide some preliminaries for our subsequent analysis.

**Definition 5.** *For a convex function $h : \mathbb{R}^{d_q} \to \mathbb{R}$ whose domain is $\mathcal{Q} \subseteq \mathbb{R}^{d_q}$, the Legendre conjugate of $h^* : \mathcal{Q}^* \to \mathbb{R}$ is defined as:*

$$h^*(q) := \sup_{q' \in \mathcal{Q}} \{\langle q', q \rangle - h(q')\} = -\inf_{q' \in \mathcal{Q}} \{-\langle q', q \rangle + h(q')\},$$

$$\forall q \in \mathcal{Q}^* := \{q \in \mathbb{R}^{d_q} : \sup_{q' \in \mathcal{Q}} \{\langle q', q \rangle - h(q')\} < \infty\}.$$

**Remark 1.** *When $h$ is strongly convex in $\mathbb{R}^{d_q}$, it is lower bounded and therefore $\mathcal{Q}^* = \mathbb{R}^{d_y}$.*

**Lemma 4.** *Suppose $h : \mathbb{R}^{d_y} \to \mathbb{R}$ is $l_{h,1}$-smooth and $\alpha_h$-strongly convex and its domain $\mathcal{Q} \subseteq \mathbb{R}^{d_q}$ is convex, closed and non-empty.*

1. *If $\mathcal{Q} = \mathbb{R}^{d_q}$, the gradient mappings $\nabla h$ and $\nabla h^*$ are inverse of each other ([58]); and $h^* : \mathbb{R}^{d_q} \to \mathbb{R}$ is $\frac{1}{\alpha_h}$-smooth and $\frac{1}{l_{h,1}}$-strongly convex (Proposition 2.6 [3]).*

2. If $\mathcal{Q} \subset \mathbb{R}^{d_q}$, $h^*$ is $\frac{1}{\alpha_h}$-smooth ([36]) and and convex (Theorem 4.43 [32]).

**Lemma 5.** *Suppose $\mathcal{Q} \subseteq \mathbb{R}^{d_q}$ is convex, closed and non-empty, $h : \mathbb{R}^{d_q} \to \mathbb{R}$ is strongly convex on $\mathcal{Q}$, $h^c : \mathbb{R}^{d_q} \to \mathbb{R}^{d_c}$ is convex on $\mathcal{Q}$ and $d_c$ is finite, and $\{q \in \mathcal{Q} : h^c(q) \leq 0\}$ is non-empty.*

1. *The problem $\min_{q \in \{q \in \mathcal{Q}: h^c(q) \leq 0\}} h(q)$ has a unique feasible solution.*

2. *When linear independence constraint qualification (LICQ) condition holds for $h^c(q)$, the Lagrange multiplier for the problem $\min_{q \in \{q \in \mathcal{Q}: h^c(q) \leq 0\}} h(q)$, i.e. solution to the problem $\max_{\mu \in \mathbb{R}_+^{d_c}} \min_{q \in \mathcal{Q}} h(q) + \langle \mu, h^c(q) \rangle$ is unique [69].*

**Lemma 6** (Lemma 3.1 in [9]). *Suppose $\mathcal{Q} \subseteq \mathbb{R}^{d_q}$ is convex, closed, and nonempty. For any $q_1 \in \mathbb{R}^{d_q}$ and any $q_2 \in \mathcal{Q}$, it follows that*

$$\langle \mathrm{Proj}_{\mathcal{Q}}(q_1) - q_2, \mathrm{Proj}_{\mathcal{Q}}(q_1) - q_1 \rangle \leq 0. \tag{25}$$

*In this way, take $q_1 = q_3 - \eta g$ for any $q_3 \in \mathcal{Q}$, and denote $q_3^{\eta, g} = \mathrm{Proj}_{\mathcal{Q}}(q_3 - \eta g)$ as a projected gradient update in direction $g$ with stepsize $\eta$, we have,*

$$\langle g, q_3^{\eta, g} - q_2 \rangle \leq -\frac{1}{\eta} \langle q_3^{\eta, g} - q_2, q_3^{\eta, g} - q_3 \rangle, \quad \forall q_2, q_3 \in \mathcal{Q}. \tag{26}$$

**Lemma 7** (Theorem 3.10 [9]). *Suppose a differentiable function $h$ is $l_{h,1}$-smooth and $\alpha_{h_2}$-strongly convex. Consider the constrained problem $\min_{q \in \mathcal{Q}} h(q)$ where $\mathcal{Q}$ is non-empty, closed and convex. Projected Gradient Descent with $\eta \leq \frac{1}{l_{h,1}}$ converges linearly to the unique $q^* = \arg\min_{q \in \mathcal{Q}} h(q)$:*

$$\| \mathrm{Proj}_{\mathcal{Q}}(q - \eta \nabla h(q)) - q^* \| \leq (1 - \alpha\eta)^{1/2} \|q - q^*\| \leq (1 - \alpha\eta/2) \|q - q^*\|, \quad \forall q \in \mathcal{Q}. \tag{27}$$

# B  Analysis of the Penalty-Based Lagrangian Reformulation

## B.1  Proof of Lemma 1

According to Lemma 5, for any fixed $x$, there exists a unique $\mu_g^*(x)$. Therefore, according to Lagrange duality theorem, for any fixed $x$, the primal problem

$$\min_{y \in \mathcal{Y}} g(x, y) \quad \text{s.t.} \quad g^c(x, y) \leq 0 \quad \Leftrightarrow \quad \min_{y \in \mathcal{Y}} g(x, y) + \langle \mu_g^*(x), g^c(x, y) \rangle.$$

As $g(x, y)$ is $\alpha_g$-strongly convex in $y$ and $g^c(x, y)$ is convex in $y$, we know $g(x, y) + \langle \mu_g^*(x), g^c(x, y) \rangle$ is $\alpha_g$-strongly convex in $y$, where the modulus $\alpha_g$ is independent of $x$. Therefore, according to Appendix F and G in [37], and Theorem 3.3 in [18], the quadratic growth in statement 1 can be concluded.

As $g(x, y)$ is strongly convex in $y$, and $\mathcal{Y}(x)$ is a non-empty, closed and convex set under assumption 3, there exists a unique solution $y_g^*(x)$ such that $g(x, y_g^*(x)) = v(x)$ by Lemma 5. In this way, if $y \neq y_g^*(x)$ and $y \in \mathcal{Y}(x)$, we have $g(x, y) > v(x)$. This completes the proof of statement 2.

## B.2  Proof of Theorem 1

We know from Lemma 1 that $g(x, y) - v(x) \geq \frac{\alpha_g}{2} \|y - y_g^*(x)\|^2$ and $g(x, y) = v(x)$ if and only if $y = y_g^*(x)$. This is squared-distance bound follows Definition 1 in [62]. Under Lipschitzness of $f(x, y)$ with respect to $y$, finding the solutions to the $\epsilon$-approximate problem in (4) is equivalent to finding the solutions to its penalty reformulation

$$\min_{(x,y) \in \{\mathcal{X} \times \mathcal{Y}: g^c(x,y) \leq 0\}} f(x, y) + \gamma(g(x, y) - v(x)) \tag{28}$$

with $\gamma = \mathcal{O}(\epsilon^{-0.5})$ following Theorems 1 and 2 in [62].

Moreover, jointly finding solution for $(x, y)$ in (28) is in equivalence to finding solutions in

$$\min_{x \in \mathcal{X}} \min_{y \in \mathcal{Y}(x)} f(x, y) + \gamma(g(x, y) - v(x)). \tag{29}$$

The proof of this equivalence is as follows. Suppose $(x_0, y_0) \in \{\mathcal{X} \times \mathcal{Y} : g^c(x, y) \leq 0\}$ being a solution to (28). Suppose for any $x \in \mathcal{X}$, $y_F^*(x) \in \arg\min_{y \in \mathcal{Y}(x)} f(x, y) + \gamma(g(x, y) - v(x))$. We know that for any $x \in \mathcal{X}$, $y \in \mathcal{Y}(x)$, it follows that

$$
\begin{aligned}
f(x_0, y_0) + \gamma(g(x_0, y_0) - v(x_0)) &\leq f(x, y_F^*(x)) + \gamma(g(x, y_F^*(x)) - v(x)) \\
&\leq f(x, y) + \gamma(g(x, y) - v(x)).
\end{aligned}
$$

This means any solution to (28) is a solution to (29). On the other hand, suppose $x_0 \in \mathcal{X}$, $y_F^*(x_0) \in \mathcal{Y}(x_0)$ is a solution to (29). We know that for any $(x, y) \in \{\mathcal{X} \times \mathcal{Y} : g^c(x, y) \leq 0\}$,

$$
\begin{aligned}
f(x_0, y_F^*(x_0)) + \gamma(g(x_0, y_F^*(x_0)) - v(x_0)) &\leq f(x, y_F^*(x)) + \gamma(g(x, y_F^*(x)) - v(x)) \\
&\leq f(x, y) + \gamma(g(x, y) - v(x)).
\end{aligned}
$$

This means any solution to (29) is a solution to (28).

Besides, we know $f(x, y)$ is $l_{f,1}$-smooth, $g(x, y)$ is $\alpha_g$-strongly convex in $y$. By the definitions of strongly convexity and smoothness, we know for fixed $x$, for any $y_1, y_2 \in \mathcal{Y}$,

$$
\begin{aligned}
&f(x, y_1) + \gamma(g(x, y_1) - v(x)) - f(x, y_2) + \gamma(g(x, y_2) - v(x)) \\
={}& f(x, y_1) - f(x, y_2) + \gamma(g(x, y_1) - g(x, y_2)) \\
\geq{}& \langle \nabla_y f(x, y_2), y_1 - y_2 \rangle - \frac{l_{f,1}}{2}\|y_1 - y_2\|^2 + \gamma \langle \nabla_y g(x, y_2), y_1 - y_2 \rangle + \gamma \frac{\alpha_g}{2}\|y_1 - y_2\|^2 \\
={}& \langle \nabla_y f(x, y_2) + \gamma \nabla_y g(x, y_2), y_1 - y_2 \rangle + \frac{\gamma \alpha_g - l_{f,1}}{2}\|y_1 - y_2\|^2.
\end{aligned}
\tag{30}
$$

This proves that $f(x, y) + \gamma(g(x, y) - v(x))$ is $(\gamma \alpha_g - l_{f,1})$-strongly convex in $y$. Moreover, according to Assumption 2, the constraint $g^c(x, y)$ is convex in $y$, and $\min_{y \in \mathcal{Y}(x)} f(x, y) + \gamma(g(x, y) - v(x))$ is equivalent to its equivalent *Lagrangian Dual Form* [59]

$$
\max_{\mu \in \mathbb{R}_+^{d_c}} \min_{y \in \mathcal{Y}} f(x, y) + \gamma(g(x, y) - v(x)) + \langle \mu, g^c(x, y) \rangle.
\tag{31}
$$

Therefore, (28) can be recovered to (2a) and this completes the proof.

## C  Analysis of the Differentiability of Value Functions

**Lemma 8** (Theorem 2.16 in [32]). *Suppose $h(x, y)$ is strongly convex in $y \in \mathcal{Y}$ and is Lipschitz with respect to $x \in \mathcal{X}$, $h^c(x, y)$ is convex in $y$ and is Lipschitz with respect to $x$, and both $\mathcal{Y}$ and $\{y \in \mathcal{Y} : h^c(x, y) \leq 0\}$ are non-empty, closed, and convex. For the problem $\min_{y \in \{y \in \mathcal{Y} : h^c(x, y) \leq 0\}} h(x, y)$, the unique solution $y_h^*(x)$ and unique Lagrange multiplier $\mu_h^*(x)$, defined as*

$$
(y_h^*(x), \mu_h^*(x)) := \arg \max_{\mu \in \mathbb{R}_+^{d_x}} \min_{y \in \mathcal{Y}} h(x, y) + \langle \mu, h^c(x, y) \rangle,
\tag{32}
$$

*is Lipschitz in $x$. In other words, there exist $L_h \geq 0$ that, for all $x_1, x_2 \in \mathcal{X}$,*

$$
\|(y_h^*(x_1); \mu_h^*(x_1)) - (y_h^*(x_2); \mu_h^*(x_2))\| \leq L_h \|x_1 - x_2\|.
$$

Before proving Lemmas 2 and 3, we would like to introduce a more general form.

**Lemma 9.** *Suppose $\mathcal{Y}$ and $\{y \in \mathcal{Y} : h^c(x, y) \leq 0\}$ are both non-empty, closed and convex, $h(x, y)$ is jointly smooth in $(x, y)$ and is strongly convex in $y$, $h^c(x, y)$ is convex in $y$, and both $h(x, y)$ and $h^c(x, y)$ are Lipschitz with respect to $x$. Then we have*

$$
v_h(x) = \min_{y \in \mathcal{Y}} h(x, y) \quad s.t. \quad h^c(x, y) \leq 0
$$

*is differentiable with its gradient as*

$$
\nabla v_h(x) = \nabla_x h(x, y_h^*(x)) + \langle \mu_h^*(x), h^c(x, y_h^*(x)) \rangle,
\tag{33}
$$

*where $(y_h^*(x), \mu_h^*(x))$ defined in (32) are unique.*

*Proof.* We prove this using Theorem 4.24 in [7].

i) As $h(x, y)$ being strongly convex in $y$, Condition 1 in Theorem 4.24 in [7] is satisfied and the solution sets are of singleton value $(y_h^*(x), \mu_h^*(x))$ according to Lemma 5.

ii) Moreover, the smoothness of $h(x, y)$ guarantees Robinson's constraint qualification [2], which implies the directional regularity condition (Definition 4.8 in [7]) for any direction $d$ (Theorem 4.9. (ii) in [7]). This guarantees Condition 2 in Theorem 4.24 in [7] can be satisfied for all directions $d$.

iii) Additionally, under the Lipschitzness of $h(x, y)$ and $h^c(x, y)$ with respect to $x$, $y_h^*(x), \mu_h^*(x)$ are Lipschitz according to Lemma 8. This implies condition 3 in Theorem 4.24 in [7] holds.

In this way, all conditions in Theorem 4.24 in [7] hold and it gives the gradient as in (33) for unique $(y_h^*(x), \mu_h^*(x))$. This completes the proof. $\qquad\square$

## C.1 Proof of Lemma 2

*Proof.* The problem $\min_{y \in \mathcal{Y}} g(x, y)$ s.t. $g^c(x, y) \leq 0$ fits in the setting of Lemma 9 by taking $h(x, y) = g(x, y)$ and $h^c(x, y) = g^c(x, y)$. Therefore the derivative (8) can be obtained accordingly. Moreover, for any $x_1, x_2 \in \mathcal{X}$,

$$
\begin{aligned}
&\|\nabla v(x_1) - \nabla v(x_2)\| \\
=&\|\nabla_x g(x_1, y_g^*(x_1)) + \langle \mu_g^*(x_1), \nabla_x g^c(x_1, y_g^*(x_1)) \rangle - \nabla_x g(x_2, y_g^*(x_2)) \\
&\quad - \langle \mu_g^*(x_2), \nabla_x g^c(x_2, y_g^*(x_2)) \rangle \| \\
\overset{(a)}{\leq}&\|\nabla_x g(x_1, y_g^*(x_1)) - \nabla_x g(x_2, y_g^*(x_2))\| \\
&\quad + \|\langle \mu_g^*(x_1), \nabla_x g^c(x_1, y_g^*(x_1)) \rangle - \langle \mu_g^*(x_1), \nabla_x g^c(x_2, y_g^*(x_2)) \rangle \| \\
&\quad + \|\langle \mu_g^*(x_1), \nabla_x g^c(x_2, y_g^*(x_2)) \rangle - \langle \mu_g^*(x_2), \nabla_x g^c(x_2, y_g^*(x_2)) \rangle \| \\
\overset{(b)}{\leq}&(l_{g,1} + B_g l_{g^c,1})(\|x_1 - x_2\| + \|y_g^*(x_1) - y_g^*(x_2)\|) + l_{g^c,0}\|\mu_g^*(x_1) - \mu_g^*(x_2)\| \\
\overset{(c)}{\leq}&((l_{g,1} + B_g l_{g^c,1})(1 + L_g) + l_{g^c,0}L_g)\|x_1 - x_2\|,
\end{aligned}
$$

where $(a)$ follows triangle inequality; $(b)$ leverage on the Lipschitzness of $\nabla g$, $g^c$ and $\nabla g^c$ in $x$, and the upper bound for $\|\mu_g^*(x)\|$; and $(c)$ uses the Lipschitzness of $y_g^*(x)$ and $\mu_g^*(x)$ from Lemma 8. As the bound is loose due to the use of triangle inequality, we can conclude that $v(x)$ is $l_{v,1}$-smooth where $l_{v,1} \leq ((1 + B_g)(1 + L_g)l_{g^c,1} + l_{g^c,0}L_g)$. $\qquad\square$

## C.2 Proof of Lemma 3

By assumption, $f(x, y)$ is $l_{f,1}$-smooth and $g(x, y)$ is $\alpha_g$-strongly convex in $y$. We know $f(x, y) + \gamma(g(x, y) - v(x))$ is $(\gamma \alpha_g - l_{f,1})$-strongly convex when $\gamma > \frac{l_{f,1}}{\alpha_g}$ as discussed in (30). Moreover, constraint $g^c(x, y)$ is convex in $y$ by Assumption 2. In this way, the problem

$$
\min_{y \in \mathcal{Y}} f(x, y) + \gamma(g(x, y) - v(x)) \quad \text{s.t. } g^c(x, y) \leq 0
$$

features strong convexity according to Chapter 4 in [59] and it equals to

$$
F_\gamma(x) = \max_{\mu \in \mathbb{R}_+^{d_c}} \min_{y \in \mathcal{Y}} f(x, y) + \gamma(g(x, y) - v(x)) + \langle \mu, g^c(x, y) \rangle.
$$

Considering the smoothness of $v(x)$ as presented in Lemma 2, all assumptions in Lemma 9 are satisfied. Therefore the derivative (9) can be obtained. In addition, for any $x_1, x_2 \in \mathcal{X}$,

$$
\begin{aligned}
&\|\nabla F(x_1) - \nabla F(x_2)\| \\
=&\|\nabla_x f(x_1, y_F^*(x_1)) + \gamma(\nabla_x g(x_1, y_F^*(x_1)) - \nabla v(x_1)) + \langle \mu_F^*(x_1), \nabla_x g^c(x_1, y_F^*(x_1)) \rangle \\
&\quad - \nabla_x f(x_2, y_F^*(x_2)) - \gamma(\nabla_x g(x_2, y_F^*(x_2)) - \nabla v(x_2)) - \langle \mu_F^*(x_2), \nabla_x g^c(x_2, y_F^*(x_2)) \rangle \| \\
\overset{(a)}{\leq}&\|\nabla_x f(x_1, y_F^*(x_1)) - \nabla_x f(x_2, y_F^*(x_2))\| + \gamma \|\nabla_x g(x_1, y_F^*(x_1)) - \nabla_x g(x_2, y_F^*(x_2))\| \\
&\quad + \gamma \|\nabla v(x_1) - \nabla v(x_2)\| + \|\langle \mu_F^*(x_1), \nabla_x g^c(x_1, y_F^*(x_1)) \rangle - \langle \mu_F^*(x_1), \nabla_x g^c(x_2, y_F^*(x_2)) \rangle \|
\end{aligned}
$$

$$+ \|\langle \mu_F^*(x_1), \nabla_x g^c(x_2, y_F^*(x_2))\rangle - \langle \mu_F^*(x_2), \nabla_x g^c(x_2, y_F^*(x_2))\rangle\|$$

$$\overset{(b)}{\leq} (l_{f,1} + \gamma l_{g,1} + B_F l_{g^c,1})(\|x_1 - x_2\| + \|y_F^*(x_1) - y_F^*(x_2)\|) + \gamma l_{v,1}\|x_1 - x_2\|$$
$$+ l_{g^c,0}\|\mu_F^*(x_1) - \mu_F^*(x_2)\|$$

$$\overset{(c)}{\leq} ((l_{f,1} + \gamma l_{g,1} + B_F l_{g^c,1})(1 + L_F) + \gamma l_{v,1} + l_{f^c,0} L_F)\|x_1 - x_2\|,$$

where $(a)$ follows triangle inequality; $(b)$ leverage on the Lipschitzness of $\nabla f$, $\nabla g$, $g^c$ and $\nabla g^c$ in $x$, and the upper bound for $\|\mu_F^*(x)\|$; and $(c)$ uses the Lipschitzness of $y_F^*(x)$ and $\mu_F^*(x)$ from Lemma 8. As the bound is loose due to the use of triangle equality, we can conclude that $F(x)$ is $l_{F,1}$-smooth where $l_{F,1} \leq (l_{f,1} + \gamma l_{g,1} + B_F l_{g^c,1})(1 + L_F) + \gamma l_{v,1} + l_{f^c,0} L_F$.

# D Convergence Analysis of the Main Result

## D.1 Proof of Theorem 2

Define the bias term $b(x_t)$ of the gradient $\nabla F_\gamma(x_t)$ as

$$b(x_t) := \nabla F_\gamma(x_t) - g_{F,t}$$

$$= \Big( \nabla_x f(x_t, y_F^*(x_t)) + \gamma\big(\nabla_x g(x, y_F^*(x_t)) - (\nabla_x g(x_t, y_g^*(x_t)) + \langle \mu_g^*(x_t), \nabla_x g^c(x_t, y_g^*(x_t))\rangle)\big)$$

$$+ \langle \mu_F^*(x_t), \nabla_x g^c(x_t, y_F^*(x_t))\rangle \Big)$$

$$- \Big( \nabla_x f(x_t, y_{F,t}^{T_F}) + \gamma\big(\nabla_x g(x_t, y_{F,t}^{T_F}) - (\nabla_x g(x_t, y_{g,t}^{T_g}) + \langle \mu_{g,t}^{T_g}, \nabla_x g^c(x_t, y_{g,t}^{T_g})\rangle)\big)$$

$$+ \langle \mu_F^{T_F}, \nabla_x g^c(x_t, y_{F,t}^{T_F})\rangle \Big).$$

In this way, we have

$$\|b(x_t)\| \overset{(a)}{\leq} \|\nabla_x f(x_t, y_{F,t}^{T_F}) - \nabla_x f(x_t, y_F^*(x_t))\|$$

$$+ \gamma\Big( \|\nabla_x g(x_t, y_{F,t}^{T_F}) - \nabla_x g(x_t, y_F^*(x_t))\| + \|\nabla_x g(x_t, y_{g,t}^{T_g}) - \nabla_x g(x_t, y_g^*(x_t))\|$$

$$+ \Big\| \langle \mu_g^*(x_t), \nabla_x g^c(x_t, y_g^*(x_t))\rangle - \langle \mu_g^*(x_t), \nabla_x g^c(x_t, y_{g,t}^{T_g})\rangle \Big\|$$

$$+ \Big\| \langle \mu_g^*(x_t), \nabla_x g^c(x_t, y_{g,t}^{T_g})\rangle - \langle \mu_{g,t}^{T_g}, \nabla_x g^c(x_t, y_{g,t}^{T_g})\rangle \Big\| \Big)$$

$$+ \|\langle \mu_F^{T_F}, \nabla_x g^c(x_t, y_{F,t}^{T_F})\rangle - \langle \mu_F^*(x_t), \nabla_x g^c(x_t, y_{F,t}^{T_F})\rangle\|$$

$$+ \|\langle \mu_F^*(x_t), \nabla_x g^c(x_t, y_{F,t}^{T_F})\rangle - \langle \mu_F^*(x_t), \nabla_x g^c(x_t, y_F^*(x_t))\rangle\|$$

$$\overset{(b)}{\leq} l_{f,1}\|y_{F,t}^{T_F} - y_F^*(x_t)\| + \gamma\Big( l_{g,1}\|y_{F,t}^{T_F} - y_F^*(x_t)\| + l_{g,1}\|y_{g,t}^{T_g} - y_g^*(x_t)\|$$

$$+ l_{g^c,0}\|\mu_{g,t}^{T_g} - \mu_g^*(x_t)\| + B_g l_{g^c,1}\|y_{g,t}^{T_g} - y_g^*(x_t)\| \Big)$$

$$+ l_{g^c,0}\|\mu_{F,t}^{T_F} - \mu_F^*(x_t)\| + B_F l_{g^c,1}\|y_{F,t}^{T_F} - y_F^*(x_t)\|$$

$$\overset{(c)}{=} (l_{f,1} + \gamma l_{g,1} + B_F l_{g^c,0})\|y_{F,t}^{T_F} - y_F^*(x_t)\| + l_{g^c,0}\|\mu_{F,t}^{T_F} - \mu_F^*(x_t)\|$$

$$+ \gamma\Big( (l_{g,1} + B_g l_{g^c,1})\|y_{g,t}^{T_g} - y_g^*(x_t)\| + l_{g^c,0}\|\mu_{g,t}^{T_g} - \mu_g^*(x_t)\| \Big),$$

where $(a)$ uses triangle inequality, $(b)$ relies on the Lipschitzness of $\nabla f$, $\nabla g$, $g^c$, and $\nabla g^c$ in $x$, the upper bounds for $\|\mu_F^*(x)\|$ and $\|\mu_g^*(x)\|$, and Cauchy-Schwartz inequality, and $(c)$ is by rearrangement.

Furthermore, according to Young's inequality, it follows that

$$\|b(x_t)\|^2 \leq 2\left((l_{f,1} + \gamma l_{g,1} + B_F l_{g^c,0})\|y_{F,t}^{T_F} - y_{F,t}^*\| + l_{g^c,0}\|\mu_{F,t}^{T_F} - \mu_{F,t}^*\|\right)^2$$
$$+ 2\gamma^2\left((l_{g,1} + B_g l_{g^c,1})\|y_{g,t}^{T_g} - y_g^*(x_t)\| + l_{g^c,0}\|\mu_{g,t}^{T_g} - \mu_g^*(x_t)\|\right)^2$$
$$= \mathcal{O}(\gamma^2 \epsilon_F + \gamma^2 \epsilon_g).$$

According to Lemma 3, $F_\gamma(x)$ is $l_{F,1}$-smooth in $\mathcal{X}$. In this way, by the smoothness, we have

$$F(x_{t+1}) \leq F(x_t) + \langle \nabla F(x_t), x_{t+1} - x_t \rangle + \frac{l_{F,1}}{2}\|x_{t+1} - x_t\|^2$$
$$\leq F(x_t) + \langle g_{F,t}, x_{t+1} - x_t \rangle + \frac{1}{2\eta}\|x_{t+1} - x_t\|^2 + \langle b(x_t), x_{t+1} - x_t \rangle, \qquad (34)$$

where the second inequality is by $\eta \leq \frac{1}{l_{F,1}}$ and $\nabla F(x_t) = g_{F_t} + b(x_t)$.

The projection guarantees that $x_{t+1}$ and $x_t$ are in $\mathcal{X}$. Following Lemma 6, we know that

$$\langle g_{F,t}, x_{t+1} - x_t \rangle \leq -\frac{1}{\eta}\|x_{t+1} - x_t\|^2.$$

Plugging this back to (34), it follows

$$F(x_{t+1}) \leq F(x_t) - \frac{1}{2\eta}\|x_{t+1} - x_t\|^2 + \langle b(x_t), x_{t+1} - x_t \rangle$$
$$\leq F(x_t) - \frac{1}{2\eta}\|x_{t+1} - x_t\|^2 + \eta\|b(x_t)\|^2 + \frac{1}{4\eta}\|x_{t+1} - x_t\|^2$$
$$= F(x_t) - \frac{1}{4\eta}\|x_{t+1} - x_t\|^2 + \eta\|b(x_t)\|^2,$$

where the second inequality is from Young's inequality. Telescoping therefore gives

$$\frac{1}{T}\sum_{t=0}^{T-1}\|G_\eta(x_t)\|^2 \leq \frac{4}{\eta T}(F(x_0) - F(x_T)) + \frac{4}{T}\sum_{t=0}^{T-1}\|b(x_t)\|^2$$
$$= \mathcal{O}(\eta^{-1}T^{-1}) + O(\gamma^2 \epsilon_F + \gamma^2 \epsilon_g)$$
$$= \mathcal{O}(\gamma T^{-1} + \gamma^2 \epsilon_F + \gamma^2 \epsilon_g)$$

where last equality comes from $\eta = \mathcal{O}(\gamma^{-1})$ as $\eta \leq \frac{1}{l_{F,1}}$ and $l_{F,1} \leq (l_{f,1} + \gamma l_{g,1} + B_F l_{g^c,1})(1 + L_F) + \gamma l_{v,1} + l_{f^c,0}L_F = \mathcal{O}(\gamma)$. This completes the proof.

### D.2  Proof of Theorem 3

To restate, we are viewing $L_g(\mu, y; x)$ in (7) and $L_F(\mu, y; x)$ in (5) respectively as $L(\mu, y)$ and considering the following max-min problem

$$\max_{\mu \in \mathbb{R}_+^{d_c}} \min_{y \in \mathcal{Y}} L(\mu, y).$$

In (16), we defined the minimization part as

$$D(\mu) := \min_{y \in \mathcal{Y}} L(\mu, y) = L(\mu, y_\mu^*(\mu)) \quad \text{where} \quad y_\mu^*(\mu) := \arg\min_{y \in \mathcal{Y}} L(\mu, y).$$

To evaluate $D(\mu)$ for $L_g(\mu, y; x)$ in (7) and $L_F(\mu, y; x)$ in (5) as $L(\mu, y)$, we define the following mappings for fixed $x \in \mathcal{X}$:

$$y_{\mu,g}^*(\mu) := \arg\min_{y \in \mathcal{Y}} L_g(\mu, y; x), \qquad (35)$$
$$y_{\mu,F}^*(\mu) := \arg\min_{y \in \mathcal{Y}} L_F(\mu, y; x). \qquad (36)$$

In this way, for $L_g(\mu, y; x)$ in (7) and $L_F(\mu, y; x)$ in (5) respectively as $L(\mu, y)$, $D(\mu)$ defined in (16) equals to $D_g(\mu)$ and $D_F(\mu)$ respectively, where

$$D_g(\mu) = \min_{y \in \mathcal{Y}} L_g(\mu, y; x) = L_g(\mu, y^*_{\mu, g}(\mu); x), \tag{37}$$

$$D_F(\mu) = \min_{y \in \mathcal{Y}} L_F(\mu, y; x) = L_F(\mu, y^*_{\mu, F}(\mu); x). \tag{38}$$

In the following lemma, we show that $D(\mu)$ exhibits concavity and smoothness, which are favorable properties for conducting gradient-based algorithm.

**Lemma 10** (Smoothness and Concavity of $D(\mu)$). *Suppose all the assumptions in Theorem 3 hold. For fixed $x \in \mathcal{X}$, the following holds*

1. *$y^*_{\mu, g}(\mu)$ in (35) and $y^*_{\mu, F}(\mu)$ in (36) are respectively $\frac{1}{\alpha_g}$ and $\frac{1}{\gamma \alpha_g - l_{f,1}}$-Lipschitz to $\mu$.*

2. *$D_g(\mu)$ in (37) is concave and $\frac{l_{g^c,0}}{\alpha_g}$-smooth, and $D_F(\mu)$ in (38) is concave and $\frac{l_{g^c,0}}{\gamma \alpha_g - l_{f,1}}$-smooth.*

*Proof.* To restate,

$$L_g(\mu, y; x) = g(x, y) + \langle \mu, g^c(x, y) \rangle,$$
$$L_F(\mu, y; x) = f(x, y) + \gamma(g(x, y) - v(x)) + \langle \mu, g^c(x, y) \rangle.$$

Under Assumption 1 and 2, for fixed $x$ and given $\mu$, $L_g(\mu, y; x)$ is $\alpha_g$-strongly convex and $(l_{g,1} + \|\mu\| l_{g^c,1})$-smooth in $y$, and $L_F(\mu, y; x)$ is $(\gamma \alpha_g - l_{f,1})$-strongly convex and $(l_{f,1} + \gamma l_{g,1} + \|\mu\| l_{g^c,1})$-smooth in $y$. Therefore, we know $y^*_{\mu, g}(\mu)$ in (35) and $y^*_{\mu, F}(\mu)$ in (36) are respectively $\frac{1}{\alpha_g}$ and $\frac{1}{\gamma \alpha_g - l_{f,1}}$-Lipschitz to $\mu$ by directly quoting Theorem F.10 in [17] or Theorem 4.47 in [32]. This proves the first part of the Lemma.

For the second part, the concavity of $D_g(\mu)$ and $D_F(\mu)$ can be directly obtained by Lemma 2.58 in [59] as $L_g(\mu, y; x)$ and $L_g(\mu, y; x)$ are both convex in $y$.

Moreover, following Theorem 4.24 in [7], we have

$$\nabla D_g(\mu) = \nabla_\mu L_g(\mu, y^*_{\mu, g}(\mu)) = g^c(x, y^*_{\mu, g}(\mu)),$$
$$\nabla D_F(\mu) = \nabla_\mu L_F(\mu, y^*_{\mu, F}(\mu)) = g^c(x, y^*_{\mu, F}(\mu)).$$

As $g^c(x, y)$ is $l_{g^c,0}$-Lipschitz by Assumption 1, for any $\mu_1, \mu_2 \in \mathbb{R}^{d_c}_+$:

$$\|\nabla D_g(\mu_1) - \nabla D_g(\mu_2)\| = \|g^c(x, y^*_{\mu, g}(\mu_1)) - g^c(x, y^*_{\mu, g}(\mu_2))\|$$
$$\leq l_{g^c,0} \|y^*_{\mu, g}(\mu_1) - y^*_{\mu, g}(\mu_2)\| \leq \frac{l_{g^c,0}}{\alpha_g} \|\mu_1 - \mu_2\|, \tag{39}$$

and similarly,

$$\|\nabla D_F(\mu_1) - \nabla D_F(\mu_2)\| = \|g^c(x, y^*_{\mu, F}(\mu_1)) - g^c(x, y^*_{\mu, F}(\mu_2))\|$$
$$\leq l_{g^c,0} \|y^*_{\mu, F}(\mu_1) - y^*_{\mu, F}(\mu_2)\| \leq \frac{l_{g^c,0}}{\gamma \alpha_g - l_{f,1}} \|\mu_1 - \mu_2\|. \tag{40}$$

We can conclude that $D_g(\mu)$ and $D_F(\mu)$ are respectively $\frac{l_{g^c,0}}{\alpha_g}$ and $\frac{l_{g^c,0}}{\gamma \alpha_g - l_{f,1}}$-smooth. $\qquad \square$

In Algorithm 2, we are implementing an accelerated projected gradient descent on $-D(\mu)$ where the gradient bias is controlled by $T_y$. Following [16], the following lemma presents the convergence analysis of the accelerated method on smooth and convex functions.

**Lemma 11** (Section 5 and 6 in [16]). *Suppose $D(\mu)$ is concave and $l_{D,1}$-smooth. Consider the constrained problem $\max_{\mu \in \mathbb{R}^{d_c}_+} D(\mu)$. At iteration $t = 0, \ldots, T - 1$, perform accelerated projected gradient update with stepsize $\eta \leq \frac{1}{l_{D,1}}$ and initial value $\mu_0 = \mu_{-1}$:*

$$\mu_{t+\frac{1}{2}} = \mu_t + \frac{t - 1}{t + 2}(\mu_t - \mu_{t-1}) \tag{41a}$$

$$\mu_{t+1} = \text{Proj}_{\mathbb{R}_+^{d_c}}(\mu_{t+\frac{1}{2}} + \eta g_{t+\frac{1}{2}}) \tag{41b}$$

where $g_{t+\frac{1}{2}}$ is an $\epsilon^{1.5}$-approximate to $\nabla D(\mu_{t+\frac{1}{2}})$ satisfying $\|g_{t+\frac{1}{2}} - \nabla D(\mu_{t+\frac{1}{2}})\| = \mathcal{O}(\epsilon^{1.5})$, for a given accuracy $\epsilon > 0$. Denote $D^* = \max_{\mu \in \mathbb{R}_+^{d_c}} D(\mu)$, performing $T = \mathcal{O}(\epsilon^{-0.5})$ iterations leads to

$$D^* - D(\mu_T) = \mathcal{O}(\epsilon).$$

**Remark 2.** *The domain for $\mu \in \mathbb{R}_+^{d_c}$ can be replaced by any closed, convex, non-empty domain.*

In this way, we are ready to proceed to the **proof of Theorem 3**.

*proof of Theorem 3.* Algorithm 2 solves (7) and (5) by taking $L_g(\mu, y; x)$ and $L_F(\mu, y; x)$ respectively as $L(\mu, y)$ and run respectively iterations $T$ equals $T_g$ and $T_F$

We begin our proof with analysis on (7). The accelerated version of Algorithm 2 solves this by taking $L_g(\mu, y; x)$ in (7) with fixed $x \in \mathcal{X}$ as $L(\mu, y)$.

Fixing $\mu_{t+\frac{1}{2}}$, steps 4-6 are $T_y$-step projected gradient descent in $y$. As $L_g(\mu, y; x)$ is $(l_{g,1} + l_{g^c,1})$-smooth and $\alpha_g$-strongly convex in $y$, taking the inner loop stepsize $\eta_1$ as $\eta_{g,1} \leq \frac{1}{l_{g,1}+l_{g^c,1}}$ ensures linear convergence according to Lemma 7. Choosing $T_y = \mathcal{O}(\ln((\epsilon^{1.5})^{-1})) = \mathcal{O}(\ln(\epsilon_g^{-1}))$ leads to

$$\|y_{t+1} - y^*_{\mu,g}(\mu_{t+\frac{1}{2}})\| = \mathcal{O}(\epsilon_g^{1.5})$$

for target accuracy $\epsilon_g > 0$ where $y^*_{\mu,g}(\mu)$ is defined in (35).

In step 7, we update $\mu$ with a projected gradient descent step which takes $\nabla_\mu L(\mu_{t+\frac{1}{2}}, y_{t+1}) = g^c(x, y_{t+1})$ as an estimate of $\nabla D_g(\mu_{t+\frac{1}{2}}) = g^c(x, y^*_{\mu,g})$. The estimation bias is bounded by

$$\|\nabla_\mu L(\mu_{t+\frac{1}{2}}, y_{t+1}) - \nabla D_g(\mu_{t+\frac{1}{2}})\| = \|g^c(x, y_{t+1}) - g^c(x, y^*_{\mu,g}(\mu_{t+\frac{1}{2}}))\|$$
$$\leq l_{g^c,0}\|y_{t+1} - y^*_{\mu,g}(\mu_{t+\frac{1}{2}})\| = \mathcal{O}(\epsilon_g^{1.5}).$$

By Lemma 11, we can conclude the complexity is $\tilde{\mathcal{O}}(\epsilon_g^{-0.5})$ for conducting the accelerated version of Algorithm 2 on $L_g(\mu, y; x)$ in (7) as $L(\mu, y)$ to achieve

$$D_g(\mu_g^*(x)) - D_g(\mu_{T_g}) = \mathcal{O}(\epsilon_g). \tag{42}$$

Similarly, the complexity of the accelerated version of Algorithm 2 for (5) is $\tilde{\mathcal{O}}(\epsilon_F^{-0.5})$, i.e.,

$$D_F(\mu_F^*(x)) - D_F(\mu_{T_F}) = \mathcal{O}(\epsilon_F). \tag{43}$$

In the following, we are going to show that (42) and (43) and respectively sufficient to bound $\|\mu_{T_g} - \mu_g^*(x)\|$ and $\|\mu_{T_F} - \mu_F^*(x)\|$ considering Assumption 5 is satisfied.

As $D_g(\mu)$ and $D_F(\mu)$ are both concave in $\mu$ and $\mu \in \mathbb{R}_+^{d_c}$ is equivalent to $\mu \geq 0$, the problems

$$\max_{\mu \in \mathbb{R}_+^{d_c}} D_g(\mu) \quad \text{and} \quad \max_{\mu \in \mathbb{R}_+^{d_c}} D_F(\mu)$$

are respectively equivalent to the unconstrained problems

$$\max_{\mu \in \mathbb{R}^{d_c}} \tilde{D}_g(\mu) := D_g(\mu) + \lambda_g^\top \mu \quad \text{and} \quad \max_{\mu \in \mathbb{R}^{d_c}} \tilde{D}_F(\mu) := D_F(\mu) + \lambda_F^\top \mu$$

with the Lagrange multipliers $\lambda_g, \lambda_F$ being non-negative and finite in all dimension, i.e. $0 \leq \lambda_g < \infty$, $0 \leq \lambda_F < \infty$ according to Lagrange duality Theorem. Moreover, it is well known (Chapter 4 in [59]) that the Largrangian terms respectively equal zero when the problems attain the optimals. i.e.

$$\lambda_g^\top \mu_g^*(x) = 0 \quad \text{and} \quad \lambda_F^\top \mu_F^*(x) = 0. \tag{44}$$

Moreoever, the first-order stationary condition requires $\nabla \tilde{D}_g(\mu_g^*(x)) = \nabla D_g(\mu_g^*(x)) + \lambda_g = 0$ and $\nabla \tilde{D}_F(\mu_F^*(x)) = \nabla D_F(\mu_F^*(x)) + \lambda_F = 0$ and therefore

$$\nabla D_g(\mu_g^*(x)) = -\lambda_g \quad \text{and} \quad \nabla D_F(\mu_F^*(x)) = -\lambda_F. \tag{45}$$

In this way, for all $\mu \in \mathcal{B}(\mu_g^*(x); \delta_g) \cap \mathbb{R}_+^{d_c}$.

$$
\begin{aligned}
D_g(\mu_g^*(x)) - D_g(\mu) &= \int_{\tau=0}^1 \langle \nabla D_g(\mu + \tau(\mu_g^*(x) - \mu)), \mu_g^*(x) - \mu \rangle d\tau \\
&= \int_{\tau=0}^1 \frac{1}{\tau} \langle \nabla D_g(\mu_g^*(x)) - D_g(\mu + \tau(\mu_g^*(x) - \mu)), \tau(\mu - \mu_g^*(x)) \rangle d\tau \\
&\quad - \langle \nabla D_g(\mu_g^*(x)), \mu - \mu_g^*(x) \rangle \\
&\overset{(a)}{\geq} \int_0^1 C_{\delta_g} \|\mu - \mu_g^*(x)\|^2 \tau d\tau - \langle \nabla D_g(\mu_g^*(x)), \mu - \mu_g^*(x) \rangle \\
&\overset{(b)}{=} \frac{C_{\delta_g}}{2} \|\mu - \mu_g^*(x)\|^2 + \langle \lambda_g, \mu - \mu_g^*(x) \rangle \\
&\overset{(c)}{\geq} \frac{C_{\delta_g}}{2} \|\mu - \mu_g^*(x)\|^2,
\end{aligned}
$$

where $(a)$ uses (21a) in Assumption 5 and the fact that the $\mu, \mu_g^*(x) \in \mathcal{B}(\mu_g^*(x); \delta_g) \cap \mathbb{R}_+^{d_c}$ implies $\mu + \tau(\mu_g^*(x) - \mu) \in \mathcal{B}(\mu_g^*(x); \delta_g) \cap \mathbb{R}_+^{d_c}$; $(b)$ solves the integral and uses $\lambda_g = -\nabla D_g(\mu_g^*(x))$ in (45); and $(c)$ follows from the fact that $\langle \lambda, \mu_g^*(x) \rangle = 0$ in (44) and $\mu, \lambda_g \geq 0$.

Analogously, for all $\mu \in \mathcal{B}(\mu_F^*(x); \delta_F) \cap \mathbb{R}_+^{d_c}$, it follows that

$$
D_F(\mu_F^*(x)) - D_F(\mu) \geq \frac{C_{\delta_F}}{2} \|\mu - \mu_F^*(x)\|^2.
$$

In this way, for arbitrary $\epsilon_g < \frac{C_{\delta_g}}{2} \delta_g$, the complexity of Algorithm 2 to solve (7) is $\tilde{\mathcal{O}}(\epsilon_F^{-0.5})$, i.e.,

$$
\begin{aligned}
\|\mu_{T_g} - \mu_g^*(x)\|^2 &= \mathcal{O}(\epsilon_g), \\
\text{and } \|y_{T_g} - y_g^*(x)\|^2 &\leq \|y_{T_g} - y_g^*(\mu_{T_g}; x)\|^2 + \|\mu_{T_g} - \mu_g^*(x)\|^2 \\
&\leq (1/\alpha_g + 1)\|\mu_{T_g} - \mu_g^*(x)\|^2 = \mathcal{O}(\epsilon_g).
\end{aligned}
$$

At each iteration $t$ in Algorithm 1, $x = x_t$, and the output $(y_{T_g}, \mu_{T_g})$ is chosen as $(y_{g,t}^{T_g}, \mu_{g,t}^{T_g})$.

Similarly, to solve (5), for arbitrary $\epsilon_g < \frac{C_{\delta_g}}{2} \delta_g$, applying Algorithm 2 with complexity $\tilde{\mathcal{O}}(\epsilon_F^{-0.5})$ to achieve (43) can achieve

$$
\begin{aligned}
\|\mu_{T_F} - \mu_F^*(x)\|^2 &= \mathcal{O}(\epsilon_F), \\
\text{and } \|y_{T_F} - y_F^*(x)\|^2 &\leq \|y_{T_F} - y_F^*(\mu_{T_F}; x)\|^2 + \|\mu_{T_F} - \mu_F^*(x)\|^2 \\
&\leq (1/\alpha_F + 1)\|\mu_{T_F} - \mu_F^*(x)\|^2 = \mathcal{O}(\epsilon_F).
\end{aligned}
$$

At each iteration $t$ in Algorithm 1, $x = x_t$, and the output $(y_{T_F}, \mu_{T_F})$ is chosen as $(y_{F,t}^{T_F}, \mu_{F,t}^{T_F})$.

This completes the proof. $\qquad\square$

**Remark 3.** *Under the same assumptions as in Theorem 3, we cam choose $\eta_{g,1} \leq (l_{g,1} + l_{g^c,1})^{-1}$, $\eta_{g,2} \leq \frac{\alpha_g}{l_{g^c,0}}$ as stepsizes for running the accelerated version of Algorithm 2 to solve (7), and $\eta_{F,1} \leq (l_{f,1} + \gamma l_{g,1} + l_{g^c,1})^{-1}$, $\eta_{F,2} \leq \frac{\gamma\alpha_g - l_{f,1}}{l_{g^c,0}}$ as the ones for (5).*

### D.3 Proof of Theorem 4

In this section, we consider

$$
g^c(x, y) = g_1^c(x)^\top y - g_2^c(x) \tag{46}
$$

being affine in $y$, and $\mathcal{Y} = \mathbb{R}^{d_y}$.

Therefore, for a fixed $x$, taking either $L_g(\mu, y; x)$ in (7) or $L_F(\mu, y; x)$ in (5) as $L(\mu, y)$ fits into a special case of *strongly-convex-concave saddle point problems* in the following form:

$$
\max_{\mu \in \mathbb{R}_+^{d_c}} \min_{y \in \mathbb{R}^{d_y}} L(\mu, y) = -h_1(\mu) + y^\top A\mu + h_2(y) \tag{47}
$$

where $h_1(\mu)$ is smooth and linear (concave) in $\mu$ and $h_2(y)$ is smooth and strongly convex in $y$. Specifically, for $L_g(\mu, y; x)$ as $L(\mu, y)$, $h_1(\mu) = \langle g_2^c(x), \mu \rangle$ is 0-smooth and linear (concave), $A = g_1^c(x)$, and $h_2(y) = g(x, y)$ is $l_{g,1}$-smooth and $\alpha_g$-strongly convex.

In this context, performing PGD on $L(\mu, y)$ on $y$ is equivalent to a gradient descent step since $\mathcal{Y} = \mathbb{R}^{d_y}$. The following lemma summarizes the error of $\|y_t - y_\mu^*(\mu_t)\|$ where $y_\mu^*(\mu)$ is defined in (16) and the update of $\mu_{t+1}$ and $y_{t+1}$ is the non-accelerated version of Algorithm 2 with $T_y = 1$.

**Lemma 12** (Update of $\|y_t - \nabla h_2^*(-A\mu_t)\|$). *Consider the problem* (47) *where $A$ is of full column rank and $h_2(y)$ is $\alpha_{h_2}$-strongly convex and $l_{f,1}$-smooth. $y_\mu^*(\mu)$ defined in* (16) *satisfies*

$$y_\mu^*(\mu) = \nabla h_2^*(-A\mu) \tag{48}$$

*where $h_2^*(y)$ is the conjugate function of $h_2(y)$ by Definition 5. At iteration t, $\mu_t, y_t$ are known, and conduct $y_{t+1} = y_t - \eta_1 \nabla_y L(\mu_t, y_t)$, a gradient descent step for $L(\mu_t, y)$ in $y$. This gives*

$$\|y_{t+1} - \nabla h_2^*(-A\mu_t)\| \le (1 - \eta_1 \alpha_{h_2}/2)\|y_t - \nabla h_2^*(-A\mu_t)\| \tag{49}$$

*when $\eta_1 \le \frac{1}{l_{h_2,1}}$. Additionally, given another $\mu_{t+1}$, we know*

$$\|y_{t+1} - \nabla h_2^*(-A\mu_{t+1})\| \le (1 - \eta_1 \alpha_{h_2}/2)\|y_t - \nabla h_2^*(-A\mu_t)\| + \frac{\sigma_{\max}(A)}{\alpha_{h_2}}\|\mu_{t+1} - \mu_t\|. \tag{50}$$

*Proof.* Recall the definition $y_\mu^*(\mu) = \arg\min_y L(\mu, y)$ following (16). The first-order stationary optimality condition requires that for any given $\mu$, it holds

$$\nabla_y L(\mu, y_\mu^*) = A\mu + \nabla h_2(y_\mu^*) = 0 \quad \Leftrightarrow \quad \nabla h_2(y_\mu^*) = -A\mu.$$

As the mapping $\nabla h_2$ and $\nabla h_2^*$ are the inverse of each other according to Lemma 4, for any $\mu$:

$$y_\mu^* = \nabla h_2^*(-A\mu).$$

At iteration $t$, conducting a gradient descent step on $L(\mu_t, y)$ gives $y_{t+1} = y_t - \eta_1 \nabla_y L(\mu_t, y_t)$. As $L(\mu_t, y)$ is $\alpha_{h_2}$-strongly convex and $l_{h_2,1}$-smooth in $y$, following Lemma 7, take $\eta_1 \le \frac{1}{l_{h_2,1}}$, we have

$$\|y_{t+1} - \nabla h_2^*(-A\mu_t)\| \le (1 - \eta_1 \alpha_{h_2}/2)\|y_t - \nabla h_2^*(-A\mu_t)\|.$$

Following triangle inequality, we also have

$$\begin{aligned}
&\|y_{t+1} - \nabla h_2^*(-A\mu_{t+1})\| \\
&\le \|y_{t+1} - \nabla h_2^*(-A\mu_t)\| + \|\nabla h_2^*(-A\mu_t) - \nabla h_2^*(-A\mu_{t+1})\| \\
&\le (1 - \eta_1 \alpha_{h_2}/2)\|y_t - \nabla h_2^*(-A\mu_t)\| + \frac{\sigma_{\max}(A)}{\alpha_{h_2}}\|\mu_{t+1} - \mu_t\|
\end{aligned} \tag{51}$$

where the second term comes from the smoothness of the conjugate function (see Lemma 4). $\square$

In (50), the update behavior of $\|y_t - \nabla h_2^*(-A\mu_t)\|$ depends on $\|\mu_{t+1} - \mu_t\|$. Therefore, we are interested in the the update behavior of $\|\mu_{t+1} - \mu_t\|$ where $\mu_{t+1} = \text{Proj}_{\mathbb{R}^{dc}}(\mu_t + \eta \nabla_\mu L(\mu_t, y_{t+1}))$ as in the non-accelerated version of Algorithm 2 with $T_y = 1$. Before proceeding, we would like to look into the properties of $D(\mu)$ defined in (16) under the setting (47):

$$D(\mu) = \min_y -h_1(\mu) + \langle y, A\mu \rangle + h_2(y) \tag{52}$$

where $h_1(\mu)$ is smooth and linear (concave) in $\mu$, $h_2(y)$ is smooth and strongly convex in $y$, and $A$ is of full rank in column. The next lemma shows that $D(\mu)$ features strong concavity and smoothness.

**Lemma 13** (Smoothness and strongly concavity of $D(\mu)$). *Suppose $h_1$ is concave and $l_{h_1,1}$-smooth, $h_2$ is $\alpha_{h_2}$-strongly convex and $l_{h_2,1}$-smooth, and $A$ is full column rank. Then $D(\mu)$ in* (52) *satisfies*

$$D(\mu) = -h_1(\mu) - h_2^*(-A\mu),$$

*and is $\frac{\sigma_{\min}^2(A)}{l_{h_2,1}}$-strongly concave and $(l_{h_1,1} + \frac{\sigma_{\max}^2(A)}{\alpha_{h_2}})$-smooth with respect to $\mu$.*

*Proof.* Following Definition 5, we have

$$D(\mu) = -h_1(\mu) - h_2^*(-A\mu)$$

where $h_2^*(y)$ is $\frac{1}{l_{h_2,1}}$-strongly convex and $\frac{1}{\alpha_{h_2}}$-smooth according to Lemma 4.

For all $\mu_1, \mu_2$,

$$\begin{aligned}
-D(\mu_1) - (-D(\mu_2)) =& h_2^*(-A\mu_1) - h_2^*(-A\mu_2) + h_1(\mu_1) - h_1(\mu_2) \\
\geq& \langle \frac{\partial h_2^*(-A\mu_2)}{\partial - A\mu_2}, -A\mu_1 + A\mu_2 \rangle + \frac{1/l_{h_2,1}}{2}\|A\mu_1 - A\mu_2\|^2 \\
& + \langle \nabla h_1(\mu_2), \mu_1 - \mu_2 \rangle \rangle \\
\geq& \langle \nabla D(\mu_2), \mu_1 - \mu_2 \rangle + \frac{\sigma_{\min}^2(A)/l_{h_2,1}}{2}\|\mu_1 - \mu_2\|^2.
\end{aligned}$$

where the first inequality follows the strong convexity of $h_2^*(y)$ and the fact that $-h_1(\mu)$ is convex as $h_1(y)$ is concave. and the second inequality follows the chain rule to formulate $\nabla D(\mu_2)$. Therefore, $-D(\mu)$ is $\frac{\sigma_{\min}^2(A)}{l_{h_2,1}}$-strongly convex, and $D(\mu)$ is $\frac{\sigma_{\min}^2(A)}{l_{h_2,1}}$-strongly concave.

Moreover $D(\mu)$ is $(l_{h_1,1} + \frac{\sigma_{\max}^2(A)}{\alpha_{h_2}})$-smooth as

$$\begin{aligned}
D(\mu_1) - D(\mu_2) =& -h_2^*(-A\mu_1) - (-h_2^*(-A\mu_2)) - h_1(\mu_1) + h_1(\mu_2) \\
\leq& \langle \frac{\partial - h_2^*(-A\mu_2)}{\partial - A\mu_2}, -A\mu_1 - (-A\mu_2) \rangle + \frac{1/\alpha_{h_2}}{2}\| - A\mu_1 - (-A\mu_2)\|^2 \\
& + \langle -\nabla h_1(\mu_2), \mu_1 - \mu_2 \rangle \rangle + \frac{l_{h_1,1}}{2}\|\mu_1 - \mu_2\|^2 \\
\leq& \langle \nabla D(\mu_2), \mu_1 - \mu_2 \rangle + \frac{l_{h_1,1} + \frac{\sigma_{\max}^2(A)}{\alpha_{h_2}}}{2}\|\mu_1 - \mu_2\|^2.
\end{aligned}$$

The first inequality holds as $h_2^*(y)$ and $h_1(\mu)$ are smooth. The second follows the chain rule.

Note $\sigma_{\max}(A) \geq \sigma_{\min}(A) > 0$ as $A$ is full column rank. This completes the proof. $\square$

Knowing $D(\mu)$ has such favorable properties, we next analyze the update of $\|\mu_{t+1} - \mu_t\|$, where $\{\mu_t\}$ is the sequence generated in the non-accelerated version of Algorithm 2 with $T_y = 1$.

**Lemma 14** (Update of $\|\mu_{t+1} - \mu_t\|$)**.** *Consider the problem in (47) where $h_1$ is concave and $l_{h_1,1}$-smooth, $h_2$ is $\alpha_{h_2}$-strongly convex and $l_{h_2,1}$-smooth, and $A$ is full column rank. Running the non-accelerated version of Algorithm 2 with $T_y = 1$ and $\eta_1 \leq l_{h_2,1}^{-1}$ gives*

$$\begin{aligned}
\frac{1}{\eta_2}\|\mu_{t+1} - \mu_t\| \leq& \left( l_{h_1,1} + \frac{\sigma_{\max}^2(A)}{\alpha_{h_2}} \right) \|\mu_t - \mu^*\| \\
& + \sigma_{\max}(A)(1 - \eta_1\alpha_{h_2}/2)\|y_t - \nabla h_2^*(-A\mu_t)\| + \|\lambda\|
\end{aligned} \tag{53}$$

*where the constant $\lambda$ satisfies $0 \leq \lambda < \infty$ and $\mu^* = \arg\max_{\mu \in \mathbb{R}_+^{d_c}} D(\mu)$ with $D(\mu)$ defined in (52).*

*Proof.* According to Lemma 13,

$$D(\mu) = \min_{y \in \mathbb{R}^{d_y}} -h_1(\mu) + y^\top A\mu + h_2(y) = -h_1(\mu) - h_2^*(-A\mu)$$

is $\frac{\sigma_{\min}^2(A)}{l_{h_2,1}}$-strongly concave and $(l_{h_1,1} + \frac{\sigma_{\max}^2(A)}{\alpha_{h_2}})$-smooth with respect to $\mu$. Moreover, the problem $\max_{\mu \in \mathbb{R}_+^{d_c}} D(\mu)$ is equivalent to the unconstrained problem with the Lagrange multiplier

$$\max_{\mu \in \mathbb{R}^{d_c}} \tilde{D}(\mu) := D(\mu) + \lambda^\top \mu$$

where unique $\lambda$ is non-negative and finite in all dimension, i.e. $0 \leq \lambda < \infty$, as $D(\mu)$ is strongly convex and $\mu \in \mathbb{R}_+^{d_c}$ is equivalent to $\mu \geq 0$ satisfying the LICQ condition (Lemma 5). In this way,

$$\nabla \tilde{D}(\mu) = \nabla D(\mu) + \lambda = -\nabla h_1(\mu) + A^\top \nabla h_2^*(-A\mu) + \lambda. \tag{54}$$

We can see that $\tilde{D}(\mu)$ is smooth and strongly concave with the same modulus as $D(\mu)$. The first-order stationary condition requires

$$\nabla \tilde{D}(\mu^*) = -\nabla h_1(\mu^*) + A^\top \nabla h_2^*(-A\mu^*) + \lambda = 0. \tag{55}$$

In this way,

$$\frac{1}{\eta_2}\|\mu_{t+1} - \mu_t\| = \frac{1}{\eta_2}\| \operatorname{Proj}_{\mathbb{R}_+^{d_c}} \left(\mu_t + \eta_2(-\nabla h_1(\mu_t) + A^\top y_{t+1})\right) - \mu_t\|$$

$$\overset{(a)}{\leq} \| -\nabla h_1(\mu_t) + A^\top y_{t+1}\|$$

$$= \| -\nabla h_1(\mu_t) + A^\top \nabla h_2^*(-A\mu_t) + \lambda + A^\top y_{t+1} - A^\top \nabla h_2^*(-A\mu_t) - \lambda\|$$

$$\overset{(b)}{\leq} \|\nabla \tilde{D}(\mu_t)\| + \sigma_{\max}(A)\|y_{t+1} - \nabla h_2^*(-A\mu_t)\| + \|\lambda\|$$

$$\overset{(c)}{\leq} \|\nabla \tilde{D}(\mu_t) - \nabla \tilde{D}(\mu^*)\| + \sigma_{\max}(A)(1 - \eta_1 \alpha_{h_2}/2)\|y_t - \nabla h_2^*(-A\mu_t)\| + \|\lambda\|$$

$$\overset{(d)}{\leq} \left(l_{h_1,1} + \frac{\sigma_{\max}^2(A)}{\alpha_{h_2}}\right)\|\mu_t - \mu^*\| + \sigma_{\max}(A)(1 - \eta_1 \alpha_{h_2}/2)\|y_t - \nabla h_2^*(-A\mu_t)\| + \|\lambda\|$$

Inequality $(a)$ comes from the non-expansiveness (1-Lipschitzness) of the projection operation, $(b)$ follows triangle inequality and uses (54), $(c)$ uses (55) and (49) in Lemma 12, and $(d)$ comes from the smoothness of $\tilde{D}(\mu)$, which is of the same modulus as $D(\mu)$. This completes the proof. $\qquad \square$

In (53), the update behavior of $\|\mu_{t+1} - \mu_t\|$ depends on $\|\mu_t - \mu^*\|$. We further look into the update of $\|\mu_t - \mu^*\|$ and summarize in the following lemma the bound of the update of $\|\mu_t - \mu^*\|$.

**Lemma 15** (Update of $\|\mu_t - \mu^*\|$). *Consider the problem in (47) where $h_1$ is concave and $l_{h_1,1}$-smooth, $h_2$ is $\alpha_{h_2}$-strongly convex and $l_{h_2,1}$-smooth, and $A$ is full column rank. Conduct the non-accelerated version of Algorithm 2 with $T_y = 1$, gives*

$$\|\mu_{t+1} - \mu^*\| \leq \left(1 - \eta_2 \frac{\sigma_{\min}^2(A)}{2l_{h_2,1}}\right)\|\mu_t - \mu^*\| + \eta_2 \sigma_{\max}(A)(1 - \eta_1 \alpha_{h_2}/2)\|y_t - \nabla h_2^*(-A\mu_t)\|. \tag{56}$$

*when $\eta_1 \leq l_{h_2,1}^{-1}$ and $\eta_2 \leq \left(l_{h_1,1} + \frac{\sigma_{\max}^2(A)}{\alpha_{h_2}}\right)^{-1}$. Here $\mu^* = \arg\min_{\mu \in \mathbb{R}_+^{d_c}} D(\mu)$ where $D(\mu)$ is defined in (52).*

*Proof.* Define an auxiliary update as

$$\tilde{\mu}_{t+1} := \operatorname{Proj}_{\mathbb{R}_+^{d_c}}(\mu_t + \eta_2 \nabla D(\mu_t)) = \operatorname{Proj}_{\mathbb{R}_+^{d_c}}\left(\mu_t + \eta_2(-\nabla h_1(\mu_t) + A^\top \nabla h_2^*(-A\mu_t))\right). \tag{57}$$

This is a projected gradient descent on strongly convex $-D(\mu)$. As $\mathbb{R}_+^{d_c}$ is closed and convex, following Lemma 7, for $\eta_2 \leq \left(l_{h_1,1} + \frac{\sigma_{\max}^2(A)}{\alpha_{h_2}}\right)^{-1}$, where $\left(l_{h_1,1} + \frac{\sigma_{\max}^2(A)}{\alpha_{h_2}}\right)$ is the modulus for smoothness of $D(\mu)$ by Lemma 13, we have

$$\|\tilde{\mu}_{t+1} - \mu^*\| \leq \left(1 - \eta_2 \frac{\sigma_{\min}^2(A)}{2l_{h_2,1}}\right)\|\mu_t - \mu^*\|.$$

As the real update is $\mu_{t+1} = \operatorname{Proj}_{\mathbb{R}_+^{d_c}}\left((\mu_t + \eta_2(-\nabla h_1(\mu_t) + A^\top y_t)\right)$, by the non-expansiveness (1-Lipschitzness) of projection operation, we have

$$\|\tilde{\mu}_{t+1} - \mu_{t+1}\| \leq \|\eta_2 A^\top(y_{t+1} - \nabla h_2^*(-A\mu_t))\| \leq \eta_2 \sigma_{\max}(A)\|y_{t+1} - \nabla h_2^*(-A\mu_t)\|$$

By triangle inequality and (49), we have

$$\|\mu_{t+1} - \mu^*\| \leq \left(1 - \eta_2 \frac{\sigma_{\min}^2(A)}{2l_{h_2,1}}\right)\|\mu_t - \mu^*\| + \eta_2 \sigma_{\max}(A)\|y_{t+1} - \nabla h_2^*(-A\mu_t)\|$$

$$\leq \left(1 - \eta_2 \frac{\sigma_{\min}^2(A)}{2l_{h_2,1}}\right)\|\mu_t - \mu^*\| + \eta_2 \sigma_{\max}(A)(1 - \eta_1 \alpha_{h_2}/2)\|y_t - \nabla h_2^*(-A\mu_t)\|. \tag{58}$$

This completes the proof. $\qquad \square$

We are ready to proceed with the convergence analysis for the single-loop algorithm (Algorithm 2) without acceleration and $T_y = 1$, on the problems (47), which is a general form to (7) and (5). In this way, Theorem 4 follows directly from the following theorem.

**Theorem 6.** *Suppose $L(\mu, y)$ is in the form of (47) where $A$ is full column rank, $h_1$ is concave and $l_{h_1,1}$-smooth, $h_2$ is $\alpha_{h_2}$-strongly convex and $l_{h_2,1}$-smooth satisfying $l_{h_1,1} = \mathcal{O}(1)$, $l_{h_2,1}, l_{\alpha_2} \geq \mathcal{O}(1)$, and $\frac{l_{h_2,1}}{\alpha_{h_2}} = \mathcal{O}(1)$. Conduct the non-accelerated version of Algorithm 2 with $T_y = 1$. For arbitrary small positive $\epsilon \leq \left( \frac{4 l_{h_2,1} \sigma_{\max}(A)}{\alpha_{h_2} \sigma_{\min}^2(A)} (l_{h_1,1} + \frac{\sigma_{\max}^2(A)}{\alpha_{h_2}}) \right)^{-1}$, when $\eta_1 = \mathcal{O}(\frac{1}{l_{h_2,1}}) \leq \frac{1}{l_{h_2,1}}$ and $\eta_2 = \mathcal{O}(\epsilon) \leq \frac{1}{l_{h_1,1} + \sigma_{\max}^2(A)/\alpha_{h_2}}$, the algorithm yields output $(\mu_T, y_T)$ such that*

$$\|\mu_T - \mu^*\|^2 < \epsilon, \quad and \quad \|y_T - y^*\|^2 < \epsilon$$

*with complexity $T = \mathcal{O}(\ln(\epsilon^{-1}))$. Here, $(\mu^*, y^*) = \arg\max_{\mu \in \mathbb{R}^{d_c}} \min_{y \in \mathbb{R}^{d_y}} L(\mu, y)$.*

*Proof.* For some positive constant $\rho > 0$, denote

$$P_t := \rho \|\mu_t - \mu^*\| + \|y_t - \nabla h_2^*(-A\mu_t)\|. \tag{59}$$

Plugging (50) in Lemma 12 , (53) in Lemma 14, and (56) in Lemma 15 to (59), we know

$$P_{t+1} = \rho \|\mu_{t+1} - \mu^*\| + \|y_{t+1} - \nabla h_2^*(-A\mu_{t+1})\|$$

$$\leq \rho \left( \left( 1 - \eta_2 \frac{\sigma_{\min}^2(A)}{2 l_{h_2,1}} \right) \|\mu_t - \mu^*\| + \eta_2 \sigma_{\max}(A)(1 - \eta_1 \alpha_{h_2}/2) \|y_t - \nabla h_2^*(-A\mu_t)\| \right)$$

$$+ (1 - \eta_1 \alpha_{h_2}/2) \|y_t - \nabla h_2^*(-A\mu_t)\| + \frac{\sigma_{\max}(A)}{\alpha_{h_2}} \eta_2$$

$$\times \left( (l_{h_1,1} + \frac{\sigma_{\max}^2(A)}{\alpha_{h_2}}) \|\mu_t - \mu^*\| + \sigma_{\max}(A)(1 - \eta_1 \alpha_{h_2}/2) \|y_t - \nabla h_2^*(-A\mu_t)\| + \|\lambda\| \right)$$

$$= \left( 1 - \eta_2 \frac{\sigma_{\min}^2(A)}{2 l_{h_2,1}} + \frac{1}{\rho} \frac{\sigma_{\max}(A)}{\alpha_{h_2}} \eta_2 (l_{h_1,1} + \frac{\sigma_{\max}^2(A)}{\alpha_{h_2}}) \right) \rho \|\mu_t - \mu^*\|$$

$$+ (1 - \eta_1 \alpha_{h_2}/2) \left( 1 + \rho \eta_2 \sigma_{\max}(A) + \frac{\sigma_{\max}^2(A)}{\alpha_{h_2}} \eta_2 \right) \|y_t - \nabla h_2^*(-A\mu_t)\| + \frac{\sigma_{\max}(A)}{\alpha_{h_2}} \eta_2 \|\lambda\|.$$

To construct $P_{t+1} \leq (1 - c)P_t + \frac{\sigma_{\max}(A)}{\alpha_{h_2}} \eta_2 \|\lambda\|$ for some constant $0 < c < 1$, it is sufficient to find $\eta_1 \leq \frac{1}{l_{h_2,1}}, \eta_2 \leq \frac{1}{(l_{h_1,1} + \frac{\sigma_{\max}^2(A)}{\alpha_{h_2}})}$, and $\rho > 0$ such that

$$\begin{cases} 0 < \left( 1 - \eta_2 \frac{\sigma_{\min}^2(A)}{2 l_{h_2,1}} + \frac{1}{\rho} \frac{\sigma_{\max}(A)}{\alpha_{h_2}} \eta_2 (l_{h_1,1} + \frac{\sigma_{\max}^2(A)}{\alpha_{h_2}}) \right) \leq 1 - \eta_2 \frac{\sigma_{\min}^2(A)}{4 l_{h_2,1}} < 1 \\ 0 < (1 - \eta_1 \alpha_{h_2}/2) \left( 1 + \rho \eta_2 \sigma_{\max}(A) + \frac{\sigma_{\max}^2(A)}{\alpha_{h_2}} \eta_2 \right) \leq (1 - \eta_1 \alpha_{h_2}/2)(1 + \eta_1 \alpha_{h_2}/2) < 1 \end{cases}$$

This can be obtained when

$$\begin{cases} \rho \geq \frac{4 l_{h_2,1} \sigma_{\max}(A)}{\alpha_{h_2} \sigma_{\min}^2(A)} (l_{h_1,1} + \frac{\sigma_{\max}^2(A)}{\alpha_{h_2}}) \\ \eta_2 \leq \frac{\eta_1 \alpha_{h_2}}{2 \left( \rho \sigma_{\max}(A) + \frac{\sigma_{\max}^2(A)}{\alpha_{h_2}} \right)} \end{cases} \tag{60}$$

Conditions in (60) can be satisfied when $\epsilon > 0$ is sufficiently small such that $\rho = \epsilon^{-1} \geq \frac{4 l_{h_2,1} \sigma_{\max}(A)}{\alpha_{h_2} \sigma_{\min}^2(A)} (l_{h_1,1} + \frac{\sigma_{\max}^2(A)}{\alpha_{h_2}})$, $\eta_1 = \mathcal{O}(\frac{1}{l_{h_2,1}})$ and $\eta_2 = \mathcal{O}(\frac{\alpha_{h_2}}{l_{h_2,1}} \rho^{-1}) = \mathcal{O}(\epsilon^{-1})$. In this way,

$$P_{t+1} \leq (1 - c)P_t + \mathcal{O}(\alpha_{h_2}^{-1} \epsilon)$$

where $c > 0$ is of the order $\mathcal{O}(\epsilon)$. Iteration gives

$$P_t \leq (1 - c)^t P_0 + \mathcal{O}(\alpha_{h_2}^{-1}). \tag{61}$$

Notice $\mathcal{O}(\alpha_{h_2}^{-1}) < \mathcal{O}(1)$ and $P_0 = \mathcal{O}(\epsilon^{-1})$ as $\rho = \epsilon^{-1}$. In this way, there exist $T_1 = \mathcal{O}(\ln(\epsilon^{-1}))$ such that for all $t > T_1$, $(1-c)^t P_0 = \mathcal{O}(1)$ and $\mathcal{O}(\alpha_{h_2}^{-1}) \leq \mathcal{O}(1)$. Accordingly, we can achieve

$$P_t = \mathcal{O}(1), \quad \forall t > T_1.$$

Moreover, as $P_t = \epsilon^{-1}\|\mu_t - \mu^*\| + \|y_t - \nabla h_2^*(-A\mu_t)\|$,

$$\|\mu_t - \mu^*\| \leq \epsilon P_t = \mathcal{O}(\epsilon), \quad \forall t > T_1. \tag{62}$$

Furthermore, choose $\eta_1 = \mathcal{O}(\frac{1}{l_{h_2,1}})$ satisfying $\eta_1 \leq \frac{1}{l_{h_2,1}}$, for $t > T_1$,

$$\|y_t - \nabla h_2^*(-A\mu_t)\| \leq (1 - \eta_1\alpha_{h_2}/2)^{t-T_1}\|y_{T_1} - \nabla h_2^*(-A\mu_{T_1})\| + \mathcal{O}(\epsilon).$$

This is an iteration outcome using (50) in Lemma 12, (62), and the fact that $\eta_1\alpha_{h_2}/2 = \mathcal{O}(\frac{\alpha_{h_2}}{l_{h_2,1}}) = \mathcal{O}(1)$. In this way, for another $T_2 = \mathcal{O}(\ln(\epsilon^{-1}))$ steps, we have

$$\|y_t - \nabla h_2^*(-A\mu_t)\| = \mathcal{O}(\epsilon),$$
$$\text{and} \quad \|y_t - y^*\| \leq \|y_t - \nabla h_2^*(-A\mu_t)\| + \|\nabla h_2^*(-A\mu_t) - \nabla h_2^*(-A\mu^*)\|$$
$$\leq \|y_t - \nabla h_2^*(-A\mu_t)\| + \frac{\sigma_{\max}^2(A)}{\alpha_{h_2}}\|\mu_t - \mu^*\|$$
$$= \mathcal{O}\left(\epsilon + \alpha_{h_2}^{-1}\epsilon\right) = \mathcal{O}(\epsilon), \quad \forall t > T_1 + T_2.$$

We can see that the algorithm converges linearly with complexity $\mathcal{O}(T_1 + T_2) = \mathcal{O}(\ln(\epsilon^{-1}))$. In this way, obtaining

$$\|y_T - y^*\|^2 = \mathcal{O}(\epsilon) \quad \text{and} \quad \|\mu_T - \mu^*\|^2 = \mathcal{O}(\epsilon),$$

requires complexity $T = \mathcal{O}(\ln((\sqrt{\epsilon})^{-1})) = \mathcal{O}(\ln(\epsilon^{-1}))$. This completes the proof. $\square$

**Remark 4.** *Under the same assumptions as in Theorem 4, we cam choose $\eta_{g,2} \lesssim (l_{g,1})^{-1}$, $\eta_{g,1} \leq \alpha_g^2\epsilon_g(2s_{\max}\alpha_g + s_{\max}^2\epsilon^{-1})^{-1}\eta_{g,2}$ as stepsizes for running the single-loop version of Algorithm 2 to solve (7), and $\eta_{F,2} \lesssim (l_{f,1}+\gamma l_{g,1})^{-1}$, $\eta_{F,2} \leq (\gamma\alpha_g - l_{f,1})^2\epsilon_F(2s_{\max}(\gamma\alpha_g - l_{f,1}) + s_{\max}^2\epsilon^{-1})^{-1}\eta_{F,2}$ as the ones for (5).*

# E   Applications to Hyperparameter Optimization for SVM

In this section, we provide additional details about the SVM model training experiment for the linear SVM model, including the problem formulation and analysis of the results.

## E.1   Problem formulation

SVMs train a machine learning model by finding the optimal hyperplane that separates data points of different classes with the maximum margin $w$. Misclassification is not tolerated for hard-margin SVMs. In contrast, some samples are allowed to be misclassified for soft-margin SVMs. Specifically, one first introduces variables $\xi_i$, which measure the violation associated with the classification of sample $i$, and then augments the original SVM objective with the norm of $\xi$, which is a vector collecting the values of $\xi_i$ for all the samples in the training set.

BLO can be applied to the hyperparameter selection task of SVM. For example, it can be used to choose the value of the regularization parameters during soft-margin linear SVM training. Let us consider a classification problem and define $\mathcal{D}_{\text{tr}} := \{(z_{\text{tr},i}, l_{\text{tr},i})\}_{i=1}^{|\mathcal{D}_{\text{tr}}|}$ as the training set, with $z_{\text{tr},i}$ being the input feature vector for sample $i$ and $l_{\text{tr},i}$ being its associated binary label. Similarly, let $\mathcal{D}_{\text{val}} := \{(z_{\text{val}}, l_{\text{val}})\}_{i=1}^{|\mathcal{D}_{\text{val}}|}$ be the validation set. For a linear classification problem, the parameters of the SVM are $w$, a vector of coefficients with the same size as $z_{\text{tr},i}$, and the intercept $b$.

In short, we are interested in the following constrained BLO problem

$$\min_c \quad \mathcal{L}_{\mathcal{D}_{\text{val}}}(w^*, b^*) = \sum_{(z_{\text{val}}, l_{\text{val}}) \in \mathcal{D}_{\text{val}}} \exp\left(1 - l_{\text{val}}\left(z_{\text{val}}^\top w^* + b^*\right)\right) + \frac{1}{2}\|c\|^2 \tag{63a}$$

$$\text{with } w^*, b^*, \xi^* = \arg\min_{w,b,\xi} \frac{1}{2}\|w\|^2 \tag{63b}$$

$$\text{s.t. } l_{\text{tr},i}(z_{\text{tr},i}^\top w + b) \geq 1 - \xi_i \quad \forall\, i \in \{1, \ldots, |\mathcal{D}_{\text{tr}}|\} \tag{63c}$$

$$\xi_i \leq c_i \qquad\qquad\qquad \forall\, i \in \{1, \ldots, |\mathcal{D}_{\text{tr}}|\}. \tag{63d}$$

The first term of upper-level objective (63a) is a validation loss, evaluated on the validation set $\mathcal{D}_{\text{val}}$, the second is a regulation term on the upper-level variable $c$; the lower-level problem is designed to train the parameters of the hyperplane on the training set $\mathcal{D}_{\text{tr}}$, with the soft margin violation $\xi_i$ being upper bounded by the hyperparameter $c_i$ [cf. (63d), which are the CCs]. The lower-level objective (63b) focuses on maximizing the margin by minimizing $\|w\|^2$ while allowing for violations $\xi$ to the separating hyperplane, which are regulated by the hyperparameter $c$. The BLO formulation aims to adjust the hyperparameter $c$ using the validation loss (upper-level objective), ensuring that the model parameters (lower-level variables) are optimal for the training dataset.

### E.2 Experiment details

In this section, we present detailed experimental results for the training of the SVM model in (63) using our BLOCC algorithm. We compare our algorithm to two baselines, LV-HBA [75] and GAM [72], both designed for BLO problems with inequality CCs. We use $\gamma = 12$ and $\eta = 0.01$ for running our BLOCC and we apply $\alpha = 0.01$, $\gamma_1 = 0.1$, $\gamma_2 = 0.1$, $\eta = 0.001$ for LV-HBA and $\alpha = 0.05$, $\epsilon = 0.005$ for GAM respectively. The hyper-parameters for LV-HBA and GAM are the ones used in the SVM experiments in their paper. The GAM algorithm often encounters issues with matrix inversion, as the matrix required for the GAM algorithm to solve the problem in (63) can be singular, preventing it from finding a solution. Consequently, only BLOCC and LV-HBA solve the problem in (63), while GAM uses a different formulation as introduced in [72]. We compare the algorithms based on validation loss (the first term in the upper-level objective) and classification accuracy. These metrics are computed for both validation and test datasets. We evaluate the algorithms on two datasets: diabetes [20] and fourclass [29].

The detailed results are shown in Figure 4 and 5. Ten realizations of the problem are run, each with different training and validation sets. The plots show both the mean (line) and standard deviation (shaded region). The results reveal that our algorithm converges faster in terms of accuracy and loss, achieving a lower loss value than the alternatives for both datasets, in validation and test sets. Furthermore, in terms of accuracy, we observe that for the diabetes dataset, our algorithm reaches a higher accuracy value faster than the alternatives in both validation and test sets.

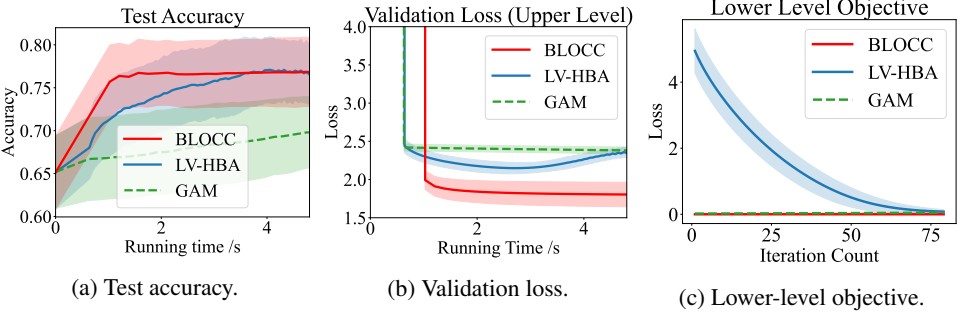

(a) Test accuracy.      (b) Validation loss.      (c) Lower-level objective.

Figure 4: Plots for diabetes dataset

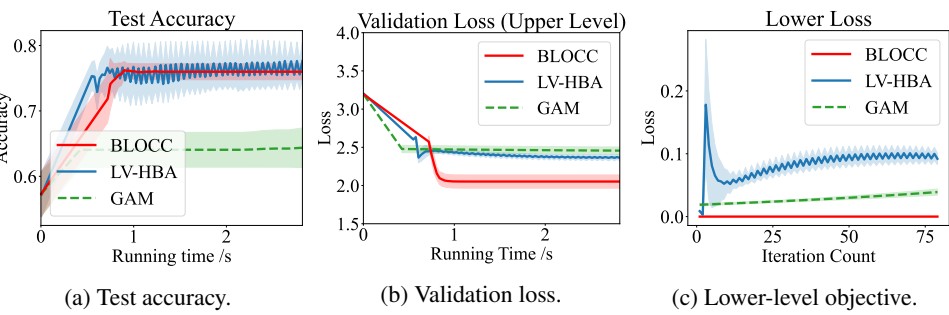

(a) Test accuracy.      (b) Validation loss.      (c) Lower-level objective.

Figure 5: Plots for fourclass dataset

# F  Applications to Transportation Network Planning

This section applies the proposed BLOCC algorithm to a transportation network design problem, comparing it with the baselines from [72] and [75].

## F.1  Problem formulation

In transportation network planning, the planning operator is to construct a new transportation network (upper level) connecting a collection of stations $\mathcal{S}$, and the passengers will decide whether to use the new network (lower-level), considering the options given by the new network and existing alternatives (constraints). The goal is to design a network that maximizes the operator's benefit, knowing the passengers will make rational choices depending on the given design.

The operator considers building a new network on a collection of links $\mathcal{A} \subseteq \mathcal{S} \times \mathcal{S}$. For any link $(i,j) \in \mathcal{A}$ connecting stations $i \in \mathcal{S}$ to $j \in \mathcal{S}$, the operator needs to design the capacity $x_{ij}$. If the link's capacity is set to zero, then the link is not constructed. The larger the capacity, the larger the number of travelers, thus the larger the revenue while the higher the construction cost $c_{ij}$.

The passengers are in demand to travel in the network $\mathcal{K} \subseteq \mathcal{S} \times \mathcal{S}$. For every origin-destination pair $(o,d)$, the traffic demand $w^{od}$ and the traffic time on the existing route $t^{od}_{\text{ext}}$ is known. We assume that there is only *one* existing network. The proportion of passengers choosing the new network $y^{od}$ and $y^{od}_{ij}$ the fraction of passengers using link $(i,j)$ to travel from $o$ to $d$ will be determined following passengers' rational choice, modeled by logit choice model [5, 11] that will be explained later. If $y^{od}_{ij} = 0$, then the link does not belong to the route that passengers follow to travel from $o$ to $d$.

In a) of Figure 6, the station $\mathcal{S}$ are presented as the dots, and links $\mathcal{A}$ are as the dashed lines (input topology); b) of Figure 6 shows a heatmap for the demand matrix for each $(o,d) \in \mathcal{K}$ (input to the design); c) is an example of the constructed network connecting some of the stations (output of the design); and d) illustrates the number of passengers using the constructed network (design output).

To summarize, in this experiment, we assume that

    **A1)** Only one existing alternative network exists;

    **A2)** Passengers decide whether to use our network rationally, which is modeled by the logit choice model considering a utility depends on travel attributes (travel time) [5, 11]; and,

    **A3)** The demand per market, the trip prices, and the travel times per link are known to operators.

We summarize the optimization variables as follows

- $x_{ij} \in \mathbb{R}_+$, the capacity constructed for the link $(i,j) \in \mathcal{A}$. The number of capacity variables is $|\mathcal{A}|$. The link is not constructed if $x_{ij} = 0$.

- $y^{od} \in [0,1]$, the fraction (proportion) of passengers from market $(o,d) \in \mathcal{K}$ choosing the new network for their travel. The number of flow variables is $|\mathcal{K}|$. Since we consider only one competitor, $1 - y^{od}$ represents the fraction of incumbent network passengers. If $y^{od} = 0$, then the passengers of market $(o,d)$ do not use the new network.

- $y^{od}_{ij} \in [0,1]$, the fraction of passengers from market $(o,d) \in \mathcal{K}$ that, when choosing the new network to travel from $o$ to $d$, use the link $(i,j) \in \mathcal{A}$ in their route to the destination. The number of link flow variables is $|\mathcal{A}||\mathcal{K}|$. With this definition, it holds that $y^{od}_{ij} \leq y^{od}$; and, if $y^{od}_{ij} = 0$, then link $(i,j)$ does not belong to the route when traveling from $o$ to $d$.

To simplify the notation and to make it consistent with the notations used in the paper, we denote

- $x = \{x_{ij}\}_{\forall(i,j)\in\mathcal{A}}$ represents the upper-level variables to be optimized.

- $\mathcal{X} = \mathbb{R}^{|\mathcal{A}|}_+$ represents the domain of $x$.

- $y = \{y^{od}, \{y^{od}_{ij}\}_{\forall(i,j)\in\mathcal{A}}\}_{\forall(o,d)\in\mathcal{K}}$ represents the lower-level variables to be optimized

- $\mathcal{Y} = [\varepsilon, 1-\varepsilon]^{|\mathcal{K}|} \times [\varepsilon, 1-\varepsilon]^{|\mathcal{A}||\mathcal{K}|}$, where $\varepsilon$ is a small positive number set by the designer of the network, represents the domain of $y$.

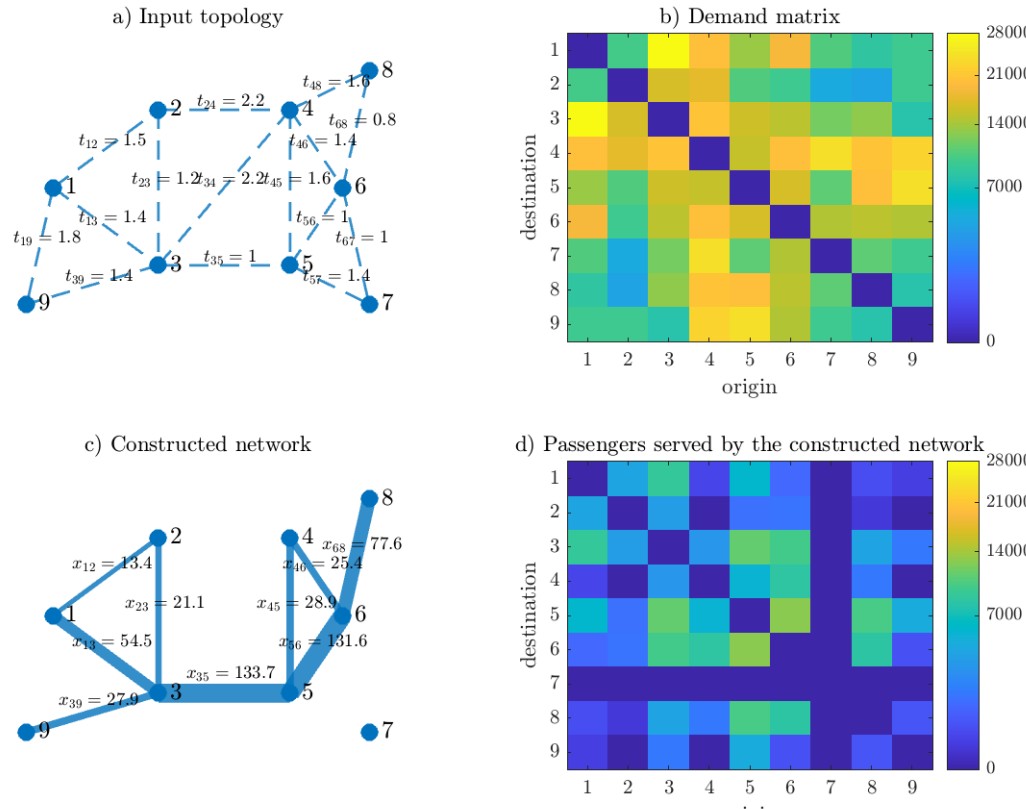

Figure 6: Example of a transportation network design. (a) represents the set of stations $\mathcal{S} = \{1, 2, 3, 4, 5, 6, 7, 8, 9\}$ and the set of links $\mathcal{A} \subseteq \mathcal{S} \times \mathcal{S}$, where the number of links is $|\mathcal{A}| = 30$ (15 segments with two orientations each) and the value on each edge represents the travel time. (b) represents the demand $\{w^{od}\}$ between all $(o, d)$ pairs, where $|\mathcal{K}| = 9 \times 8 = 72$ values are provided (those in the off-diagonal elements of the heatmap). (c) represents the constructed network, where, to facilitate visualization, we have assumed that the capacity is symmetric and used the width of the edge to represent the capacity of every link. (d) represents $\{w^{od}y^{od}\}_{\{\forall(o,d)\in\mathcal{K}\}}$, the number of passengers served by the constructed network.

Besides the optimization variables, our objective and constraints include the following parameters

- $w^{od}$, the total estimated demand (number of passengers) for the market $(o, d) \in \mathcal{K}$.
- $m^{od}$, the revenue obtained by the operator from a passenger in the market $(o, d) \in \mathcal{K}$.
- $c_{ij}$, the construction cost per passenger associated with link $(i, j) \in \mathcal{A}$.
- $t_{ij}$, the travel time for link $(i, j) \in \mathcal{A}$.
- $t_{\text{ext}}^{od}$, travel time on the alternative network for passengers in the market $(o, d) \in \mathcal{K}$.
- $\omega_t < 0$, the coefficient associated with the travel time in passengers' utility function.

In transportation network design, a bilevel formulation is essential due to the interaction between two players with different levels of influence: the operator, who constructs the network, and the passengers, who choose their routes based on the network's characteristics. The operator's goal is to maximize their benefit by minimizing construction costs and maximizing attracted demand, with link capacity as the optimization variable. Conversely, passengers aim to maximize their trip utility, determining the proportion of demand using each link. This dual optimization requires that passenger choices comply with link capacity constraints set by the operator, coupling the variables at both levels. This necessitates a bilevel optimization approach, specifically using BLOCC, to address the interdependent decisions and constraints effectively.

Now, we are ready to introduce the objective formulations of our BLO problem. For the upper level, the network operator aims to maximize profits and minimize costs, and therefore, its interest is

$$\min_{x \in \mathcal{X}} f(x, y_g^*(x)) := -\left( \underbrace{\sum_{\forall (o,d) \in \mathcal{K}} m^{od} y^{od*}(x)}_{\text{profit}} - \underbrace{\sum_{\forall (i,j) \in \mathcal{A}} c_{ij} x_{ij}}_{\text{cost}} \right), \tag{64}$$

where $y^{od*}(x)$ are optimal lower-level passenger flows associated with the network design $x$.

For the lower-level, we model the passenger's behavior by finding the flow variables that maximize utility and minimizes flow entropy cost.

$$\min_{y \in \mathcal{Y}} g(x, y) := -\left( \underbrace{\sum_{(o,d) \in \mathcal{K}} \sum_{(i,j) \in \mathcal{A}} w^{od} \omega_t t_{ij} y_{ij}^{od} + \sum_{(o,d) \in \mathcal{K}} w^{od} \omega_t t_{\text{ext}}^{od} (1 - y^{od})}_{\text{passengers utility}} \right. \tag{65}$$

$$\left. + \underbrace{\sum_{(o,d) \in \mathcal{K}} w^{od} y^{od} (\ln(y^{od}) - 1) + \sum_{(o,d) \in \mathcal{K}} w^{od} (1 - y^{od})(\ln(1 - y^{od}) - 1)}_{\text{flow entropy cost}} \right).$$

The passengers' utility considers the time cost of choosing the new network and the existing network. The users set the flow variables $y$ so that the transportation network yielding a higher utility is preferred. Here, the probability of chosing new network is modeled by a logistic (softmax) model. However, setting the objective as a simple linear utility maximization would lead to an all-or-nothing policy, which is not the behavior observed in practice. Hence, the second term is brought to consider. The approach considered here is to formulate an objective given by the Legendre transform (cf. Definition 5) of the softmax sharing (see [54, 57, 56] for additional details). Intuitively, this means that rather than imposing the softmax sharing a fortiori, we formulate a *convex* problem whose KKT conditions lead to the softmax sharing. Using the fact that the Legendre transform of an exponential $e^y$ is the negative entropy function $y(\ln(y) - 1)$, the lower-level objective is traditional as in [57].

Having introduced the optimization variables, parameters, and objective functions, we next formulate our BLO problem, where we also incorporate the network constraints:

$$\min_{x \in \mathcal{X}} \quad - \sum_{\forall (o,d) \in \mathcal{K}} m^{od} y^{od*} + \sum_{\forall (i,j) \in \mathcal{A}} c_{ij} x_{ij} \tag{66a}$$

$$\text{s.t. } (y^{od*}, y_{ij}^{od*}) = \arg\min_{y \in \mathcal{Y}} - \sum_{(o,d) \in \mathcal{K}} \sum_{(i,j) \in \mathcal{A}} w^{od} \omega_t t_{ij} y_{ij}^{od} - \sum_{(o,d) \in \mathcal{K}} w^{od} \omega_t t_{\text{ext}}^{od} (1 - y^{od}) \tag{66b}$$

$$+ \sum_{(o,d) \in \mathcal{K}} w^{od} y^{od} (\ln(y^{od}) - 1) + \sum_{(o,d) \in \mathcal{K}} w^{od} (1 - y^{od})(\ln(1 - y^{od}) - 1)$$

$$\text{s.t. } \sum_{\forall j | (i,j) \in \mathcal{A}} y_{ij}^{od} - \sum_{\forall j | (j,i) \in \mathcal{A}} y_{ji}^{od} = \begin{cases} y^{od} & \text{if } i = o \\ -y^{od} & \text{if } i = d \\ 0 & \text{otherwise} \end{cases} \quad \forall i, (o,d) \in \mathcal{S} \times \mathcal{K} \tag{66c}$$

$$\sum_{\forall (o,d) \in \mathcal{K}} w^{od} y_{ij}^{od} \leq x_{ij} \quad \forall (i,j) \in \mathcal{A} \tag{66d}$$

where (66c) are the flow-conservation constraints, (66d) are the capacity constraints that involve both upper and lower-level variables, coupling the optimization and motivating the use of BLOCC. Note that, for networks with $n$ stations (nodes): i) the number of CCs is approximately $n^2$; and ii) each of the constraints involves approximately $n^2$ variables. Hence, even for a moderate-size network (say 30-50 nodes), we may have thousands of CCs involving millions of variables.

**Experiment roadmap.** To provide numerical results illustrating the behavior of our algorithm, we solve the optimization in (66) for three scenarios:

**S1)** The design of a 3-node simple synthetic network;

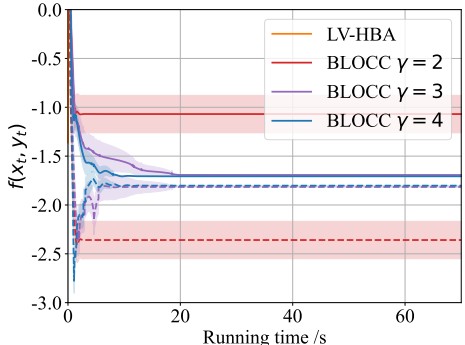 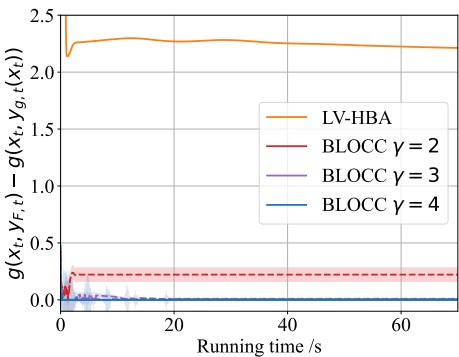

Figure 7: Upper-level objective $f(x_t, y_t)$ for a 3-node network design problem; Solid lines show mean value of $f(x_t, y_{g,t}^{T_g})$ with the shaded region as a standard deviation; Dashed lines show the mean value of $f(x_t, y_{F,t}^{T_F})$ with the shaded region as a standard deviation; Three different $\gamma$ values (red, purple, blue) and fixed stepsize $\eta = 1.6 \times 10^{-4}$; The orange color represents the result of the LV-HBA algorithm.

Figure 8: Optimality gap of the lower-level problem for a 3-node network design problem $g(x_t, y_t) - g(x_t, y_t^*)$. Solid lines represent the mean value of the 10 realizations of the upper-level variables, dashed lines represent $g(x_t, y_{F,t}^{T_F}) - g(x_t, y_t^*)$, and the shaded region is the standard deviation. Three different $\gamma$ values are represented in our algorithm, and fixed stepsize $\eta = 1.6 \times 10^{-4}$.

**S2)** The design of a 9-node synthetic network from the prior transportation literature; and

**S3)** The design of a (real-world) subway network for the city of Seville, Spain, with 24 nodes

where **S1)** involves 6 CCs and 48 variables, and **S3)** involves around 100 CCs and 50,000 variables.

For the 3-node network scenario, we will conduct a comparative analysis against other algorithms to evaluate the efficacy of our approach. In the other two scenarios, the baseline algorithms cannot find a solution; hence, for the 9-node and Seville networks, we will focus on providing insights into the performance and behavior of our algorithm under varying parameters, shedding light on the versatility and adaptability of our approach to real-world transportation networks.

Before delving into the presentation and analysis of the results, two additional remarks are in order:

**R1)** While one of the goals of these experiments was to compare our BLOCC algorithm against LV-HBA [75] and GAM [72], for the scenario at hand, the GAM algorithm cannot be implemented, since the inverse of a matrix at each iteration for the problem in (66) is not tractable. In this way, we only conducted the experiments using our BLOCC and LV-HBA.

**R2)** The BLOCC algorithm produces two lower-level variables: $y_{F,T}^{T_F}$ and $y_{g,T}^{T_g}$. While Theorems 1 and 2 guarantee that $(x_T, y_{F,T}^{T_F})$ is an $\epsilon$-approximate solution, the pair $(x_T, y_{g,T}^{T_g})$ is strictly feasible, meaning that $y_{g,T}^{T_g} = y_g^*(x_T)$. Since strict feasibility is important in the transportation network, this section will report the results using both $y_{F,T}^{T_F}$ and $y_{g,T}^{T_g}$.

### F.2 Numerical results for the 3-node network

In this section, we address the problem defined in equation (66) for a network comprising 3 nodes (stations). We assume that the graph of potential links $\mathcal{A}$ is complete, leading to a total of 6 link capacities that need to be determined in the upper level. Additionally, as in the rest of the manuscript, we assume that there is demand for all markets, meaning that the set $\mathcal{K}$ is complete and includes all 6 possible origin-destination pairs. The specific values of the key parameters can be found in Table 4, with further details about the simulated scenario available in the online code repository.

For BLOCC, we set the stepsize to $\eta = 1.6 \times 10^{-4}$ and analyze the algorithm's convergence for three different values of $\gamma$: $\gamma = 2$, $\gamma = 3$, and $\gamma = 4$. The upper-level objective values are computed using $f(x_t, y_{g,t}^{T_g})$ and $f(x_t, y_{F,t}^{T_F})$. The results are presented in Figures 7 and 8.

| Parameter | Market $(o,d)$ | | | | | |
|---|---|---|---|---|---|---|
| | (1,2) | (1,3) | (2,1) | (2,3) | (3,1) | (3,2) |
| demand $w^{od}$ | 1 | 1 | 1 | 1 | 1 | 1 |
| revenue $m^{od}$ | 2 | 6 | 2 | 1 | 6 | 1 |
| travel time incumbent $t_{\text{ext}}^{od}$ | 3 | 3 | 3 | 3 | 3 | 3 |

| Parameter | Link $(i,j)$ | | | | | |
|---|---|---|---|---|---|---|
| | (1,2) | (1,3) | (2,1) | (2,2) | (3,1) | (3,2) |
| link construction cost $c_{ij}$ | 1 | 10 | 1 | 3 | 10 | 3 |
| link travel time $t_{ij}$ | 1 | 10 | 1 | 2 | 10 | 2 |

Table 4: Value of the parameters for scenario 1 (3-node network). The value of $\omega_t$ is set to 0.1.

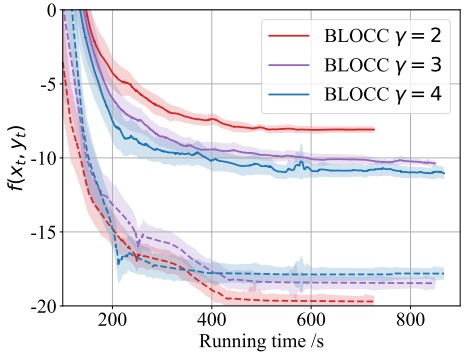

Figure 9: Upper-level objective $f(x_t, y_t)$ for a 9-node network design problem; Solid lines show the mean value of $f(x_t, y_{g,t}^{T_g})$ with the shaded region as a standard deviation; Dashed lines show the mean value of $f(x_t, y_{F,t}^{T_F})$ with the shaded region as a standard deviation; Three different $\gamma$ values (red, purple, blue) and fixed stepsize $\eta = 1.6 \times 10^{-4}$; The orange color represents the result of the LV-HBA algorithm.

Figure 10: Upper-level objective $f(x_t, y_t)$ for a metro network design problem in Seville, Spain, for 2 random initializations of the upper-level variables. Solid lines represent the mean of $f(x_t, y_{g,t}^{T_g})$, and the shaded region is the standard deviation. The dashed lines represent the mean of $f(x_t, y_{F,t}^{T_F})$, and the shaded region is the standard deviation. Three different $\gamma$ values are tested with a fixed stepsize $\eta = 1.6 \times 10^{-4}$.

Figure 7 illustrates the performance of our BLOCC algorithm over time. The orange line represents the evolution of $f(x_t, y_t)$ for the LV-HBA algorithm, while the other six lines represent different implementations of BLOCC. Each color represents a different value of $\gamma$; solid lines represent $f(x_t, y_{g,t}^{T_g})$ and dashed lines represent $f(x_t, y_{F,t}^{T_F})$. Each simulation is conducted 10 times with 10 different random initializations of the upper-level variables, and both the mean and the standard deviation values are displayed. The results indicate that all versions of our BLOCC algorithm converge in less than 10 seconds, while LV-HBA shows slight fluctuations even after running for more than 50 seconds. This may be attributed to the fact that the LV-HBA algorithm requires a joint projection into $\{\mathcal{X} \times \mathcal{Y} : g^c(x,y) \leq 0\}$ at each iteration, involving 42 variables and 6 CCs.

Figure 8 shows the lower-level optimality gap, namely $g(x_t, y_t) - g(x_t, y^*(x_t))$ for LV-HBA and BLOCC with 3 different values of $\gamma$. In the case of LV-HBA, lower-level optimality is not attained within 70 seconds of running time. Conversely, for BLOCC, lower-level optimality is achieved by construction when $y_t$ is set to $y_{g,T}^{T_g}$, resulting in a zero gap. Additionally, when $y_t$ is set to $y_{F,T}^{T_F}$, we observe that: i) lower-level optimality is accomplished for $\gamma \geq 3$, and ii) when $\gamma = 2$, an optimality gap exists, but it is one order of magnitude smaller than that for LV-HBA.

### F.3 Numerical results for the 9-node network

In this case, we consider the network in [21], see also Figure 6, which has $|\mathcal{S}| = 9$ nodes and $|\mathcal{A}| = 30$ potential links. As before, we consider that all markets exist, so that $|\mathcal{K}| = 9 \cdot 8 = 72$. The remaining parameters are described in Figure 6 and the code repository.

Figure 9 is the counterpart of Figure 7 for the 9-node scenario, showing the behavior of our BLOCC algorithm for $\eta = 1.6 \times 10^{-4}$ and $\gamma \in \{2, 3, 4\}$. Each simulation is repeated 10 times (using 10 different random initializations of the upper-level variables), and both the mean and the standard deviation values are shown. Since the number of variables and constraints is almost one order of magnitude larger, the algorithm requires more time to converge. However, convergence takes place in a reasonable amount of time (20-40 times longer than in the previous 3-node test case). Regarding the optimal value, we observe that: i) the sensitivity of the (steady-state) optimal value with respect to $\gamma$ is not too large; ii) the solutions based on $y_{F,T}^{T_F}$ yield better upper-level values than those based on $y_{g,T}^{T_g}$; and iii) the gap between $f(x_T, y_{g,T}^{T_g})$ and $f(x_T, y_{F,T}^{T_F})$ decreases as $\gamma$ increases. Observation ii) is due to the fact that $y_{F,T}^{T_F}$ is not feasible (meaning that it violates the optimality of the lower-level); hence, it is able to achieve a better upper-level objective. In addition, the behavior observed in iii) is consistent with the discussion in Section 2.2, with the value of $\gamma$ having an impact on the suboptimality of $y_{F,t}^{T_F}$ at the lower-level. Specifically, higher values of $\gamma$ push $y_{F,t}^{T_F}$ closer to $y_{g,t}^{T_g}$ and, hence, decrease the lower-level optimality gap. Finally, we must note that for the solid lines (associated with $y_{g,t}^{T_g}$), it holds that $f(x_t, y_{g,t}^{T_g}) = f(x_t, y^*(x_t))$. This implies that if we need a solution that is feasible at the lower-level, then better objective values are associated with higher values of $\gamma$.

### F.4 Numerical results for the Seville network

In this section, we demonstrate the practical use of BLOCC in a real transportation network design problem. Specifically, we address the design of a metro network in the city of Seville, which has approximately one million inhabitants and is located in the south of Spain, with the bus system as its competitor. The data for the demand, number of stations, and locations have been taken from [21]. Information about the construction costs, capacity, and travel time has been obtained from Spanish rapid transit operators (see references in [10, 56] for full details). The city authorities considered $|\mathcal{S}| = 24$ potential station locations. Regarding the links, the following assumptions are made:

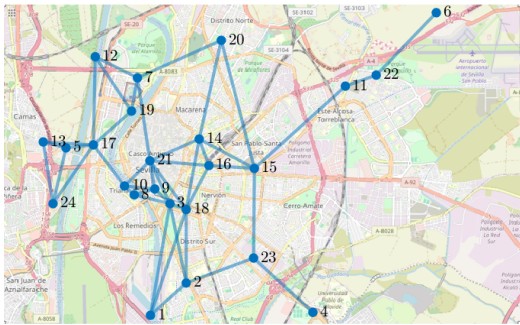

Figure 11: Topology of the Seville network.

**A1)** The link between nodes $(i, j) \in \mathcal{S} \times \mathcal{S}$ only exists if node $j$ is one of the three closest neighbors to $i$, or vice versa. The distance here is measured in terms of travel time. This assumption limits the number of lines in any station to be at most 3, which is a very mild assumption for a network with 24 stations.

**A2)** The link between nodes $(i, j) \in \mathcal{S} \times \mathcal{S}$ only exists if the travel time $t_{ij}$ is less than 7 minutes. This is also a mild assumption, since it enables all the stations to be connected, including those that are further away from the city center (the airport and the university campus [21]).

Under these two conditions, the set $\mathcal{A}$ of potential links contains $|\mathcal{A}| = 88$ links. The sets of links and stations, along with their actual locations, are shown in Figure 11. Finally, we consider all possible markets between nodes, so $|\mathcal{K}| = 24 \times 23 = 552$.

Following the narrative in Sections F.2 and F.3, Figure 10 presents the evolution of the upper-level objective function with time for three values of the parameter $\gamma \in \{2, 3, 4\}$. The stepsize value has been set to $\eta = 1.6 \times 10^{-4}$, and two different initializations have been considered. Regarding the behavior of the algorithm with respect to the value of $\gamma$ and the particular output chosen ($y_{T,g}^{T_g}$ vs. $y_{T,F}^{T_F}$), the findings are very similar to those in Figure 9. Namely, smaller gaps are found for larger values of $\gamma$, and if feasibility at the lower-level (i.e., consistency with the user preference level) must be preserved, better objective values are achieved for larger values of $\gamma$.

The most important observation, however, is related to the running time. Specifically, Figure 10 reveals that, for this real-world scenario, our BLOCC algorithm converges in 10-20 hours. While this is more than 1,000 times larger than the convergence interval for the 3-node network, the number

of variables and constraints here is 100 times larger. More importantly, in the context of network transportation design, optimization times of 100 hours are widely accepted even for single-level formulations. Overall, we believe that the numerical results demonstrate that the BLOCC algorithm proposed in this work can solve problems with a large number of variables and CCs, which can have practical value in real-world applications, such as the one studied in this section.

# G  Sensitivity Analysis

Regarding the selection and impact of hyper-parameters on the performance of BLOCC, we conducted an ablation study on various values of the two critical parameters, $\gamma$ and $\eta$, and measured their effects on the optimal value and computational time. In this section, we present the sensitivity analysis on the toy example that we introduced in Section 4.1 and the 3-node network example for the transportation network planning problem in Section 4.3.

## G.1  Sensitivity analysis for the toy example

We present in Table 5 the results of lower-level optimality $\|y_g^*(x_T) - y_{F,T}\|$ using different $\gamma$ and $\eta$ to conduct BLOCC (Algorithm 1) on the toy example in Section 4.1.

| | $\|y_g^*(x_T) - y_{F,T}\|$ | | | |
|---|---|---|---|---|
| $\eta$ | $\gamma = 0.001$ | $\gamma = 0.01$ | $\gamma = 0.1$ | $\gamma = 1.0$ |
| 0.001 | $0.028 \pm 0.042$ | $0.035 \pm 0.076$ | $0.027 \pm 0.064$ | $0.000 \pm 0.000$ |
| | $(3.314 \pm 0.042)$ | $(3.186 \pm 0.076)$ | $(2.049 \pm 0.064)$ | $(1.967 \pm 0.000)$ |
| 0.01 | $0.020 \pm 0.031$ | $0.020 \pm 0.041$ | $0.009 \pm 0.019$ | $0.000 \pm 0.000$ |
| | $(2.242 \pm 0.031)$ | $(1.575 \pm 0.041)$ | $(1.118 \pm 0.019)$ | $(0.585 \pm 0.000)$ |
| 0.1 | $0.006 \pm 0.010$ | $0.017 \pm 0.027$ | $0.011 \pm 0.033$ | $0.000 \pm 0.000$ |
| | $(1.616 \pm 0.010)$ | $(1.475 \pm 0.027)$ | $(1.257 \pm 0.033)$ | $(0.383 \pm 0.000)$ |
| 1.0 | $0.023 \pm 0.019$ | $0.030 \pm 0.022$ | $0.020 \pm 0.045$ | $0.000 \pm 0.000$ |
| | $(7.118 \pm 0.019)$ | $(4.570 \pm 0.022)$ | $(3.477 \pm 0.045)$ | $(1.645 \pm 0.000)$ |
| 10.0 | $0.034 \pm 0.123$ | $0.019 \pm 0.014$ | $0.025 \pm 0.070$ | $0.000 \pm 0.000$ |
| | $(6.565 \pm 0.123)$ | $(4.365 \pm 0.014)$ | $(2.808 \pm 0.070)$ | $(1.750 \pm 0.000)$ |

Table 5: Sensitivity analysis for the hyperparameters in Section 4.1. Top line in each cell represents the optimality gap $\|y_g^*(x_T) - y_{F,T}\|$, while the bottom line represents the time required for the algorithm to converge. Both the mean and the standard deviation are for 40 simulations.

From the table, we can draw the following empirical observations:

**O1)** *Larger values of $\gamma$ bring the upper-level objectives closer to optimal.* This is consistent with Theorem 1 which illustrated that larger $\gamma$ improves the accuracy of the lower-level optimality. Since the obtained solution $y_{F,T}$ is closer to $y_g^*(x_T)$ for larger $\gamma$ values, the distance between the $f(x_T, y_{F,T})$ and $f(x_T, y_g^*(x_T))$ will be closer as well.

**O2)** *For a sufficiently small fixed $\eta$, larger $\gamma$ lead to faster convergence.* This implies that *smaller $\eta \leq \frac{1}{l_{F,1}}$ choice due to larger $\gamma$ will not significantly dampen the convergence time.* This is because a large value of $\gamma$ increases $l_{F,1}$, sharpening the function $F_\gamma(x)$ and its gradient will be larger for most points. Thus, the gradient update $\eta \nabla F_\gamma(x_t)$ will not be very small and thus won't make the convergence slower.

## G.2  Sensitivity Analysis for the 3-node network

We present in Table 6 the results of the upper-level objective value $f(x, y)$ in network planning problem for a 3-node network, whose detailed framework is introduced in Section F.

| $\eta$ | $\gamma = 1$ | $\gamma = 2$ | $\gamma = 3$ | $\gamma = 4$ | $\gamma = 5$ | $f(x,y)$ $\gamma = 6$ | $\gamma = 7$ | $\gamma = 8$ | $\gamma = 9$ | $\gamma = 10$ |
|---|---|---|---|---|---|---|---|---|---|---|
| $1.6 \times 10^{-4}$ | $-0.9625 \pm 0.76$ $13.99 \pm 6.32$ | $-1.4274 \pm 0.28$ $5.43 \pm 1.94$ | $-1.6081 \pm 0.27$ $7.85 \pm 2.92$ | $-1.4507 \pm 0.41$ $13.02 \pm 4.47$ | $0.0000 \pm 0.00$ $5.85 \pm 1.77$ | $0.0000 \pm 0.00$ $4.51 \pm 1.72$ | $0.0000 \pm 0.00$ $3.06 \pm 0.97$ | $0.0000 \pm 0.00$ $2.71 \pm 0.95$ | $0.0000 \pm 0.00$ $2.18 \pm 0.81$ | $0.0000 \pm 0.00$ $1.77 \pm 0.68$ |
| $3.2 \times 10^{-5}$ | $-0.9649 \pm 0.76$ $51.85 \pm 12.12$ | $-1.4768 \pm 0.22$ $15.61 \pm 6.33$ | $-1.6081 \pm 0.27$ $22.56 \pm 6.29$ | $-1.6214 \pm 0.27$ $20.83 \pm 8.59$ | $-1.4582 \pm 0.41$ $22.48 \pm 7.89$ | $-1.3772 \pm 0.44$ $29.13 \pm 10.19$ | $-1.1278 \pm 0.69$ $0.56 \pm 0.31$ | $0.0000 \pm 0.00$ $14.38 \pm 3.63$ | $0.0000 \pm 0.00$ $7.59 \pm 2.84$ | $0.0000 \pm 0.00$ $6.05 \pm 2.12$ |
| $1.6 \times 10^{-5}$ | $-1.0461 \pm 0.66$ $21.26 \pm 12.85$ | $-1.4744 \pm 0.22$ $30.82 \pm 8.59$ | $-1.6081 \pm 0.27$ $46.02 \pm 9.89$ | $-1.6214 \pm 0.27$ $35.98 \pm 11.00$ | $-1.4582 \pm 0.41$ $32.53 \pm 9.90$ | $-1.3772 \pm 0.44$ $32.37 \pm 11.80$ | $-1.3808 \pm 0.45$ $40.79 \pm 12.98$ | $-1.2106 \pm 0.73$ $40.31 \pm 13.95$ | $-0.6792 \pm 0.59$ $1.47 \pm 0.57$ | $0.0000 \pm 0.00$ $17.27 \pm 4.27$ |
| $3.2 \times 10^{-6}$ | $-0.7819 \pm 0.68$ $888.38 \pm 256.38$ | $-1.4725 \pm 0.22$ $213.22 \pm 33.88$ | $-1.6081 \pm 0.27$ $335.63 \pm 43.42$ | $-1.6214 \pm 0.27$ $253.26 \pm 46.92$ | $-1.6297 \pm 0.27$ $210.03 \pm 44.41$ | $-1.6355 \pm 0.27$ $206.14 \pm 48.21$ | $-1.4671 \pm 0.42$ $172.50 \pm 43.31$ | $-1.4700 \pm 0.42$ $153.57 \pm 43.90$ | $-1.3856 \pm 0.45$ $138.12 \pm 38.92$ | $-1.3873 \pm 0.45$ $117.68 \pm 34.65$ |
| $1.6 \times 10^{-6}$ | $-0.9665 \pm 0.63$ $805.74 \pm 268.20$ | $-1.4723 \pm 0.23$ $377.37 \pm 54.25$ | $-1.6840 \pm 0.03$ $629.90 \pm 65.16$ | $-1.6214 \pm 0.27$ $496.31 \pm 69.79$ | $-1.6297 \pm 0.27$ $464.33 \pm 68.94$ | $-1.6355 \pm 0.27$ $418.23 \pm 67.45$ | $-1.6397 \pm 0.27$ $390.74 \pm 65.78$ | $-1.4700 \pm 0.42$ $356.06 \pm 60.70$ | $-1.4722 \pm 0.42$ $334.36 \pm 61.03$ | $-1.4741 \pm 0.42$ $305.70 \pm 57.01$ |

Table 6: Sensitivity analysis for hyperparameters $\eta$ and $\gamma$ in the experiment of the 3-node network transportation design described in Section 4.3 and Appendix F. Top line in each cell represents the upper-level objective value $f(x_T, y_{g,T})$, while the bottom line represents convergence time. Both the mean and the standard deviation of 10 simulations are provided.

We know from Section F that the goal is to minimize $f(x, y)$, and negative values of $f(x, y)$ are expected. Therefore, $f(x_T, y_T) = 0$ in the table indicates a failure to converge and we can see that

**O3)** *Smaller $\eta$ is needed to satisfy $\eta \leq \frac{1}{l_{F,1}}$ to ensure convergence if $\gamma$ is chosen larger.* The algorithm tends to converge for different $\gamma$ values when $\eta$ is small enough. This is because the Lipschitz smoothness constant $l_{F,1}$ of $F_\gamma(x)$ increases with $\gamma$ (Lemma 3), and the BLOCC algorithm is guaranteed to converge if $\eta \leq \frac{1}{l_{F,1}}$ (Theorem 2).

In summary, the experimental results align with the theoretical analysis, demonstrating that the penalty constant $\gamma$ is robust and can be effectively set around 10, and the stepsize should be adjusted to ensure smooth and monotonic convergence.

## H   Analysis of the Computational Complexity

### H.1   Complexity comparison

We present in the following the computational complexity of our BLOCC algorithm in comparison with LV-HBA [75] and GAM [72] as baselines.

| Method | Iteration Cost | Computational Cost per Iteration |
|---|---|---|
| **BLOCC** | Outer loop (Algorithm 1): $T = \mathcal{O}(\epsilon^{-1.5})$ | **General setting (Theorem 3)**: $\tilde{\mathcal{O}}(d_c d_x + d_x^2 + \epsilon^{-1}(d_c d_y + d_y^2))$; **Special case (Theorem 4)**: $\tilde{\mathcal{O}}(d_x^2 + d_y^2 + d_c(d_x + d_y))$ |
| **LV-HBA** | $\mathcal{O}(\epsilon^{-3})$ | $\mathcal{O}(d_x d_y d_c + (d_x + d_y)^{3.5})$ |
| **GAM** | Asymptotic | More than $\mathcal{O}(d_y^3 + d_x^2 + d_c(d_x + d_y))$ |

Table 7: Complexity comparison of our work with LV-HBA [75] and GAM [72].

The iteration costs are detailed in the earlier sections, and in [75] and [72]. Hence, we discuss next the per-iteration cost. In the following discussion, we assume

**A1)** *the complexity of calculating a function is proportional to the dimension of the inputs.* For example, The complexity of finding $\nabla_x g(x,y), \nabla_x f(x,y)$ are $\mathcal{O}(d_x)$, and the one for $g^c(x,y)\rangle$ is $\mathcal{O}(d_x d_y)$.

**A2)** *Projection cost on $\mathcal{X}$ and $\mathcal{Y}$ is respectively no more than $\mathcal{O}(d_x^2)$ and $\mathcal{O}(d_y^2)$.* As discussed in the introduction, $\mathcal{X}$ and $\mathcal{Y}$ are assumed to be easy-to-project domains, i.e. projection can be done using a projection matrix or a simple formula, and the projection costs are no more than $\mathcal{O}(d_x^2)$ and $\mathcal{O}(d_y^2)$, respectively.

In this way, **BLOCC** is a first-order method with gradient calculation costs of $\mathcal{O}(d_x d_c)$ and $\mathcal{O}(d_y d_c)$, and the projection cost of $\mathcal{O}(d_x^2)$ and $\mathcal{O}(d_y^2)$. We present the detailed analysis of achieving the computational cost in the table in the following Section H.2.

In the SVM model training and network planning experiments in Section 4, projections are simpler (e.g., truncation), resulting in $\mathcal{O}(d_x)$ and $\mathcal{O}(d_y)$ costs. In this scenario, for the general setting (Theorem 3), the complexity is $\tilde{\mathcal{O}}(\epsilon^{-1.5} d_c d_x + \epsilon^{-2.5} d_c d_y)$; and for the $g^c$ affine in $y$ setting (Theorem 4), it is $\tilde{\mathcal{O}}(\epsilon^{-1.5} d_c (d_x + d_y))$. Therefore, our BLOCC is especially robust to large-scale problems.

**LV-HBA** [75], also a first-order method, has similar gradient calculation costs. However, its projection onto $\{\mathcal{X} \times \mathcal{Y} : g^c(x,y) \leq 0\}$ is expensive, with a complexity of $\mathcal{O}((d_x + d_y)^{3.5})$ of using interior point method to find the projected point [38], and evaluating $g^c(x,y) \leq 0$ adds $\mathcal{O}(d_x d_y d_c)$ as it requires $d_c$ inequality judgment on functions taking input dimension $d_x, d_y$.

**GAM** [72] lacks an explicit algorithm for lower-level optimality and Lagrange multipliers. Even if we omit this, calculating $\nabla^2_{yy} g$ and its inverse incurs a cost of $\mathcal{O}(d_y^3)$. Additionally, it has cost $\mathcal{O}(d_x^2 + d_c(d_x + d_y))$ for projection onto $\mathcal{X}$ and calculating gradients.

We can see that BLOCC stands out with the lowest computational cost in both iterational and overall complexity thanks to its first-order and joint-projection-free nature. Consequently, BLOCC is particularly well-suited for large-scale applications with high-dimensional parameters.

## H.2 Complexity analysis of BLOCC

At each iteration $t$, our proposed BLOCC algorithm in Algorithm 1 involves:

**Step 1:** Solve two max-min problems.

**Step 2:** Calculate $g_{F,t}$ in (13) and update $x_{t+1} = \text{Proj}_{\mathcal{X}}\left(x_t - \eta g_{F,t}\right)$.

where **Step 1** can be achieved by our proposed max-min solver in Algorithm 2. In the following, we provide analysis for using the accelerated version of the Algorithm 2 in the *general case* (Theorem 3) and the single-loop version for the *special case* (Theorem 4) as discussed in Section 3.3.

In the *general case*, we use the accelerated version of Algorithm 2 to solve both (7) and (5). For solving (7), at each inner loop iteration $t$, line 3 of the algorithm is of computational cost $\mathcal{O}(d_c)$. The update of $y$ in line 4-6 involves calculating $\nabla_y g(x, y)$ with a cost of $\mathcal{O}(d_y)$, finding $\langle \mu, \nabla_y g^c(x, y) \rangle$ for fixed $x$ of $\mathcal{O}(d_y d_c)$ following assumption **A1**), and the projection $\text{Proj}_{\mathcal{Y}}$ of complexity $\mathcal{O}(d_y^2)$ according to **A2**). Moreover, $y$ converges linearly as $L(\mu, y)$ is strongly convex in $y$ (Lemma 7), this inner update for $y$ gives an iteration complexity of $\mathcal{O}(\ln(\epsilon^{-1}))$. Therefore, the cost of the update for $y$ totals up to $\mathcal{O}(\ln(\epsilon^{-1})(d_y d_c + d_y^2))$. The update of $\mu$ in line 7 involves $\mathcal{O}(d_y d_c)$ for calculating $\nabla_\mu L(\mu_{t+1/2}, y_{t+1}) = g^c(x, y_{t+1})$ with $x$ being fixed, and $\mathcal{O}(d_c)$ for projection onto $\mathbb{R}^{d_c}_+$.

Moreover, to achieve target accuracy $\epsilon$ on the metric $\frac{1}{T}\sum_{t=0}^{T}\|G_\eta(x_t)\|^2 \leq \epsilon$, with $\gamma = \mathcal{O}(\epsilon^{-1})$ using the BLOCC algorithm, the iteration complexity of the max-min solver is $\mathcal{O}(\epsilon^{-1})$ according to Theorem 3. Therefore, the complexity for solving (7) in the *general case* using the accelerated version of Algorithm 2 (Theorem 3) is

$$\mathcal{O}(\epsilon^{-1}(\ln(\epsilon^{-1})(d_y d_c + d_y^2) + d_c d_y + d_c)) = \tilde{\mathcal{O}}(\epsilon^{-1}(d_y d_c + d_y^2)). \tag{67}$$

Solving (5) is of the same order as the only difference is in calculating $\nabla_y f(x, y)$. In this way, we can conclude that **Step 1** is at a cost of $\tilde{\mathcal{O}}(\epsilon^{-1}(d_y d_c + d_y^2))$ in the *general case*.

In the *special case* (Theorem 4), the non-accelerated (single-loop) version of Algorithm 2 involves $y_{t+1} = \text{Proj}_{\mathcal{Y}}(y_t - \eta_1 \nabla_y L(\mu_t, y_t))$ with complexity $\mathcal{O}(d_y^2 + d_y d_c)$, and $\mu_{t+1} = \text{Proj}_{\mathbb{R}^{d_c}_+}(\mu_t + \eta_2 \nabla_\mu L(\mu_t, y_{t+1}))$ with complexity $\mathcal{O}(d_c + d_y d_c)$ following a similar analysis as in the general case.

Moreover, the algorithm converges linearly in the special case of $g^c$ being affine in $y$ and the computational complexity is $\mathcal{O}(\ln(\epsilon^{-1}))$, according to Theorem 4. Therefore, the complexity for the max-min **Step 1** in the *special case* is

$$\mathcal{O}(\ln(\epsilon^{-1})(d_y d_c + d_y^2 + d_c)) = \tilde{\mathcal{O}}(d_y d_c + d_y^2 + d_c). \tag{68}$$

**Step 2** involves calculating $\nabla_x g(x, y)$ with with a fixed $y$, and $\langle \mu, \nabla_x g^c(x, y) \rangle$ for fixed $\mu, y$, which are of complexity $\mathcal{O}(d_x)$ and $\mathcal{O}(d_x d_c)$ respectively, and conducting a projection on $\mathcal{X}$ of cost $\mathcal{O}(d_x^2)$ according to assumption **A2**).

In this way, the computational complexity of **Step 2** is

$$\mathcal{O}(d_x d_c + d_x^2). \tag{69}$$

We can therefore conclude the complexity of BLOCC in Table 7.

