# OpenReview forum: "A Primal-Dual-Assisted Penalty Approach to Bilevel Optimization with Coupled Constraints"
_NeurIPS.cc/2024/Conference — NeurIPS 2024 poster_

### Official Review · Reviewer_QYWG · 2024-07-07

**Soundness:** 4
**Presentation:** 4
**Contribution:** 4
**Rating:** 5
**Confidence:** 3

**Summary:**

This works addresses a bilevel problem where the lower-level has constraints that couple both the upper- and lower-level variables. The proposed method leverages a penalty approach and Lagrangian duality theory that transform the original bilevel problem into a single-level minimization problem where the x and y variables are decoupled. A first-order algorithm is developed which relies on the solution of two min-max subproblems and a rigorous convergence theory is developed. The effectiveness of the proposed method is validated on two real-world applications: infrastructure planning in transportation networks and support vector machine model training.

**Strengths:**

* This works deals with a challenging bilevel problem, as not only the lower-level is constrained, but in fact these constraints couple both the upper- and lower-level variables.
* The proposed algorithm is first-order, even though the problem is a challenging one due to the presence of coupled constraints in the lower-level. In contrast, methods relying on implicit gradients, regardless if the problem is constrained or not, usually require access to second-order information.
* Evaluation of the BLOCC algorithm on both toy examples and real-world applications (SVM model training and Transportation network design).

**Weaknesses:**

* Lemmas 2 and 3, which derive some key properties of the inner min-max problems, assume that the Lagrangian multipliers are bounded for every $x \in X$. This is not a common assumption in literature and thus I am not sure why it is seen as mild. The authors cite Theorem 1.6 from [30] as a justification, however it is not immediately clear how the bounded property follows from that theorem. In fact, Theorem 1.6 is written for a more general (abstract) setting. As this is an important point, I think the authors should explain this in more detail. They might even include the theorem (or perhaps a different version of the theorem, tailored to the specific setting of this problem) in the Appendix and provide a clearer explanation.
* Requires the solution of two min-max subproblems where the solution of each subproblem requires a double-loop algorithm. As a result, the proposed BLOCC method is a triple loop algorithm. On the contrary, most hypergradient methods rely only on the solution of a minimization problem (the lower problem).
* The convergence result requires additional assumptions (assumption 5) beyond the ones concerning the objectives ($f,g$) and the constraints ($g_{c}$) of the original bilevel problem. It is unclear how restrictive assumption 5 is, as it is not expressed directly over $f,g,g_c$, but rather over some dual function. On the other hand, at least from the perspective of the dual function, the assumption is mild as the dual function is already concave.

**Questions:**

* Can you elaborate on Assumption 3? What do you mean by saying that $g^{c}$ satisfies the LICQ in $y$? Does this mean that for a given x the LICQ holds: a) for every $y \in Y(x)$ or b) for every KKT point $y^{*}(x)$ of the lower-level problem?
* Do the Assumptions 1-5 and the conditions within the theorems hold in the bilevel problems of the experiments (SVM model training and Transportation network design)?
* Typos
  * In equation (7): the + symbol is misplaced
  * In the proof of Lemma 10 in page 17: in the formula of $\nabla u_{h}(x)$ we should have $\nabla h^{c}$ instead of $h^{c}$

**Limitations:**

No negative societal impact.

---

> ### Author Rebuttal · Authors · 2024-08-07
>
> We thank the reviewer for appreciating our presentation and recognizing our work. Our response to your comments follows.
>
> **Q1. Bounded optimal Lagrange Multiplier assumption for all $x\in\mathcal{X}$.**
>
> We appreciate the reviewer for highlighting this issue. We apologize for the incorrect reference to "Theorem 1.6 in [30]". The accurate reference is Theorem 1 in [65], and we will make this correction. We calrify in the following how how Theorem 1 in [65] supports the boundness of the Lagrange multiplier.
>
> We restate in the following of Theorem 1 in [65].
>
> 1) The set $\Lambda(f)$ is closed and convex for all $f\in \mathcal{F}$.
> 2) The set $\Lambda(f)$ is non-empty for all $f\in \mathcal{F}$ if and only if (GCQ) issatisfied.
> 3) Let (MFCQ) be satisfied. Then, the set $\Lambda(f)$ is compact for all $f\in \mathcal{F}$.
> 4) If there exists $f\in \mathcal{F}$, such that $\Lambda(f)$ is compact, then (MFCQ) is satisfied.
> 5) Let (LICQ) be satisfied. Then, the set $\Lambda(f)$ is a singleton for all $f\in \mathcal{F}$.
>
> where $\mathcal{F}$ is the set of all differentiable function, $\Lambda(f)$ is the set of Lagrange multipliers with respect to $f$.
>
> This theorem states in 3 that MFCQ, a weaker condition than LICQ, is sufficient for the compactness of the Lagrange multiplier. Since we assume LICQ in $y$ holds for all $x \in \mathcal{X}$, we believe the introduction of upper bound parameters $B_g$ and $B_F$ is reasonable.
>
> Thank you again for your careful review and valuable feedback.
>
> **Q2. Due to the max-min subproblems, BLOCC is a triple loop algorithm while most hypergradient methods rely only on the solution of a minimization problem (the lower problem).**
>
> We would like to clarify this from the following two aspects.
>
> *A1. BLOCC is an efficient triple-loop algorithm.* Firstly, having triple loops is not uncommon in solving challenging BLO problems with coupled constraints (CCs) since two of the loops are associated with the fact of BLO dealing with two nested optimization problems and the third one with the the presence of the CCs, see e.g. BVFSM [44] and GAM [67]. GAM [67], a hypergradient method, effectively is of triple-loop as it requires solving for $\mu_g^*(x),y_g^*(x)$ as well. Secondly, a triple-loop algorithm is not necessarily costly, with the key being how many iterations are required in each loop. The loop in $y$ (steps 5-7 in Algorithm 2) has negligible computational complexity of ${\cal O}(\ln(\epsilon^{-1}))$. This is insignificant (approximately ${\cal O}(1)$), as even for $\epsilon=10^{-9}$, $\ln(\epsilon^{-1}) \approx 20$. Thirdly, and more importantly, we provide a detailed computational cost per iteration (see **General Response G1**) that shows that the max-min solvers (Algorithm 2) are highly efficient compared to the baselines.
>
> *A2. Hypergradients-based methods can hardly be applied in this case.* Finding the closed-form hypergradients in the Hypergradients-based methods is always non-trivial. In the literature, they mainly rely on the implicit function theory to derive the closed-form.
> However, the LL domain constraint $y \in \mathcal{Y}$ impedes the use of the implicit function theory as it needs a set of differentiable lower-level optimality conditions (e.g., $\nabla_y g(x,y) + \langle \mu^*_g(x),\nabla_y g^c(x,y)\rangle = 0$ in an unconstrained domain), such as GAM [67] and BVFSM [44]. A detailed analysis of the non-differentiability can be seen in section 2 in [36]. Therefore, we do not use hyper-gradient methods as we are considering $y\in\mathcal{Y}$ and want to construct a fully first-order method using the penalty-based reformulation.
>
> **Q3. It is unclear how restrictive Assumption 5 is.**
>
> Assumption 5 is mild and the mildness is justified for the following reasons:
>
> - The convexity nature of the dual function and the unique solution guarantee via LICQ (lemma 5) imply that the condition $\langle -\nabla D_g(\mu) + \nabla D_g(\mu_g^*(x)), \mu - \mu_g^*(x) \rangle > 0$ holds locally for $\mu \neq \mu_g^*(x)$. This ensures that Assumption 5 is particularly mild.
>
> - We provide a detailed example in Section 3.2.1 that illustrates a specific case being sufficient to this assumption, which further clarifies its applicability.
>
>
> **Q4 The meaning of Assumption 3 of  $g^c$ satisfies the LICQ in $y$.**
>
> Asssumption 3 means that LICQ holds **(b) for every KKT point  $y^*_g(x)$** of the lower-level problem, $\nabla_yg^c(x,y^*_g(x))$ of size $d_c\times d_y$ are linear independent in rows. Thank you for asking this question. We are sorry for the confutsion and we will clarify it.
>
>
> **Q5 Whether the experiments satisfy the assumptions**
> Yes, the experiments satisfy the assumptions.
>
> In the **SVM** problem, the lowL objective (57b) is strongly convex in LL variable $w$. The coupled constraint $1 - l_{{\rm tr}, i}(z_{{\rm tr}, i}^\top w + b) \leq c_i$ is linear and satisfies LICQ, given full row rank training data (**Assumptions 2, 3, & 4** hold). The smooth and Lipschitz conditions for the LL hold, and for the UL,  $c$ is effectively in a compact set since the training data is finite. Thus, Assumption 1 holds. **Assumption 5** is satisfied as the constraint function is affine in $y$, a sufficient condition as shown in Section 3.2.1.
>
> In the **network design** problem, the Lipschitz and smooth conditions hold for (61a), (61b), and (61c) (**Assumption 1**). The LL problem (61b) is strongly convex on $y \in \mathcal{Y}$. Constraint (61d) uncoupled while the coupled constraint (61c) is linear and satisfies LICQ since $w^{od}$ does not have repeating elements in real-world problems, including this experiment (**Assumptions 2, 3, & 4** hold).
>
> **Q6. Typos.**
> *In equation (7): the + symbol is misplaced*
> *In Lemma 10 in page 17: in the formula of  $\nabla v_h(x)$, the second term should be $\nabla_x h^c$  instead of $h^c$*
>
> Thank you for pointing out. We will correct it.
>
> With the above clarifications, we hope the reviewer can kindly reconsider the evaluation of our work!

---

> > ### Comment · Reviewer_QYWG · 2024-08-10
> > **Reviewer's Comment**
> >
> > I would like to thank the authors for their detailed response. It seems like one of the major issues I raised was a misunderstanding due to an incorrect reference within the text. The new result seems  more reasonable. I am raising my score to 5.

---

### Official Review · Reviewer_BSgg · 2024-07-09

**Soundness:** 3
**Presentation:** 2
**Contribution:** 3
**Rating:** 7
**Confidence:** 3

**Summary:**

This paper proposes an algorithm named BLOCC for solving a particular class of bilevel optimization problems where so-called coupled constraints are present. Coupled constraints are constraints w.r.t. the upper and lower level variables which are applied in addition to simple constraints like bound/ball constraints which are easy to solve for. Bilevel problems with coupled constraints have not been addressed in prior works in this literature, which rely on solving the inner problem and using the implicit function theorem (IFT) to pass gradients through the inner problem.

BLOCC is particular suited for extremely large-scale optimization problems where other existing approaches which use IFT are intractable to solve due to the size of the KKT systems of the inner problems. Notably, BLOCC avoids solving the inner problem directly, instead replacing the inner problem with a simpler problem which only involves the simple constraints (and can be solved with a fast projection instead of requiring an iterative method). The coupled constraint is moved to the objective function of the outer problem, which is also modified as a result to be a saddle point problem. A custom first-order min-max solver is proposed to solve the outer problem and is an additional contribution of the paper.

The convergence properties of the algorithm is established, along with equivalence of the penalty reformulation to the original bilevel problem and BLOCC is evaluated on some interesting bilevel optimization problems against related methods. Particularly appealing are the extremely large-scale transportation network design experiments.

**Strengths:**

The proposed method handles a broader range of bilevel problems compared to prior works in this literature, i.e., large-scale problems with coupled constraints in the inner problem. The characteristics of the algorithm itself (projection free, first-order only) are appealing.

Furthermore, a reasonably detailed convergence analysis for all components of the algorithm are provided. The technical writing overall is of a reasonably high quality.

Finally, the large-scale experiments for the network planning setting help demonstrate that BLOCC is useful to solve large-scale problems in practice which are currently unattainable by methods which use implicit function theorem.

**Weaknesses:**

The main weakness in the paper in my mind is the statement of contribution, it was not immediately obvious why BLOCC is necessary on the first read-through. Only after reading the experimental section did I realize the importance of the algorithm design for large-scale problems.

As a more concrete suggestion, I think contextualising this work in the broader bi-level optimization literature would be beneficial. For instance, the value function reformulation (see [1] for example) is another way to solve an arguably broader class of optimization problems. However, I can see why directly solving the resultant single level problem may be computationally intractable for very large-scale problems using standard solvers, which motivates BLOCC. I would suggest referencing that literature better and emphasizing why the algorithmic properties of BLOCC enable application to large-scale bi-level problems earlier in the paper.

I also think more analysis around the comparison methods LV-HBA and GAM for the large-scale problem in 4.3 is required. For instance, line 330-331 states that for GAM a Hessian must be inverted (presumably as part of using the IFT and differentiating through the inner problem). However, the actual linear system of equations may be solved using something more scalable such as conjugate gradient, which may permit the IFT methods to perform reasonably on this task. Some discussion or a direct comparison would greatly improve the experimental section in the paper.

Finally, a more detailed description of GAM and LV-HBA would be helpful to better understand the differences to BLOCC. For instance, it is claimed that LV-HBA projection steps are expensive, however looking at their paper it seems that the projection is a fairly simple operation (Proj_Z defined in section 2.3 in their paper)? Adding this information will make the paper more self-contained.

[1] Sinha et al. Bilevel Optimization based on Kriging Approximations of Lower Level Optimal Value Function. (2018 IEEE Congress on Evolutionary Computation)

**Questions:**

How easy/robust is the tuning of the penalty parameters?

Typically you just have a set of constraints for a problem. What is the significance of decoupling the inner problem into two sets of constraints, i.e., g^c and \mathcal{Y}(x)? This appears to be a quirk of this literature and not say something like [1].

[1] Sinha et al. Bilevel Optimization based on Kriging Approximations of Lower Level Optimal Value Function. (2018 IEEE Congress on Evolutionary Computation)

**Limitations:**

I believe there is a general lack of discussion of limitations in this paper. It would be nice to understand at the least which parameters are difficult to tune.

---

> ### Author Rebuttal · Authors · 2024-08-07
>
> We thank the reviewer for appreciating our contributions and recognizing our algorithm on large scale experiments "particularly appealing".
>
> **Q1 More clarification for large-scale problems and textualizaing in a broader BLO literature view. e.g. as in value function reformulation [1]**
>
> We appreciate the reviewer's suggestions. We will highlight the significance of BLOCC for large-scale problems earlier in the paper, and demonstrate its robustness through computational cost analysis as in **General Response G1**.
>
> We appreciate the reviewer for directing us to the relevant literature. We will incorporate the referenced literature to show how BLO problems relate to the value function, as outlined in eq. (5)-(8) in the literature. We will emphasize that decoupling $x$ and $y$ allows us to access the value function and its gradient information, which has been crucial for establishing smoothness properties and our max-min oracle.
>
> **Q2 More analysis of the comparison on LV-HBA and GAM for the large-scale problem in 4.3.**
>
> Thank you for the suggestion. We will include the comparison of the computational complexity of the algorithms in Section 4.3. The detailed analysis is provided in **General Response G1**.
>
> **Q3 More detailed description of GAM and LV-HBA. e.g. why projection is hard for LV-HBA (Proj_Z in sec. 2.3 in their paper)**
>
> Thank you for this suggestion. We will include more details of the baselines and include the comparison of computational cost as in **General Response G1**. Regarding the difficulty of projection in LV-HBA, the algorithm includes $\text{Proj}_C$ where $C:=\{\mathcal{X}\times \mathcal{y}: g^c(x,y)\leq 0\}$ is defined in section 2.1 in their paper [69]. This projection is of high cost of ${\cal O}((d_x+d_y)^{3.5})$ as is equivalent to a convex programming problem and it requires additional ${\cal O}(d_x d_y d_c)$ cost on analyzing $g^c(x,y)\leq 0$.
>
> **Q4 How easy/robust is the tuning of the penalty parameters?**
>
> We can fairly conclude that the tunning of the penalty parameter is not very hard and it is very robust.
>
> In practice, $\gamma \approx 10$ can achieves satisfactory lower-level optimality. For simpler cases, such as the toy example in our study, $\gamma = 1$ suffices, as shown in **General Response G2**. For more complex applications, such as our SVM and network planning experiments, we use $\gamma = 12$ and $\gamma = 3, 4, 5$, respectively.
>
> The practical steps for tuning $\gamma$ and the step size $\eta$ are as follows:
>
> 1) If the upper-level objective is non-convex, choose $\gamma \geq l_{f,1}/\alpha_g$ for strong convexity with respect to $y$. If the upper-level problem is convex or if $l_{f,1}$ and $\alpha_g$ are unknown, start with $\gamma \approx 5$. Avoid excessively large values of $\gamma$.
> 2) Apply a small step size $\eta \leq l_{F,1}^{-1}$ and run the algorithm. Increase $\gamma$ if the LL optimality gap is large or if the objective function’s evolution is not smooth and monotonic, which indicates that $\gamma < l_{f,1}/\alpha_g$. Stop when satisfactory lower-level optimality and smooth, decreasing iterates are achieved.
> 3) Try larger $\eta$ that can keep the evolution of the objective function across iterations being smooth and monotonic, therefore fasten the iteration.
>
> Regarding the robustness and stability, we provide sensitivity analysis in **General Response G2**. We can see that the result is robust. Larger $\gamma$ ensures better lower level optimality while it is of no necessity to bring $\gamma$ too large for desired accuracy.
>
> **Q5. What is the significance of decoupling the inner problem into two sets of constraints, i.e., $g^c$ and $\mathcal{Y}$? This appears to be a quirk of this literature and not say something like [1].**
>
> We thank the reviewer for this question. Indeed, the constraints of BLO problems can be expressed as in equations (7)-(8) of [1].
> According to our notation, we use $g^c(x,y)\leq 0$ to denote the LL coupled constraints, $y\in \mathcal{Y}$ to denote uncoupled domain constraints and $\mathcal{Y}(x):=\{y\in \mathcal{Y}:g^c(x,y)\leq 0\}$ to denote the combination of the previous two.
>
> Decoupling these constraints makes it easier to conduct the theoretical analysis for the primal-dual method. Specifically, the functional constraint $g^c(x,y)\leq 0$ can be incorporated into the objective function through a Lagrangian term, while the domain constraint cannot be handled in this way.
>
>
> **Q6. Which parameters are difficult to tune.**
>
> We would say the parameters are generally not difficult to tune. Comparably, the step size $\eta$ and the penalty hyper-parameter $\gamma$ are harder. We have provide practical method and analysis on tunning the $\eta$ and $\gamma$ as in the **answer to Q4**.
>
> Here, we additionally provide some practical tips for choosing the number of iterations required to run BLOCC (Algorithm 1) and the min-max problem (Algorithm 2) ($T$, $T_y$, $T_g$, $T_F$), and the step sizes of the min-max problems ($(\eta_{g,1}, \eta_{g,2})$ for solving $(\mu_g^*(x), y_g^*(x))$ defined in (7), and $(\eta_{F,1}, \eta_{F,2})$ for solving $(\mu^*_F(x), y^*_F(x))$ in (5)).
>
> In practice, we can choose the number of iteration of BLOCC (Algorithm 1) as $T\approx \gamma \epsilon^{-1}$ where $\epsilon$ is the target accuracy. If the constraint $g^c$ is not affine in $y$, we can use the accelerated version of Algorithm 2 as the inner maxmin solver with $T_y \approx \ln(\epsilon^{-1})$ and choose $T_g,T_F\approx \gamma \epsilon^{-0.5}$. For the stepsizes, small enough step size is fine to use in the practice, as we only provides theoretical upper-bound requirement on them. If the constraint $g^c$ is affine in $y$, we can use the non-accelerated version of Algorithm 2 as the inner maxmin solver ($T_y = 1$) and choose $T_g,T_F\approx \ln( \epsilon^{-1})$. Similarly, we can use small enough step sizes.
>
> We hope this can help in addressing your problem.

---

> > ### Comment · Reviewer_BSgg · 2024-08-12
> > **Response to authors**
> >
> > Thank you for your response to my questions. I think the rebuttal suitably addresses my concerns around robustness to hyperparameter tuning and the time complexity of the comparison methods.
> >
> > Again, I think more discussion around the large-scale nature of the setting in the introduction (a few extra sentences, say) would have helped me appreciate the paper more on the first read-through, but this is a minor comment and does not detract from the overall contribution.
> >
> > I will raise my score to reflect the rebuttal.

---

### Official Review · Reviewer_YCHS · 2024-07-13

**Soundness:** 3
**Presentation:** 3
**Contribution:** 3
**Rating:** 6
**Confidence:** 4

**Summary:**

The paper proposed a fully first-order algorithm named BLOCC for the bilevel optimization problems with lower level coupled constraints. They provide convergence theory for the algorithm and demonstrate its effectiveness through numerical experiments in SVM-based model training and infrastructure planning in transportation networks.

**Strengths:**

This paper designs a first-order algorithm for bilevel programming problems with coupling constraints in the lower level, and it does not require joint projection operations. The effectiveness of the algorithm is validated through extensive experiments. The writing is clear and well-structured.

**Weaknesses:**

1. This paper relies on some restrictive assumptions compared to other works, such as the strong convexity of both lower level objective $g$ and constraint $g^c$ with respect to $y$ (Assumption 2) and the Linear Independence Constraint Qualification (LICQ) condition in lower level (Assumption 4). These assumptions reduce the challenge of the bilevel problem considered. It is unclear whether the application problems tested in the numerical experiments satisfy these assumptions. The authors should clearly state in the introduction that their work depends on the strong convexity of the lower level problem.

2. The proposed algorithm requires solving two minimax problems at each iteration, which can be computationally expensive for practical applications.

3. The theoretical results do not clearly state how to choose the step size \eta.

4. The experiments in the paper lack details on the chosen hyperparameters of the algorithm.

5. I have checked the code provided in the Supplementary Material. For the SVM problem, the implementation does not match the proposed algorithm; the minimax problems at each iteration are solved by different solvers than those stated in the paper.

**Questions:**

1. There are some typos in the paper. For example, in line 165, it should be \mathbb{R}^{d_x}_{+}, in line 125, there is an extra “and”.
2. How did the authors choose the iteration number T for the two inner minimax solvers in the experiments?

**Limitations:**

Relies on the strong convexity of the lower level problem.

---

> ### Author Rebuttal · Authors · 2024-08-07
>
> We thank the reviewer for appreciating our work and the constructive review. Our response to your comments follows.
>
> **Q1.1 Stating strongly convexity in introduction**
>
> Thank you for your suggestion. We have noted this in line 43 and will clarify it further in the introduction, such as around line 26.
>
> **Q1.2 Whether the problem is difficult assuming:**
> - **$g$: stongly convex in y**
> - **$g^c$: convex & LICQ in y**
>
> We thank the reviewer for this intersting question. We answer this affirmatively to say even with these assumptions, the constrained BLO remains challenging.
>
> Firstly, the problem remains difficult under the assumption of $g(x,\cdot)$ being strongly convex. Although $y^*_g(x)$ is a singleton, it remains non-differentiable [36], thus impeding using implicit gradient methods.
>
> Secondly, the challenge remains under the assumption on $g^c(x,\cdot)$ because finding $\mu^*_g(x)$ effectively is difficult while it is important for the descent direction for $x$ under $g^c(x,y)\leq 0$. Most existing solvers do not provide convergence results on the distance to optimal points, thus impeding the convergence analysis. Moreover, linear convergence of Algorithm 2 on special case (Theorem 4) is an innovative, first-time established result, and its theoretical analysis is challenging.
>
> **Q1.3 Whether the experiments satisfy the assumptions.**
>
> **G3 Whether the experiments satisfies the assumptions**
>
> Yes, the experiments satisfy the assumptions.
>
> In the **SVM** problem, the LL objective (57b) is strongly convex in LL variable $w$. The coupled constraint $1 - l_{{\rm tr}, i}(z_{{\rm tr}, i}^\top w + b) \leq c_i$ is linear and satisfies LICQ, given full row rank training data (**Assumptions 2, 3, & 4** hold). The smooth and Lipschitz conditions for the LL hold, and for the UL,  $c$ is effectively in a compact set since the training data is finite. Thus, Assumption 1 holds. **Assumption 5** is satisfied as the constraint function is affine in $y$, a sufficient condition as shown in Section 3.2.1.
>
> In the **network design** problem, the Lipschitz and smooth conditions hold for (61a), (61b), and (61c) (**Assumption 1**). The LL problem (61b) is strongly convex on $y \in \mathcal{Y}$. Constraint (61d) uncoupled while the coupled constraint (61c) is linear and satisfies LICQ since $w^{od}$ does not have repeating elements in real-world problems, including this experiment (**Assumptions 2, 3, & 4** hold).
>
> **Q2 Computational cost of inner minimax oracle**
>
> Thank you for raising this question. We provide analysis on computational cost in **General Response G1**.
>
> **Q3. Theoretical results on stepsizes.**
>
> Thank you for your question. We will explicitly present the results to avoid confusion. Below are the detailed theoretical results for stepsize choices.
>
> For BLOCC (Algorithm 1), the stepsize $\eta \leq (l_{F,1})^{-1}$ is presented in Theorem 1 (line 202).
>
> For Algorithm 2, we use $(\eta_{g,1}, \eta_{g,2})$ for solving (7), and $(\eta_{F,1}, \eta_{F,2})$ for (5). In the accelerated version (Theorem 3), we specify $\eta_{g,1} \leq (l_{g,1} + l_{g^c,1})^{-1}$ and $\eta_{F,1} \leq (l_{f,1} + \gamma l_{g,1} + l_{g^c,1})^{-1}$ in line 692 and 693. The choices of $\eta_{g,2}$ and $\eta_{F,2}$ depend on the rule in Lemma 11 (line 667). This, combined with the smoothness parameter for the duality (line 690), leads to $\eta_{g,2} \leq \frac{\alpha_g}{l_{g^c,0}}$ and $\eta_{F,2} \leq \frac{\gamma \alpha_g - l_{f,1}}{l_{g^c,0}}$. In the single-loop version (Theorem 4), the stepsize choices follow Theorem 7. Specifically, $\eta_{g,2} \leq (l_{g,1})^{-1}$, $\eta_{g,1} \leq \alpha_g^2 \epsilon (2 s_{\max} \alpha_g + s_{\max}^2 \epsilon^{-1})^{-1} \eta_{g,1}$, and $\eta_{F,2} = (l_{f,1} + \gamma l_{g,1})^{-1}$, $\eta_{F,2} \leq (\gamma \alpha_g - l_{f,1})^2 \epsilon (2 s_{\max} (\gamma \alpha_g - l_{f,1}) + s_{\max}^2 \epsilon^{-1})^{-1} \eta_{F,1}$, where $\epsilon$ is the target accuracy.
>
> **Q4. Experimental choice of hyper-parameters.**
>
> Thank you for your question. We are sorry for not including this explicitly in appendix. We will clarify it.
>
> For the SVM experiment, we used:
> - **BLOCC:** $\gamma = 12$, stepsize $\eta = 0.01$
> - **LV-HBA [69]:** $\alpha = 0.01$, $\gamma_1 = 0.1$, $\gamma_2 = 0.1$, $\eta = 0.001$
> - **GAM [67]:** $\alpha = 0.05$, $\epsilon = 0.005$
>
> The parameters for LV-HBA and GAM are consistent with those used in their respective papers for SVM experiments.
>
> For the network planning, we experimented with BLOCC using $\gamma = 2, 3, 4$ as  in Table 3, and stepsize $\eta = 1.6 \times 10^{-6}$ as in lines 910, 928, and 953 in the Appendix.
>
>
> **Q5. Another minimax solver in the SVM experiment.**
>
> Thank you for this question. BLOCC is a general framework with convergence ensured (Theorem 2) by calling **any inner max-min oracle** with specific accuracy, such as proposed Algorithm 2 and CVXpy solver as in the SVM problem.
>
> Furthermore, we recognized potential confusion and conducted additional experiments on the SVM example using Algorithm 2 (details in **the Figure in the PDF file**.) The results is similar to the presented one in our paper using CVXpy.
>
> **Q6 Typos**
>
> Thank you for pointing out, we will correct it.
>
> **Q7Iteration number for the two inner minimax solvers?**
>
> Thank you for this question. If $g^c$ is non-affine in y, we use the accelerated version of Algorithm 2 (Theorem 3) and we choose:
> - $T_g, T_F \approx \gamma\epsilon^{-0.5}$ as the number of iteration the two maxmin solvers, and
> - $T_y \approx \ln(\epsilon^{-1})$ as the inner iterate number for $y$
>
> for a target level of accuracy $\epsilon$, e.g. $\epsilon = 1e-2$.
>
> If $g^c$ is affine in $y$, we use the non-accelerated version (Theorem 4) with:
> - $T_g , T_F \approx \ln(\epsilon^{-1})$, and
> - $T_y = 1$, making the solver being single-loop.
>
> In this way, we can achieve $\frac{1}{T}\sum_{t=0}^{T-1}\| G_{\eta}(x_t)\|^2 = {\cal O}(\epsilon),\quad \text{with}~\gamma \approx \epsilon^{-0.5}$.

---

> > ### Comment · Reviewer_YCHS · 2024-08-12
> >
> > Thank you for the detailed response to my questions. I will raise my score.
> >
> > However,  the lower level objective of the SVM problem presented in (57b) does not satisfy the strongly convex assumption, as
> > $w$, $b$ and $\xi$ are all lower level variables.

---

> > > ### Author Response · Authors · 2024-08-13
> > >
> > > We sincerely thank the reviewer for recognizing our work and for providing insightful comments on the SVM experiments.
> > >
> > > In the SVM problem, while the lower-level (LL) objective is strongly convex with respect to $w$, it is indeed only convex, not strongly convex, with respect to $b$ and $\xi$. However, the variable $\xi$ can be effectively eliminated, simplifying the LL problem to:
> > > $$
> > > \min_{w,b}\| w\|^2 \quad \text{s.t.} \quad 1-l_{tr,i}(z_{tr,i}^\top w + b) \leq c_i.
> > > $$
> > >
> > > We acknowledge that the LL objective is not strongly convex with respect to $b$, only convex. It might be proved that it satisfies the Polyak-Łojasiewicz (PL) condition, which we recognize as an important area for future research. We apologize for any confusion this may have caused.
> > >
> > > After reviewing your comments, we conducted additional experiments by incorporating $b$ into $w$, redefining the model parameter as $w' = [w; b]$ and the data as $z' = [z; \mathbf{1}]$, where $\mathbf{1}$ is a vector of ones:
> > > $$
> > > \min_{w'}\| w' \|^2 \quad \text{s.t.} \quad 1-l_{tr,i}(z_{tr,i}'^\top w') \leq c_i.
> > > $$
> > >
> > > The experimental results under this new formulation are similar to the ones we have shown in the paper. Moreover, the new formulation adheres to the assumptions in the paper.
> > >
> > > We deeply appreciate your valuable comments and the opportunity to refine our work.

---

### Official Review · Reviewer_SFco · 2024-07-18

**Soundness:** 3
**Presentation:** 3
**Contribution:** 3
**Rating:** 5
**Confidence:** 4

**Summary:**

This paper presents an algorithm to solve bilevel optimization problems with coupled constraints using a primal-dual-assisted penalty reformulation. The study establishes rigorous convergence theory and demonstrates the algorithm's effectiveness through real-world applications, including SVM model training and transportation network planning.

**Strengths:**

Strength:
- The paper is well-written with clear introduction of the problem, assumptions and technical presentation for their methodology.
- The primal-dual-assisted penalty reformulation and the main optimization algorithm effectively addresses the challenges of this class of problems.
- Comprehensive theoretical analysis are provided to show the theoretical convergency.
- Application examples including SVM and transportation network design, are provided to demonstrate the effectiveness of the proposed algorithm.

**Weaknesses:**

Weakness:
- It would be great if more discussion on the computational complexity and scalability of the proposed method can be provided.
- In the experiments part, the compared baselines are very limited.

**Questions:**

How can the proposed algorithm be seamlessly integrated with other solvers?

---

> ### Author Rebuttal · Authors · 2024-08-07
>
> We thank the reviewer for appreciating our contribution and the constructive review. Our response to your comments follows.
>
> **Q1. Computational complexity and scalability of BLOCC**
>
> Thank you for raising this question. We answer this in **General Response G1**.
>
> **Q2. Limited baselines in the experiments.**
>
> We did not include many baselines because bilevel optimization with coupled constraints is a less explored area with limited existing comparative work. The considered domain constraints impede the use of implicit gradient methods, while coupled constraints complicate finding descent directions for $x.$ Table 1 summarizes existing works addressing coupled constraints. Among them, only LV-HBA [69], GAM [67], and BVFSM [44] handle inequality constraints. We select LV-HBA and GAM as baselines for the SVM training problem, as these algorithms have conducted experimental analysis on this issue in their papers. For the network planning problem, LV-HBA is chosen as the sole baseline, being the only algorithm that addresses both coupled inequality and domain constraints. GAM [67] and BVFSM [44] cannot handle lower-level domain constraints as they require LL stationarity in an unconstrained space.
>
> **Q3 How can BLOCC be integrated with other solvers?**
>
> We thank the reviewer for posing this question. BLOCC (Algorithm 1) is a general framework with convergence established (Theorem 2) by calling **any inner max-min** oracle with a given accuracy. Therefore, a) Algorithm 1 can be seamlessly integrated with any max-min or min-max algorithm that converges to the optimal solutions of the max-min subproblems of BLOCC and b) the analysis in Theorem 2 is valid regardless of the max-min solver used. For instance, for the original simulations carried out in our SVM training example, we used CVXpy, an effective solver for convex programming, to obtain the optimal $y$ and $\mu$. In constrast, Theorems 3 and 4 analyze the computational complexity of the proposed scheme when a particular type of  min-max solver (the one described in Algorithm 2, which is one of the most advanced in the current state of the art) is used.

---

> > ### Comment · Reviewer_SFco · 2024-08-13
> >
> > Thanks for the response, I understand the available baselines for comparison are limited. I will keep my positive review.

---

### Author Rebuttal · Authors · 2024-08-07

# General Response

We sincerely thank the reviewers for their constructive feedback. We appreciate their recognition of our work in tackling the challenging constrained bilevel problem with an efficient algorithm, rigorous convergence theory, and demonstrating effectiveness and robustness through real-world experiments.

The rebuttal is divided into two sections: a general response and four individual responses for each reviewer. In the current section, we focus on two issues that were raised by multiple reviewers and that we feel warrant a more detailed explanation.

**G1. Computational cost of BLOCC.**

We present in the following the computational complexity of our BLOCC algorithm in comparison with LV-HBA [69] and GAM [67] as baslines.

|Method |Iterational cost|Computational cost per iteration|
|-|-|-|
| BLOCC | Outer loop (Algorithm 1) $T = {\cal O}(\epsilon^{-1.5})$ | **General setting (Theorem 3)**:$\tilde{\cal O}(d_c d_x + d_x^2 + \epsilon^{-1}(d_c d_y +d_y^2))$; **Special case (Theorem 4)**: $\tilde{\cal O}(d_x^2+d_y^2+d_c (d_x+d_y))$|
| LV-HBA |${\cal O}(\epsilon^{-3})$|  ${\cal O}(d_xd_yd_c + (d_x+d_y)^{3.5} )$ |
| GAM | Asymptotic |More than ${\cal O}(d_y^3+d_x^2+d_c(d_x+d_y))$|

**BLOCC** is a first-order method with gradient calculation costs of $\mathcal{O}(d_x d_c)$ and $\mathcal{O}(d_y d_c)$. Given the assumption (in line 28) that $\mathcal{X}$ and $\mathcal{Y}$ are "easy to project, such as the Euclidean ball," we presume that projections can be performed using projection matrices or simple functions. Thus, the projection cost is generally $\mathcal{O}(d_x^2)$ and $\mathcal{O}(d_y^2)$. In our SVM and network planning experiments, projections are simpler (e.g., truncation), resulting in $\mathcal{O}(d_x)$ and $\mathcal{O}(d_y)$ costs. Thus, the cost per iteration of BLOCC (Algorithm 1) is $\tilde{\cal O}(d_c d_x + \epsilon^{-1}d_c d_y)$ for the general setting (Theorem 3) and $\tilde{\cal O}( d_c (d_x+d_y))$ for the special case of affine coupling constraints (Theorem 4).

**LV-HBA**, also a first-order method, has similar gradient calculation costs. However, its projection onto $\{\mathcal{X} \times \mathcal{Y} : g^c(x, y) \leq 0\}$ is expensive, with a complexity of $\mathcal{O}((d_x + d_y)^{3.5})$, and evaluating $g^c(x, y) \leq 0$ adds $\mathcal{O}(d_x d_y d_c)$ as it requires $d_c$ inequality judgement on functions taking input dimension $d_x,d_y$.

**GAM** lacks an explicit algorithm for lower-level optimality and Lagrange multipliers. Even if we omit this, calculating $\nabla_{yy}^2 g$ and its inverse incurs a cost of $\mathcal{O}(d_y^3)$. Additionally, it has cost ${\cal O}(d_x^2+d_c(d_x+d_y))$ for projection onto $\mathcal{X}$ and calculating gradients.

We can see that BLOCC stands out with the lowest computational cost in both iterational and overall complexity thanks to its first-order and joint-projection-free nature. Consequently, BLOCC is particularly well-suited for large-scale applications with high-dimensional parameters.

**G2. Sensitivity analysis of the hyperparameters.**

Regarding the selection and impact of hyper-parameters on the algorithm's performance, we conducted an ablation study on various values of the two critical parameters, $\gamma$ and $\eta$, and measured their effects on the optimal value and computational time.

We present in the following the results of the toy example in section 4.1 of our paper. We also present the results on network planning problem in the *attached PDF file*. The real-world result is consistent with the one on toy example.

|$\eta$ \ $\gamma$ |0.001|0.01|0.1 | 1.0|
|-|-|-|-|-|
| 0.001 | 0.028 ± 0.042| 0.035 ± 0.076| 0.027 ± 0.064| 0.000 ± 0.000|
| | 3.314 ± 0.042| 3.186 ± 0.076| 2.049 ± 0.064    |1.967 ± 0.000|
| 0.01| 0.020 ± 0.031| 0.020 ± 0.041| 0.009 ± 0.019| 0.000 ± 0.000|
|| 2.242 ± 0.031| 1.575 ± 0.041| 1.118 ± 0.019|0.585 ± 0.000|
| 0.1   | 0.006 ± 0.010| 0.017 ± 0.027| 0.011 ± 0.033|0.000 ± 0.000|
|| 1.616 ± 0.010| 1.475 ± 0.027|1.257 ± 0.033|0.383 ± 0.000|
| 1.0| 0.023 ± 0.019  | 0.030 ± 0.022| 0.020 ± 0.045|0.000 ± 0.000|
|| 7.118 ± 0.019    | 4.570 ± 0.022| 3.477 ± 0.045| 1.645 ± 0.000|

*Table 1: Sensitivity analysis of the toy example in our paper. (Mean ± SD on 40 simulations) **Top line** in each cell represents the optimality gap* $|y_g^* (x_T) - y_{F,T}|$*, while **bottom line** represents the time required for the algorithm to converge.*

From Table 1, we can draw the following empirical observations:

1) *Smaller $\eta$ is needed to satisfy $\eta \leq \frac{1}{l_{F,1}}$ to ensure convergence if $\gamma$ is chosen larger*. The algorithm tends to converge for different $\gamma$ values when $\eta$ is small enough. This is because the smoothness constant $l_{F,1}$ of $F_{\gamma}(x)$ increases with $\gamma$ (Lemma 2), and the algorithm is guaranteed to converge if $\eta \leq \frac{1}{l_{F,1}}$ (Theorem 1).

2) *Larger values of $\gamma$ bring the upper-level objectives closer to optimal*. Theorem 1 illustrated that larger $\gamma$ improves the accuracy of the LL optimality. Since the obtained solution $y_{F,T}$ is closer to $y_g^*(x_T)$ for larger $\gamma$ values, the distance between the $f(x_T,y_{F,T})$ and $f(x_T,y^*_g(x_T))$ will be closer as well.

3) *For a sufficiently small fixed $\eta$, larger $\gamma$ lead to faster convergence*. This implies that *smaller $\eta\leq \frac{1}{l_{F,1}}$ choice due to larger $\gamma$ will not significantly dampen the convergence time*. This is because a large value of $\gamma$ increases $l_{F,1}$, sharpening the function $F_{\gamma}(x)$ and its gradient will be larger for most points. Thus, the gradient update $\eta \nabla F_{\gamma}(x_t)$ will not be very small and thus won't make the convergence slower.

In summary, while our theoretical results highlight the impact of $\gamma$ and $\eta$ on the optimality gap and computational complexity, we believe the experimental results summarized in the tables provide additional evidence and valuable insights.

---

### Decision · Program_Chairs · 2024-09-25

**Decision:**

Accept (poster)

**Comment:**

This paper addresses the complex issue of bilevel optimization problems with coupled constraints, a topic that has been relatively underexplored in existing literature. The authors propose a first-order algorithm, BLOCC, specifically designed to tackle these challenging problems. The paper includes a rigorous theoretical analysis supporting the convergence of the proposed algorithm and demonstrates its effectiveness through applications in support vector machine (SVM) model training and infrastructure planning within transportation networks.

The paper is well-structured and clearly written, with a comprehensive introduction to the problem, assumptions, and methodology. The proposed primal-dual-assisted penalty reformulation and optimization algorithm effectively address the difficulties associated with bilevel problems involving coupled constraints. The theoretical analysis provided is thorough and establishes the convergence of the algorithm.
All reviewers agree that the paper is acceptable pending revisions. The authors are advised to revise the paper in accordance with the reviewers' feedback provided in the rebuttal.